# Hepatic zonation determines tumorigenic potential of mutant β-catenin

Alexander Raven[1], Kathryn Gilroy[1], Hu Jin[2], Joseph A. Waldron[1], Holly Leslie[3], June Munro[1], Holly Hall[1], Rachel A. Ridgway[1], Catriona A. Ford[1], Doga C. Gulhan[2], Nikola Vlahov[1], Megan L. Mills[1], Andrew Hartley[1], Eve Anderson[1], Sheila Bryson[1], Nathalie Sphyris[1], Miryam Müller[1], Stephanie May[1,3], Barbara Cadden[1], Colin Nixon[1], Scott H. Waddell[4], Rachel Guest[4,5], Luke Boulter[4,5], Nick Barker[6,7], Hans Clevers[8,9], Hao Zhu[10,11,12], Johanna Ivaska[13], Douglas Strathdee[1], Crispin J. Miller[1,3,5], Nigel B. Jamieson[3,5], Martin Bushell[1,3], Peter J. Park[2], Thomas G. Bird[1,3,5,14] & Owen J. Sansom[1,3,5 ✉]

Oncogenic mutations in phenotypically normal tissue are common across adult organs[1,2]. This suggests that multiple events need to converge to drive tumorigenesis and that many processes such as tissue differentiation may protect against carcinogenesis. WNT–β-catenin signalling maintains zonal differentiation during liver homeostasis[3,4]. However, the *CTNNB1* oncogene—encoding β-catenin—is also frequently mutated in hepatocellular carcinoma, resulting in aberrant WNT signalling that promotes cell growth[5,6]. Here we investigated the antagonistic interplay between WNT-driven growth and differentiation in zonal hepatocyte populations during liver tumorigenesis. We found that β-catenin mutations co-operate with exogenous MYC expression to drive a proliferative translatome. Differentiation of hepatocytes to an extreme zone 3 fate suppressed this proliferative translatome. Furthermore, a GLUL and *Lgr5*-positive perivenous subpopulation of zone 3 hepatocytes were refractory to WNT-induced and MYC-induced tumorigenesis. However, when mutant *CTNNB1* and *MYC* alleles were activated sporadically across the liver lobule, a subset of mutant hepatocytes became proliferative and tumorigenic. These early lesions were characterized by reduced WNT pathway activation and elevated MAPK signalling, which suppresses zone 3 differentiation. The proliferative lesions were also dependent on IGFBP2–mTOR–cyclin D1 pathway signalling, in which inhibition of either IGFBP2 or mTOR suppressed proliferation and tumorigenesis. Therefore, we propose that zonal identity dictates hepatocyte susceptibility to WNT-driven tumorigenesis and that escaping WNT-induced differentiation is essential for liver cancer.

In the homeostatic setting, where most hepatocytes are quiescent, a decreasing gradient of WNT pathway activity from the central vein to the portal node acts as a major regulator of hepatic zonation[3,4]—a phenomenon in which hepatocytes perform distinct metabolic functions based on their lobular location. As such, the liver lobule may be divided into three zones along the portal node–central vein axis, with hepatocytes termed accordingly: zone 1 (periportal), zone 2 (midlobular) and zone 3 (pericentral). This configuration restricts the activity of nuclear β-catenin to the pericentral, zone 3 hepatocytes within the liver lobule, driving their maturation and compartmentalizing the expression of WNT-target genes. By contrast, during regeneration[7] and hepatocellular carcinoma (HCC)[5,6], WNT pathway activity drives hepatocellular growth. Further complicating our understanding of WNT-driven HCC is the selection of point mutations in exon 3 of *CTNNB1* over other types of WNT pathway-activating mutation, such as *APC* truncations or *RNF43/ZNRF3* loss[8], that have nevertheless been shown to be tumorigenic in the mouse[9–12]. Understanding how *CTNNB1* mutations contribute to tumorigenesis and alter zonal specification will be important

[1]Cancer Research UK Scotland Institute, Glasgow, UK. [2]Department of Biomedical Informatics, Harvard Medical School, Boston, MA, USA. [3]School of Cancer Sciences, University of Glasgow, Glasgow, UK. [4]MRC Human Genetics Unit, Institute of Genetics and Cancer, University of Edinburgh, Edinburgh, UK. [5]CRUK Scotland Centre, Glasgow, UK. [6]Institute of Molecular and Cell Biology (IMCB), Agency for Science, Technology and Research (A*STAR), Singapore, Singapore. [7]Department of Physiology, Yong Loo Lin School of Medicine, National University of Singapore, Singapore, Singapore. [8]Pharma, Research and Early Development (pRED) of F. Hoffmann-La Roche Ltd, Basel, Switzerland. [9]Oncode Institute, Hubrecht Institute, Royal Netherlands Academy of Arts and Sciences and University Medical Center, Utrecht, the Netherlands. [10]Children's Research Institute and Children's Research Institute Mouse Genome Engineering Core, University of Texas Southwestern Medical Center, Dallas, TX, USA. [11]Simmons Comprehensive Cancer Center, University of Texas Southwestern Medical Center, Dallas, TX, USA. [12]Department of Pediatrics, University of Texas Southwestern Medical Center, Dallas, TX, USA. [13]Turku Bioscience Centre, University of Turku and Åbo Akademi University, Turku, Finland. [14]MRC Centre for Inflammation Research, The Queen's Medical Research Institute, University of Edinburgh, Edinburgh, UK. ✉e-mail: o.sansom@crukscotlandinstitute.ac.uk

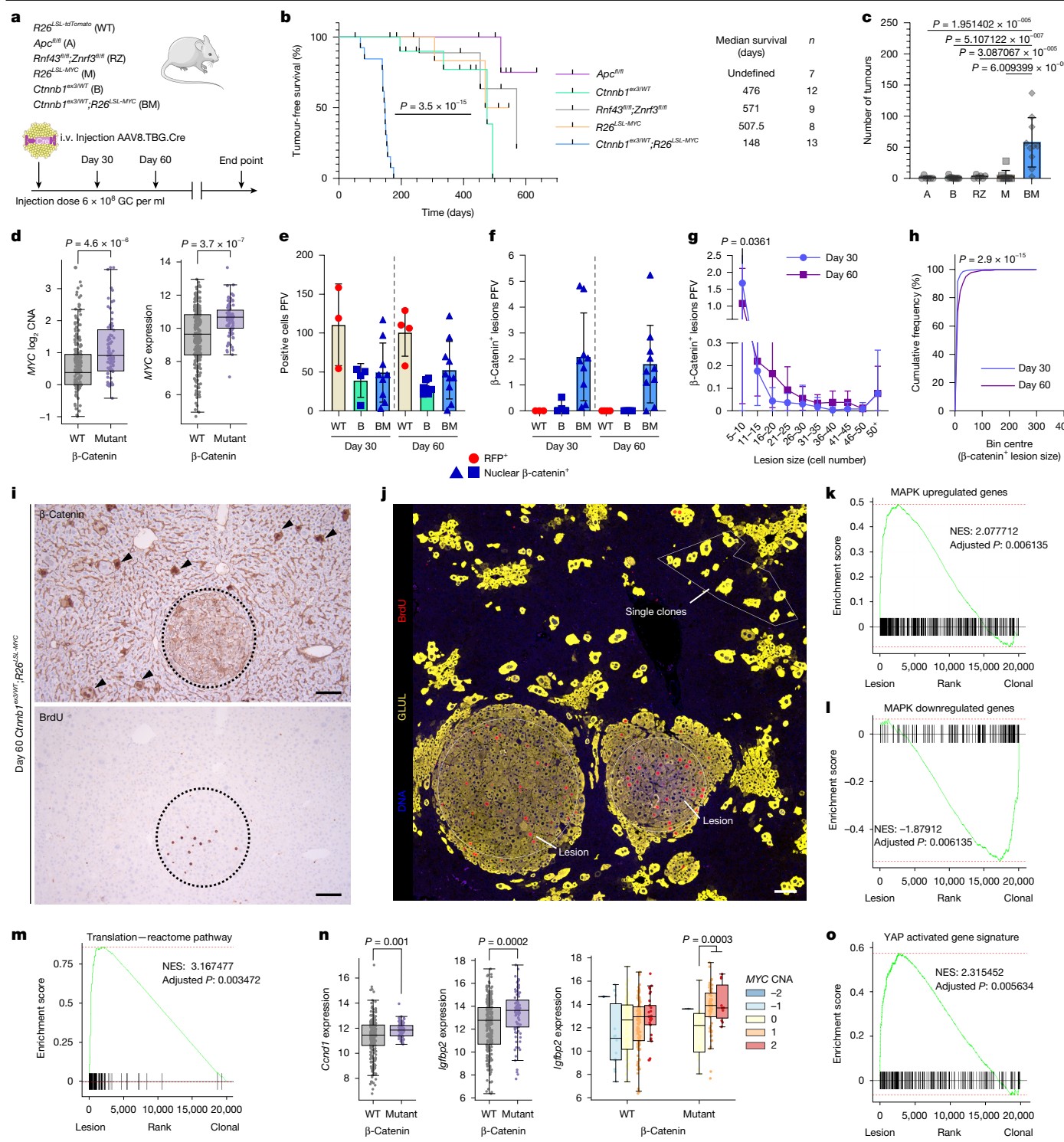

**Fig. 1** | See next page for caption.

as previous lineage-tracing studies have demonstrated zonal differences to hepatocyte growth in the homeostatic liver[13–15] and the cell of origin of HCC[16–18].

## MYC is required for WNT-driven HCC

We sought to investigate the tumorigenic potential of common WNT pathway mutations in the mouse liver. Sporadic WNT pathway mutations in the hepatocyte epithelium were only weakly oncogenic,

requiring a long latency period to elicit a low tumour burden (Fig. 1a–c and Extended Data Fig. 1a–d). Unlike other tissues that commonly acquire WNT pathway mutations, WNT activation via *Apc* loss did not increase endogenous *Myc* expression in the mouse liver (Extended Data Fig. 1e). In human HCC, 81% of *CTNNB1*-mutated tumours exhibit *MYC* copy number gain along with significantly increased *MYC* gene expression (Fig. 1d). Recapitulating previous findings[19], expression of an *R26^LSL-MYC* transgene in *Ctnnb1^ex3/WT* hepatocytes enhanced tumorigenesis (Fig. 1b,c and Extended Data Fig. 1a–d), the addition of *MYC* upregulated

**Fig. 1 | BM clones form lesions with reduced WNT pathway activation.**
**a**, Mouse model recombining alleles at a low clonal density in the hepatocyte epithelium. i.v., intravenous. The illustrations of the mouse and adenovirus were adapted from Medical Art Servier (https://servier.com) under a CC BY 4.0 licence. **b**, Tumour-free survival. A log-rank (Mantel–Cox) test was used. See Methods for the censoring criteria (censors are denoted by vertical tick marks). **c**, Tumour scoring. For biological replicates, $n = 6$ for A, $n = 12$ for B, $n = 6$ for RZ, $n = 9$ for M and $n = 10$ for BM. The bars are mean ± s.d. One-way analysis of variance (ANOVA) with Tukey's multiple comparisons test was used to determine significance. **d**, The Cancer Genome Atlas Liver Hepatocellular Carcinoma (TCGA-LIHC) data comparing *CTNNB1* activating mutations to *MYC* copy number alterations (CNAs) and mRNA expression. *P* values were calculated with a two-sided Student's *t*-tests. Each dot represents a unique HCC tumour ($n = 346$ HCCs). **e**–**h**, Quantification of recombined hepatocytes (**e**) and mutant lesions (**f**) per field of view (PFV) at days 30 and 60 post-induction. Mutant lesion size (**g**) and distribution (**h**) are also quantified. For biological replicates, for day 30, $n = 3$ for WT, $n = 4$ for B and $n = 9$ for BM; and for day 60, $n = 4$ for WT, $n = 7$ for B and $n = 10$ for BM. The bars are mean ± s.d. Two-way ANOVA with Sidak's multiple comparisons test and two-sided Kolmogorov–Smirnov cumulative frequency test were used to determine significance. **i**, Representative image of β-catenin and BrdU IHC in day 60 BM livers ($n = 5$). The black arrowheads highlight single hepatocytes with high β-catenin staining; and the dashed outline highlights a lesion. **j**, Representative image of an immunofluorescent mask used to select regions for spatial transcriptomics. BrdU (red), GLUL (yellow) and DNA (blue) are shown. **k**–**m**, Spatial transcriptomics gene set enrichment analysis (GSEA) for MAPK upregulated genes (**k**), MAPK downregulated genes (**l**) and translation reactome pathway (**m**); MAPK GSEA is from a day 10 *Braf^{V600E}*-mutated liver. NES, normalized enrichment score. **n**, TCGA-LIHC data comparing *CTNNB1* activating mutations and *MYC* CNAs (GISTIC) to *CCND1* and *IGFBP2* expression; copy number 1 (gain) or 2 (high-level amplification) is considered to have copy gains. $n = 346$ HCC tumours. *P* values were calculated with a two-sided Student's *t*-test. **o**, GSEA from the spatial transcriptomics assay, consensus, liver-specific, YAP gene targets[38]. For all boxplots, the median (central line) and 25th and 75th percentiles (box) are shown, and the whiskers represent the maximum and minimum of non-outlier values within 1.5× the interquartile range. Data beyond whiskers are outliers. For all GSEA plots, the NES was calculated by normalizing to the mean enrichment of random samples, and two-sided permutation was calculated with Benjamini–Hochberg multiple test correction. Scale bars, 100 μm.

many gene programs also commonly found in the *Apc*-deficient intestine (Extended Data Fig. 1f). The shared gene sets were all associated with mRNA translation and protein synthesis pointing to the need for an oncogenic translatome in WNT-driven cancer.

## Tumorigenesis requires optimal WNT signalling

To investigate how sporadically mutated *Ctnnb1^{ex3/WT}*;*R26^{LSL-MYC}* hepatocytes progress to tumours, we examined *Ctnnb1^{ex3/WT}*;*R26^{LSL-MYC}* mosaic livers 30 and 60 days after treatment with a low viral titre of AAV8.TBG. Cre (Fig. 1a). Dispersed, recombined hepatocytes were detected in *R26^{LSL-tdTomato}*, *Ctnnb1^{ex3/WT}* and *Ctnnb1^{ex3/WT}*;*R26^{LSL-MYC}* livers (Fig. 1e and Extended Data Fig. 1b). *Ctnnb1^{ex3/WT}*;*R26^{LSL-MYC}*-mutant hepatocytes were also found in clusters of more than five adjacent hepatocytes (Fig. 1f). The size distribution of these clusters (hereafter lesions) changed between day 30 and day 60, with a trend towards an increased number of mid-sized lesions (16–25 cells in size) and significantly fewer smaller lesions (5–10 cells in size; Fig. 1g,h). The *Ctnnb1^{ex3/WT}*;*R26^{LSL-MYC}* lesions were in proximity to single-mutant clones that had failed to expand (Extended Data Fig. 1g). There was no indication that the single-mutant clones were senescent as they were negative for both p21 and p16 (Extended Data Fig. 1g). The *Ctnnb1^{ex3/WT}*;*R26^{LSL-MYC}*-mutant proliferative lesions were distinguished by reduced WNT pathway activation, as indicated by decreased nuclear *Ctnnb1* positivity, when compared with neighbouring single *Ctnnb1^{ex3/WT}*;*R26^{LSL-MYC}*-mutant hepatocytes (Fig. 1i). This feature did not extend to *MYC* expression, which remained unchanged between *Ctnnb1^{ex3/WT}*;*R26^{LSL-MYC}* single clones and proliferative lesions (Extended Data Fig. 1h). We performed a spatial transcriptomics assay on day 60 *Ctnnb1^{ex3/WT}*;*R26^{LSL-MYC}* livers to investigate differences between single non-proliferative mutant hepatocytes and proliferative mutant lesions (Fig. 1j). Proliferative lesions were significantly enriched for the expression of gene sets associated with active MAPK signalling, mRNA translation and protein synthesis when compared with single-mutant hepatocytes (Fig. 1k–m and Extended Data Fig. 1i). Immunohistochemistry (IHC) confirmed the presence of factors associated with mTOR activity and increased mRNA translation within the proliferative lesions (Extended Data Fig. 1j). Previously, zone 2-driven homeostatic proliferation was linked to IGFBP2, mTOR and CCND1 activity[13], therefore we decided to investigate whether these factors were also relevant to WNT-driven tumorigenesis. IHC confirmed the expression of CCND1 in proliferative lesions (Extended Data Fig. 1k). Moreover, *IGFBP2* was significantly upregulated in human *CTNNB1*;*MYC*-mutated HCC as well as end point *Ctnnb1^{ex3/WT}*;*R26^{LSL-MYC}* mouse liver tumours (Fig. 1n and Extended Data Fig. 2a–c). Next, we compared the WNT-driven gene signatures in the mutant lesions and single clones with *CTNNB1*;*MYC*-mutated HCC. We found that the WNT-driven gene expression signature in the *Ctnnb1^{ex3/WT}*;*R26^{LSL-MYC}* proliferative lesions strongly correlated with the *CTNNB1*;*MYC*-mutated HCC and Hoshida subtypes S2 (characterized by proliferation and MYC activation) and S3 (associated with hepatocyte differentiation)[20], whereas the single clones more closely resembled *CTNNB1*-mutated HCC with normal *MYC* copy number (Extended Data Fig. 2d–g). Spatial transcriptomic profiling of the lesions also revealed an increase in YAP activity (Fig. 1o), which can influence cell fate[21] and antagonize WNT signalling[22]. There was also increased expression of E-cadherin in these lesions (Extended Data Fig. 2h), which can modulate oncogenic β-catenin signalling in colorectal cancer[23]. Both features could explain why the proliferative lesions show reduced WNT pathway activation.

To functionally validate the major findings from the day 60 *Ctnnb1^{ex3/WT}*;*R26^{LSL-MYC}* spatial transcriptomic assay, we inhibited mTOR activity in *Ctnnb1^{ex3/WT}*;*R26^{LSL-MYC}* hepatocytes with rapamycin between days 30 and 60 (Fig. 2a) and observed a significant reduction in the number of lesions (Fig. 2b,c). Furthermore, transient inhibition of mTOR between day 30 and day 60 (Fig. 2d) profoundly impacted end-stage tumour formation, compromising the ability of *Ctnnb1^{ex3/WT}*;*R26^{LSL-MYC}* hepatocytes to form tumours, resulting in a significant reduction in tumour number and an extension in survival (Fig. 2e). Disrupting expression of the zone 2 factor *Igfbp2*, using AAV8.U6.shRNA-*Igfbp2*, reduced the number of *Ctnnb1^{ex3/WT}*;*R26^{LSL-MYC}* lesions at day 60 (Fig. 2f–h). Finally, *Yap* and *Taz* deletion in *Ctnnb1^{ex3/WT}*;*R26^{LSL-MYC}* hepatocytes prevented lesion formation 60 days post-induction, with only occasional intensely *Ctnnb1*-positive single-cell clones visible in the *Yap^{fl/fl}*;*Taz^{fl/fl}* liver samples (Fig. 2i–k). Collectively, these data demonstrate that *Ctnnb1^{ex3/WT}*;*R26^{LSL-MYC}*-mutant hepatocytes need to dampen oncogenic WNT activation and engage the zone 2-specific IGFBP2–mTOR–CCND1 pro-growth pathway for tumorigenesis to proceed. This prompted us to examine how distinct zonal hepatocyte populations respond to combined WNT and MYC activation to initiate tumorigenesis.

## GLUL⁺ hepatocytes are less permissive

To test how zonally patterned hepatocytes respond to *Ctnnb1^{ex3/WT}*;*R26^{LSL-MYC}*-driven growth, we acutely recombined the respective alleles throughout the hepatocyte epithelium (Fig. 3a). Pan-hepatocellular *Ctnnb1^{ex3}* and *R26^{LSL-MYC}* recombination resulted in significant hepatocyte proliferation and hepatomegaly (Fig. 3b,c and Extended Data Fig. 3a). Over time (days 4–10 post-induction), two notable features became

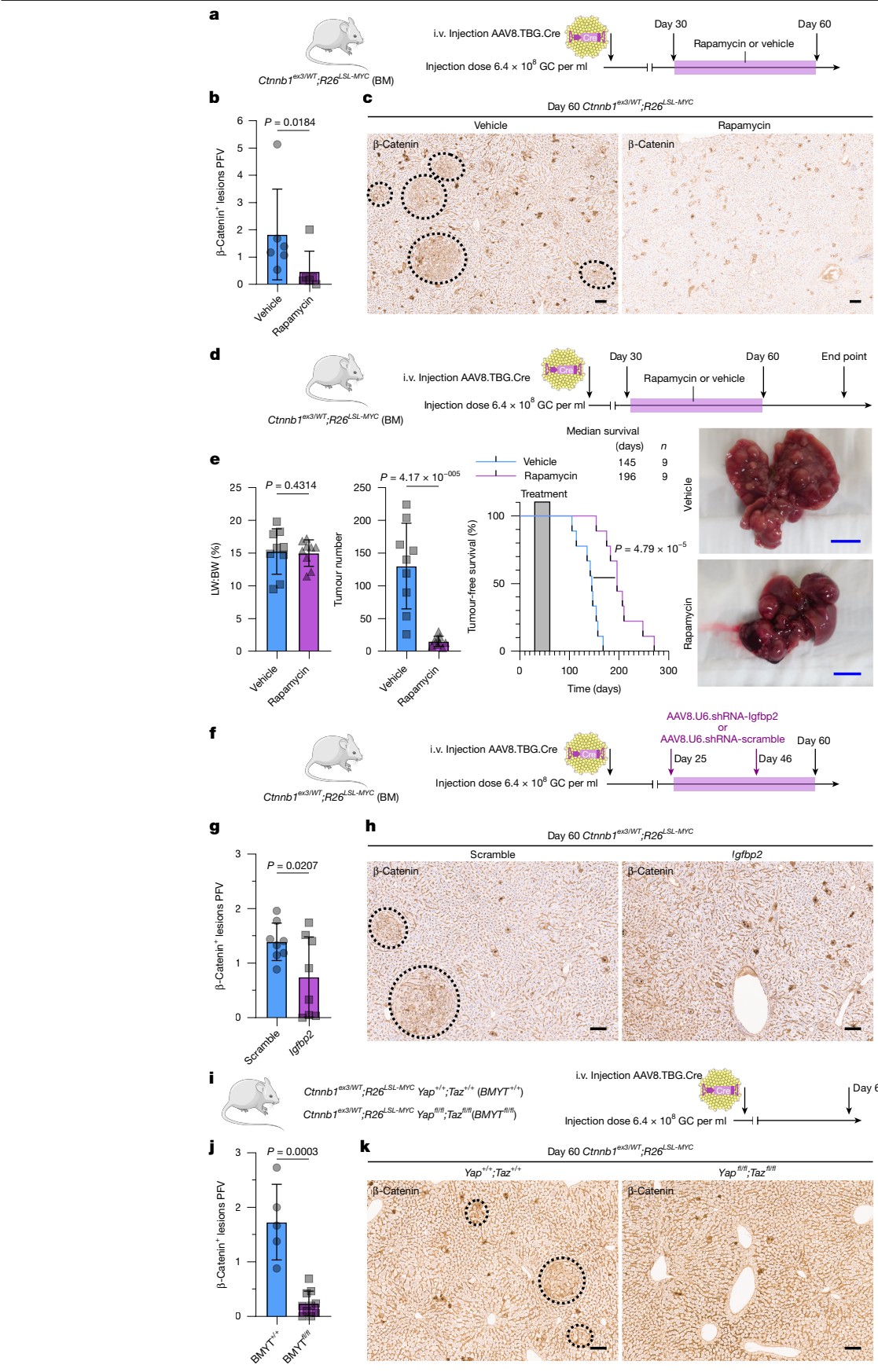

**Fig. 2 |** See next page for caption.

**Fig. 2 | Early BM lesions reactivate the zone 2-specific IGFBP2–mTOR–CCND1 axis. a**, Schematic to treat BM lesions with rapamycin. **b**, Quantification of β-catenin⁺ lesions PFV. The bars are mean ± s.d. One-tailed Mann–Whitney test was used to determine significance. For biological replicates, *n* = 6 mice per treatment. **c**, Representative image of β-catenin IHC (*n* = 6). **d**, Schematic to treat BM lesions with rapamycin for a transient 30-day period. **e**, Liver-to-body weight ratio (LW:BW), tumour scoring, tumour-free survival and liver macroscopic images of BM livers treated with rapamycin or vehicle. A log-rank (Mantel–Cox) test and one-tailed Student's *t*-tests were used to determine significance. The bars are mean ± s.d. For biological replicates, *n* = 9 for vehicle and *n* = 9 for rapamycin. **f**, Schematic to treat BM lesions with AAV8.U6.*Igfbp2*-shRNA or AAV8.U6.scramble-shRNA. **g**, Quantification of β-catenin⁺ lesions PFV.

The bars are mean ± s.d. A one-tailed Student's *t*-test was used to determine significance. For biological replicates, *n* = 8 mice per treatment. **h**, Representative image of β-catenin IHC (*n* = 8). **i**, Schematic and time course of experiment recombining *Ctnnb1*$^{ex3}$-mutant, *MYC*-mutant and *Yap*$^{fl/fl}$*Taz*$^{fl/fl}$-mutant alleles. The illustrations of the mouse and adenovirus in panels **a,d,f,i** were adapted from Medical Art Servier (https://servier.com) under a CC BY 4.0 licence. **j**, Quantification of β-catenin⁺ lesions PFV. The bars are mean ± s.d. A one-tailed Mann–Whitney test was used to determine significance. For biological replicates, *n* = 5 for *Yap*$^{+/+}$*Taz*$^{+/+}$ and *n* = 10 for *Yap*$^{fl/fl}$*Taz*$^{fl/fl}$. **k**, Representative image of β-catenin IHC (*n* = 5). The dashed lines mark β-catenin⁺ lesions (**c,h,k**). Scale bars, 100 µm (**c,h,k**), 1 cm (**e**).

---

apparent: first, the increase in hepatocyte proliferation was transient, occurring predominantly in GLUL⁻ zone 1 and zone 2 hepatocytes around day 4 and diminishing by day 10 post-induction. Second, this transient proliferative response was succeeded by marked upregulation of the zone 3 GLUL-expressing domain along the liver-lobule axis, underpinned by a significant increase in the levels of zone 3 gene transcripts (Fig. 3b–d and Extended Data Fig. 3b–e). Overexpression of *MYC* alone increased hepatocyte proliferation across all zones of the liver, with equal rates of proliferation in GLUL⁺ and GLUL⁻ hepatocytes (Extended Data Fig. 3e) but was not tumorigenic per se (Fig. 1a–c and Extended Data Fig. 1c,d). Of note, hepatocyte apoptosis was elevated in *R26*$^{LSL-MYC}$ livers but was suppressed in a *Ctnnb1*$^{ex3/WT}$ background (Extended Data Fig. 4a–c). This phenomenon was also observed in mutant *CTNNB1;MYC* samples from patients with HCC (Extended Data Fig. 4d), suggesting that synergistic WNT pathway and MYC mutations can confer both proliferative and anti-apoptotic benefits in liver tumorigenesis.

To examine zonal differences in mutant β-catenin-driven proliferation, we used a whole-mouse transcriptome spatial transcriptomics assay on day 4 *Ctnnb1*$^{ex3/WT}$ liver (Extended Data Fig. 4e). Although WNT pathway activation, as assessed through nuclear accumulation of β-catenin, was uniform across the liver lobule, there was a clear transcriptional separation between GLUL⁺ (combined GLUL high and low) and GLUL⁻ (combined GLUL-adj and portal node-adj) regions (Extended Data Fig. 4f–h), with the latter region showing significantly elevated expression of *Igfbp2* (Fig. 3e). Consistent with the notion that IGFBP2 is required for zone 2-driven homeostatic growth[13] and that it is not a direct WNT-target gene, its expression was lost by day 10 in the *Ctnnb1*$^{ex3}$-mutated series of livers (Extended Data Fig. 5a–f), coinciding with a reduction in hepatocyte proliferation (Fig. 3c). To functionally confirm that an IGFBP2, mTOR and CCND1 signalling axis was contributing to the increased proliferation in *Ctnnb1*$^{ex3/WT}$;*R26*$^{LSL-MYC}$ livers at day 4, we treated the mice with the mTOR inhibitor rapamycin, which significantly reduced proliferation (Fig. 3f and Extended Data Fig. 3f). We also induced *Ctnnb1*$^{ex3/WT}$;*R26*$^{LSL-MYC}$ recombination in an *Ifgpb2*-deficient mouse model and found that hepatocyte proliferation was again significantly reduced (Fig. 3g and Extended Data Fig. 3g). Attempts to overexpress IGFBP2 and super-stimulate mTOR activity via AKT activation in zone 3 hepatocytes did not promote proliferation, suggesting that these factors are required for oncogenic growth but are not sufficient to induce growth in terminally differentiated zone 3 hepatocytes (Extended Data Fig. 3h–s). Together, these data suggest that the same mechanism deployed during zone 2 homeostasis is permissive to *Ctnnb1*$^{ex3}$-driven growth, but that WNT-driven differentiation to a zone 3 fate blocks this effect.

### WNT–MYC support a proliferative translatome

mTOR-mediated mRNA translation is a key component of oncogenic WNT-driven growth in the intestine[24] and ribosomal genes expressed in zone 2 promote liver regeneration[25,26]. To investigate the role of mRNA translation in WNT-mutated liver, we performed ribosome profiling on AAV8.TBG.Cre-treated livers at day 4 during the proliferative phase and day 10 when a sizeable fraction of the hepatocyte population had differentiated to a zone 3 fate. *R26*$^{LSL-MYC}$ activation increased the translation efficiency of select mRNA transcripts (Extended Data Fig. 6a–d), which was further enhanced when combined with a *Ctnnb1*$^{ex3/WT}$ mutation (Fig. 3h). Transcripts associated with cell division and growth were preferentially translated in day 4 highly proliferative *Ctnnb1*$^{ex3/WT}$;*R26*$^{LSL-MYC}$ livers (Fig. 3i). The day 4 pro-growth translatome was downregulated in the *Ctnnb1*$^{ex3/WT}$;*R26*$^{LSL-MYC}$ livers at day 10, when the hepatic lobule had differentiated to a predominantly zone 3 fate (Fig. 3j and Extended Data Fig. 6e), coincident with a global reduction in ribosome occupancy (Extended Data Fig. 6f). Together, these data reveal a targeted RNA translation programme that supports the production of proteins required for the zone 1 and zone 2 hepatocyte proliferation 4 days after oncogenic WNT and MYC activation.

### Lgr5⁺ hepatocytes resist WNT-driven HCC

GLUL⁺ hepatocytes were refractory to WNT-driven proliferation. In the hepatic lobule GLUL⁺ hepatocytes reside in a single, perivenous, sublayer of zone 3, and do not occupy the entire zone. To determine whether a particular sublayer of zone 3 was resistant to oncogenic WNT, we used a range of genetically engineered mouse models to induce Cre expression in a zonally restricted manner across the hepatic lobule (Extended Data Fig. 7a). Spatial profiling of hepatocyte proliferation revealed that *R26*$^{LSL-MYC}$ recombination induced proliferation in all regions of the liver (Extended Data Fig. 8a–i), consistent with the acute AAV8.TBG.Cre model, where *MYC*-induced proliferation did not display a zonal bias (Extended Data Fig. 3e). In the absence of the *R26*$^{LSL-MYC}$ transgene, the *Ctnnb1*$^{ex3/WT}$ livers exhibited increased hepatocyte proliferation in the first 100 µm adjacent to the GLUL⁺ hepatocytes, but not in the GLUL⁺*Lgr5*⁺ hepatocytes surrounding the central vein (Extended Data Fig. 7b–f), coinciding with the region where *Igfbp2* expression begins (Extended Data Fig. 7g,h). This proliferative response was consistent in each model despite the use of different zonal Cres to induce recombination. To confirm that *Lgr5*-specific *Ctnnb1*$^{ex3/WT}$ and *R26*$^{LSL-MYC}$ activation did not have a delayed oncogenic response, we examined the livers 20 days post-induction and could not detect hepatomegaly, aberrant hepatocyte proliferation or expansion of mutant hepatocytes at the central vein (Extended Data Fig. 8j–m). Together, these data suggest that an extreme zone 3 *Lgr5*⁺ hepatocyte fate is not permissive to oncogenic WNT-driven growth.

### MAPK antagonizes WNT-driven differentiation

To examine how zonation affects tumorigenesis, we introduced various oncogenic mutations into *Lgr5*⁺ central vein hepatocytes and examined their tumorigenic potential (Fig. 4a and Extended Data Fig. 8n). The central venous, *Lgr5*⁺ *Ctnnb1*$^{ex3/WT}$;*R26*$^{LSL-MYC}$ mutants did

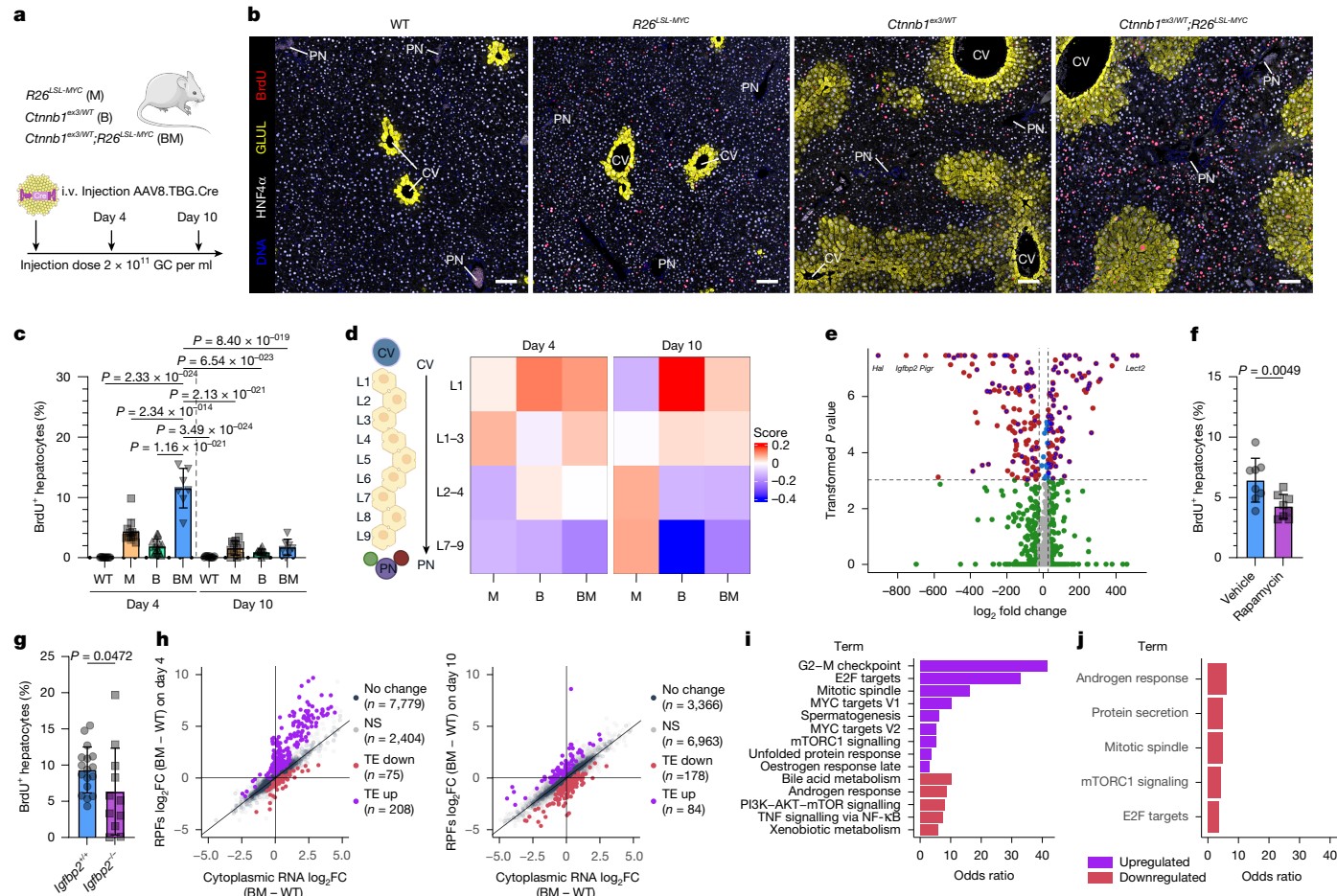

**Fig. 3 | Acute WNT and MYC activation drives hepatomegaly, via a transient proliferative response in zone 1 and zone 2 hepatocytes, before establishing a pan-lobular zone 3. a**, Schematic of acute, hepatocellular, WNT and MYC activation. The illustrations of the mouse and adenovirus were adapted from Medical Art Servier (https://servier.com) under a CC BY 4.0 licence. **b,c**, Confocal immunofluorescence staining for BrdU (red), GLUL (yellow) and HNF4α (white) in the liver 4 days post-induction (**b**). Nuclei were counterstained with DAPI (blue). Scale bars, 100 μm. Quantification of BrdU⁺ hepatocytes is also shown (**c**). For biological replicates, for day 4, $n = 10$ for WT, $n = 11$ for M, $n = 14$ for B and $n = 7$ for BM; for day 10, $n = 10$ for WT, $n = 11$ for M, $n = 11$ for B and $n = 7$ for BM. The bars are mean ± s.d. One-way ANOVA with Holm-Sidak's multiple comparisons test was used to determine significance. CV, central vein; PN, portal node. **d**, Whole-liver RNA sequencing and liver-zonation GSEA. Lobule layers (L1–9) are numbered from the central vein (L1) to the portal node (L9) according to Halpern et al.[28] ($n = 5$ on day 4 and $n = 3$ on day 10). The schematic of the lobule layers was created in BioRender. Raven, A. (2025) (https://BioRender.com/pv8v7yw). **e**, NanoString GeoMx digital spatial profiling of a *Ctnnb1*^ex3/WT^ day 4 liver, with the volcano plot comparing differentially expressed genes in GLUL⁺ and GLUL⁻ regions. Each dot denotes an individual gene. Green denotes log₂ fold change (FC) ≥ 2; blue indicates transformed $P ≥ 3$ (equates to adjusted

$P ≤ 0.001$) above dashed horizontal line; red shows over both thresholds; and grey denotes under both thresholds. The $P$ values were calculated in DESeq2 using a two-sided Wald test and the Benjamini–Hochberg method. The overlaid blue circles highlight WNT targets. $n = 4$ B mice. **f**, Quantification of BrdU⁺ hepatocytes in vehicle-treated and rapamycin-treated BM mice 4 days post-induction ($n = 8$ per group). A one tailed Student's *t*-test was used to determine significance. The bars are mean ± s.d. **g**, Quantification of BrdU⁺ hepatocytes in *Igfbp2*-deficient (*Igfbp2*⁻/⁻) BM mice 4 days post-induction. For biological replicates, $n = 17$ for *Igfbp2*⁺/⁺ and $n = 11$ for *Igfbp2*⁻/⁻. A one-tailed Student's *t*-test was used to determine significance. The bars are mean ± s.d. Data from panel **c** are included in panel **g**. **h–j**, Ribosome profiling analysis comparing the WT liver to the BM liver 4 and 10 days post-genetic recombination. For biological replicates, $n = 3$ per condition. The scatter plot presents translational efficiency (TE) changes (**h**); the colour scheme represents significantly altered mRNAs (adjused $P < 0.1$) translationally upregulated (purple dots; TE log₂FC > 0) and downregulated (red dots; TE log₂FC < 0). NS, not significant; RPF, ribosome-protected fragment. Gene Ontology over-representation analysis of TE upregulated (purple) or TE downregulated (red) mRNA using Hallmark processes on day 4 (**i**) and day 10 (**j**) livers is also shown. All terms with adjusted $P < 0.1$ are shown.

not form tumours in the liver but eventually developed intestinal polyps (Fig. 4b and Extended Data Fig. 8o–r). By contrast, the zone 1 and zone 2 *Gls2*^Cre-ER^;*Ctnnb1*^ex3/WT^;*R26*^LSL-MYC^ model did develop tumours (Fig. 4c and Extended Data Fig. 8s). Previous studies have linked RAS activity to portal node (zone 1 or 2) hepatocytes, suggesting that it may have a role in zonally specifying zone 1 of the hepatic lobule[27,28]. Introduction of oncogenic BRAF(V600E)—a mutant protein kinase that drives constitutive activation of the MAPK signalling cascade downstream of RAS—promoted tumorigenesis in the *Lgr5*⁺ hepatocyte population (Fig. 4d and Extended Data Fig. 8o–r,t–x). The resulting *Lgr5*⁺ hepatocyte-derived *Braf*^V600E^-driven tumours expressed the

zone 1 marker CDH1 and had reduced expression of the zone 3 and WNT marker GLUL (Fig. 4e). Furthermore, AAV8.TBG.Cre-mediated pan-hepatocellular *Braf*^V600E^ mutations suppressed zone 3 gene transcripts and upregulated a zone 1 gene expression programme (Fig. 4f). Combining *Braf*^V600E^ mutations with the ligand-dependent *Rnf43*^fl/fl^; *Znrf3*^fl/fl^ WNT pathway-activating mutations significantly increased organ growth (Extended Data Fig. 9a–d) and counteracted zonal differentiation, suppressing the upregulation of zone 3 features (Extended Data Fig. 9b,e–g). Increasing *Braf*^V600E^ in a gene-dosage-dependent way suppressed WNT-induced zone 3 differentiation and switched proliferation to predominantly occur in zone 3 GLUL⁺ hepatocytes

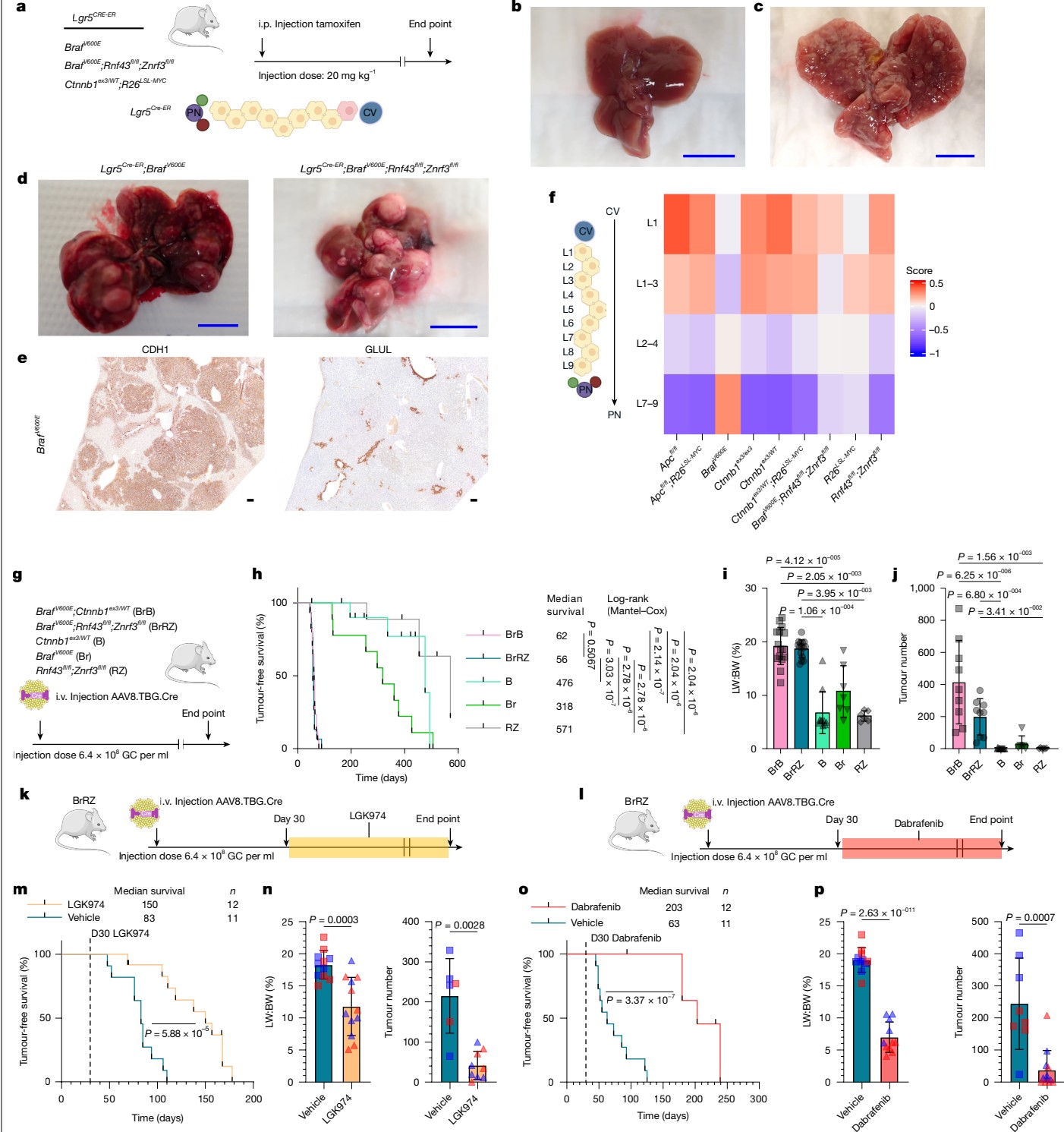

**Fig. 4** | See next page for caption.

(Extended Data Fig. 9h–n). Lowering the viral titre to restrict genetic recombination of mutant WNT pathway alleles and *Braf*[V600E], to a small number of hepatocytes, elicited pronounced tumorigenesis in the liver and reduced the median survival to 50–55 days (Fig. 4g–j). As small-molecule inhibitors are available for components of both WNT-ligand secretion (LGK974) and BRAF (dabrafenib) activation, we tested whether there was a dynamic relationship between BRAF and WNT signalling in the liver and established tumours. In the *Braf*[V600E/+];*Rnf43*[fl/fl];*Znrf3*[fl/fl] cancer model, tumorigenesis could be suppressed with either LGK974 or dabrafenib (Fig. 4k–p). Examination of dabrafenib-treated *Braf*[V600E/+];*Rnf43*[fl/fl];*Znrf3*[fl/fl] lesions revealed a reduction in proliferation, mTOR signalling and tumour growth along with an upregulation of GLUL and a decrease in CDH1 and SOX9 levels (Extended Data Fig. 9o–t). We conclude that aberrant growth in *Lgr5*[+] zone 3 hepatocytes requires factors that suppress zone 3 differentiation; combining a WNT pathway-activating mutation with MAPK signalling suppresses zone 3 differentiation, upregulates mTOR signalling and profoundly enhances tumorigenesis.

**Fig. 4 | Activated MAPK signalling antagonizes zone 3 differentiation enabling WNT-driven cancer and transformation of *Lgr5*⁺ zone 3 hepatocytes.** **a**, Central vein *Lgr5*-specific model of WNT and *Braf*$^{V600E}$ activation. i.p., intraperitoneal. **b–d**, Representative macroscopic images from *Lgr5*$^{CreER}$; *Ctnnb1*$^{ex3/WT}$;*R26*$^{LSL-MYC}$ (*n* = 10; **b**); *Gls2*$^{Cre-ER}$;*Ctnnb1*$^{ex3/WT}$;*R26*$^{LSL-MYC}$ (*n* = 5; **c**); *Lgr5*$^{CreER}$; *Braf*$^{V600E/+}$ (*n* = 7; left in **d**) and *Lgr5*$^{CreER}$;*Braf*$^{V600E/+}$;*Rnf43*$^{fl/fl}$;*Znrf3*$^{fl/fl}$ (*n* = 15; right in **d**) mouse livers. Scale bars, 1 cm. **e**, Representative images of CDH1 and GLUL IHC in *Lgr5*$^{CreER}$;*Braf*$^{V600E}$ liver tumours (*n* = 4). Scale bars, 100 μm. **f**, Day 10 whole-liver RNA sequencing and liver-zonation GSEA. For biological replicates, *n* = 9 for A, *n* = 8 for *Apc*$^{fl/fl}$;*R26*$^{LSL-MYC}$, *n* = 5 for Br, *n* = 3 for *Ctnnb1*$^{ex3/ex3}$, *n* = 3 for B, *n* = 5 for BM, *n* = 5 for BrRZ, *n* = 5 for M and *n* = 5 for RZ. The schematic of the lobule levels was created in BioRender. Raven, A. (2025) (https://BioRender. com/pv8v7yw). **g**, Schematic recombining WNT-mutant and *Braf*-mutant alleles in the hepatocyte epithelium. **h**, Tumour-free survival. For biological replicates: *n* = 17 for BrB, *n* = 15 for BrRZ, *n* = 12 for B, *n* = 9 for Br and *n* = 9 for RZ. See Methods for the censoring criteria (censors are denoted by the vertical tick marks). A log-rank (Mantel–Cox) test was used. **i**,**j**, LW:BW ratio (**i**; biological replicates: *n* = 15 for BrB, *n* = 15 for BrRZ, *n* = 10 for B, *n* = 7 for Br and *n* = 6 for RZ) and tumour scoring (**j**; biological replicates: *n* = 9 for BrB, *n* = 9 for BrRZ, *n* = 11

for B, *n* = 6 for Br and *n* = 6 for RZ) in mouse models of sporadic WNT-driven tumorigenesis. A one-way Kruskal–Wallis test and Dunn's multiple comparisons test were used to determine significance. The bars are mean ± s.d. B and RZ samples are plotted in Fig. 1b,c and Extended Data Fig. 1c. **k**,**l**, Schematic describing LGK974 (**k**) or dabrafenib (**l**) treatment from day 30 in the BrRZ tumour model. The illustrations of the mouse and adenovirus in panels **a**,**g**,**k**,**l** were adapted from Medical Art Servier (https://servier.com) under a CC BY 4.0 licence. **m–p**, Tumour-free survival (**m**,**o**; log-rank (Mantel–Cox) test; see Methods for the censoring criteria (censors are denoted by the vertical tick marks)), tumour scoring (right in **n**,**p**) and LW:BW ratio (left in **n**,**p**; two-tailed Student's *t*-test and two-tailed Mann–Whitney test) from BrRZ mice treated with either LGK974 (PORCN inhibitor; **m**,**n**) or dabrafenib (BRAF inhibitor; **o**,**p**). For biological replicates for the PORCN inhibitor experiments: for the LW:BW ratio, *n* = 11 for vehicle and *n* = 12 for LGK974; and for tumour number, *n* = 6 for vehicle and *n* = 9 for LGK974. For biological replicates for BRAF inhibitor experiments, for the LW:BW ratio, *n* = 11 for vehicle and *n* = 11 for dabrafenib; and for tumour number, *n* = 8 for vehicle and *n* = 12 for dabrafenib. The bars are mean ± s.d.

## *Apc* loss is less tumorigenic

Our analysis of mutational frequencies confirmed that *CTNNB1* exon 3 point mutations are the predominant WNT pathway mutation in HCC (Extended Data Fig. 10a). We sought to determine whether this could be explained by sporadic mutations arising from the mutational processes occurring in the liver. Comparing the predicted mutations, modelled from HCC mutational signatures, with the observed hotspot point mutations in *CTNNB1*, it appeared unlikely that random mutations alone could account for the prevalence of *CTNNB1* exon 3 point mutations in HCC (Extended Data Fig. 10b,c). To examine the over-representation of *CTNNB1* mutations in HCC, we replaced *Ctnnb1*$^{ex3/WT}$ with an *Apc*$^{fl/fl}$ allele in our WNT–MYC liver cancer model (Fig. 5a). Nuclear *Ctnnb1*⁺ *Apc*-deficient hepatocytes were detected at day 30, but there was a reduction in *Apc*$^{fl/fl}$;*R26*$^{LSL-MYC}$ hepatocytes by day 60 (Fig. 5b). In contrast to the *Ctnnb1*$^{ex3/WT}$;*R26*$^{LSL-MYC}$ model, *Apc*$^{fl/fl}$;*R26*$^{LSL-MYC}$-mutant hepatocytes formed smaller lesions at day 30, which disappeared by day 60 (Fig. 5c–e). Another key difference was the intensity of nuclear *Ctnnb1* staining, *Apc*$^{fl/fl}$;*R26*$^{LSL-MYC}$-mutant hepatocytes had uniform, intense nuclear *Ctnnb1* staining in contrast to the variable nuclear positivity observed in *Ctnnb1*$^{ex3/WT}$;*R26*$^{LSL-MYC}$ hepatocytes (Fig. 5f). High nuclear *Ctnnb1* staining was also observed in infrequent, undifferentiated, *Apc*$^{fl/fl}$;*R26*$^{LSL-MYC}$ end-stage tumours unlike the *Ctnnb1*$^{ex3/WT}$;*R26*$^{LSL-MYC}$ tumours that maintained lower levels of nuclear *Ctnnb1* (Fig. 5g). Furthermore, the reduced lesion formation in day 60 *Apc*$^{fl/fl}$;*R26*$^{LSL-MYC}$ mice prolonged survival and decreased tumorigenesis (Fig. 5h–l). These data confirm that HCC is more permissive to *Ctnnb1*$^{ex3}$ mutations than *Apc* loss. It has been proposed that the high degree of polyploidy in hepatocytes is selective for single-point mutations over tumour suppressor loss[29,30]. It appears that the heightened level of oncogenic WNT pathway activation, generated by *Apc* loss, is also less compatible with tumorigenesis in the liver, consistent with the relative absence of *APC* mutations in HCC.

To further investigate how the levels of WNT pathway activation and hepatic-lobule zonal specification affect liver tumorigenesis, we used a hypomorphic *Apc*$^{fl/fl}$ allele[31] that confers a higher baseline level of WNT pathway activation due to a 30% reduction in the levels of *Apc* than those in wild-type (WT) mice[31]. In the absence of Cre, the *Apc*$^{fl/fl}$-hypomorph liver had increased expression of the WNT and zone 3 marker GLUL and minimal expression of IGFBP2 (Extended Data Fig. 10d,e). In this background of elevated levels of WNT and an expanded zone 3, we induced genetic recombination throughout the hepatocyte epithelium of the *Apc*$^{fl/fl}$-hypomorph;*R26*$^{LSL-MYC}$ mice and compared organ growth to another non-hypomorphic conditional *Apc*$^{fl/fl}$ allele[9] (Extended Data Fig. 10f). The *Apc*$^{fl/fl}$-hypomorph;*R26*$^{LSL-MYC}$

liver had reduced proliferation at day 4, which resulted in a significantly diminished increase in organ size at day 8 (Extended Data Fig. 10g,h). In the absence of Cre-mediated recombination, the *Apc*$^{fl/fl}$-hypomorph did eventually develop liver tumours[32]. The tumours that arose in this genetic background had low WNT pathway activation, did not express nuclear *Ctnnb1* and GLUL, and expressed IGFBP2 at variable levels (Extended Data Fig. 10i,j). Together, these findings underscore that a WNT-high zone 3 hepatocyte state is not permissive to WNT-driven oncogenic growth and tumorigenesis in the liver.

## Discussion

The default role of WNT signalling in the liver is to promote zone 3 differentiation; this is recapitulated during oncogenesis when β-Catenin is mutated. Critically, differentiation to an *Lgr5*⁺GLUL⁺ zone 3 fate, which is not permissive to oncogenesis, needs to be reversed for WNT-mutant clones to progress to early tumours. Reversal of differentiation is dependent on reduction of WNT pathway activation, Yap/Taz and MAPK signalling, which then enables activation of mTOR and the proliferative translatome that support tumour outgrowth (Extended Data Fig. 11).

Together, this highlights that 'just right' WNT signalling occurs across cancers and is not just a feature of colorectal cancer[33,34]. Indeed, *Axin1* mutations—the other common WNT pathway alteration in HCC—also do not robustly activate WNT signalling but upregulate a subset of WNT-target genes permissive to tumorigenesis[35]. During HCC development, there may be different levels of 'WNT just right' activation that can affect cancer evolution depending on cancer stage[36], the origin of the cancer-initiating cell and additional co-mutational hits. Outside of cancer, the compartmentalized response to hyperactivation of the WNT pathway explains how WNT signalling could stimulate liver growth during regeneration. Injury-induced expression of ectopic WNT ligands[37], external to the homeostatic WNT signalling gradient, would stimulate growth instead of differentiation. Similar observations were made by Sun et al.[10], who showed a WNT-induced proliferative response outside of zone 3 when the pathway was stimulated with WNT ligands.

Identification of the pathways and factors that facilitate oncogenic WNT-driven growth broadens the range of targets that we could disrupt in what has been, historically, a difficult pathway to treat pharmacologically. This is best evidenced by the profound effect that rapamycin treatment had on the early mutant lesions. By understanding the molecular mechanisms in which isolated hepatocytes, harbouring oncogenic β-catenin mutations, progress to form multicellular clusters could lead to new chemopreventative approaches for early liver tumorigenesis.

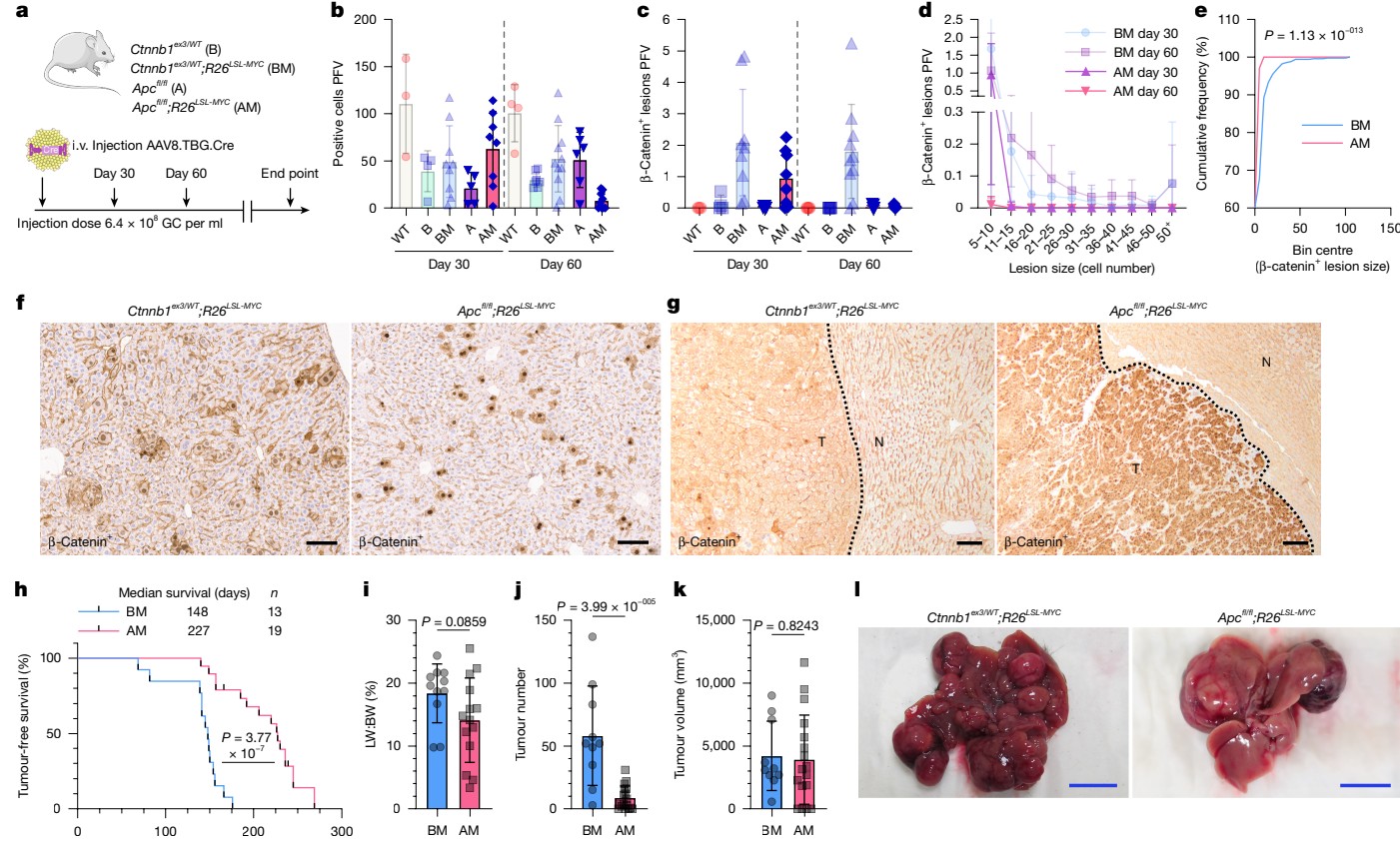

**Fig. 5 | AM hepatocytes show intense WNT pathway activation but are less tumorigenic. a**, Mouse model recombining alleles at a low clonal density in the hepatocyte epithelium. The illustrations of the mouse and adenovirus were adapted from Medical Art Server (https://servier.com) under a CC BY 4.0 licence. **b–e**, Quantification of recombined hepatocytes (**b**) and lesions (**c**) PFV in sections from day 30 ($n = 3$ for WT, $n = 4$ for B, $n = 9$ for BM, $n = 5$ for A and $n = 8$ for AM) and day 60 ($n = 4$ for WT, $n = 7$ for B, $n = 10$ for BM, $n = 6$ for A and $n = 9$ for AM) livers. Quantification of lesion size and distribution in AM livers at days 30 and 60 post-induction (**d**). Quantification of β-catenin$^+$ cluster cumulative frequency in AM livers, compared with BM counterparts, at day 30 post-induction (**e**). The bars are mean ± s.d. A two-sided Kolmogorov–Smirnov cumulative frequency test was used. **f**, Representative images of β-catenin IHC

of day 30 BM and AM livers ($n = 8$). **g**, Representative images of β-catenin IHC of end point BM and AM liver tumours ($n = 4$). The dashed line represents the tumour (T) border. N, normal tissue. **h–l**, End point survival (**h**), LW:BW ratio (**i**), tumour scoring (**j** (tumour number), **k** (tumour volume)) and liver macroscopic images (**l**) of BM and AM mice. For the LW:BW ratio, $n = 11$ BM and $n = 14$ AM mice; and for tumour scoring, $n = 10$ BM and $n = 17$ AM mice. The bars are mean ± s.d. A log-rank (Mantel–Cox) test and unpaired two-tailed Student's $t$-test were used (**b–d,i–k**). Biological replicates are indicated by $n$. Context data from Fig. 1e–h have been included in panels **b–e**; survival data from Fig. 1b have been included in panel **h**; and tumour scoring data from Fig. 1c and Extended Data Fig. 1c,d have been included in panels **i–k**. Scale bars, 100 μm (**f,g**) and 1 cm (**l**).

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

## Methods

### Mouse studies

All mouse experiments were performed according to UK Home Office regulations (project licence 70/8646 and PP3908577) following approval by the University of Glasgow Animal Welfare and Ethical Review Body. Mice were housed in a mixture of indvidually ventilated cages and conventional cages in a facility that operates as a specific-pathogen-free facility at constant temperature (19–23 °C) and humidity (55 ± 10%) under a 12-h light–dark cycle with ad libitum access to standard rodent chow and water; bedding material and tunnels were also included in all cages. All mice were genotyped from ear punch biopsies at weaning by a commercial vendor (Transnetyx). To study tumorigenesis in ageing cohorts, mice were monitored daily. Humane end points were determined based on loss of weight and body condition and the appearance of abdominal swelling. Liver cancer end point was generally identified by significant abdominal swelling or progressive weight loss; tumour volume measurements were not used to determine end point. The censoring criteria for ageing liver cancer cohorts: liver tumour-free mice were censored either due to health reasons not related to liver tumours (for example, epidermal wounds and age-associated lymphoma) or when they had passed 500 days post-induction. For genetically engineered mouse models, animals were assigned to groups according to their genotype. Treatment groups were randomly assigned; however, steps were taken during group assignment to avoid separating males in to singly housed cages. Selection of groups also aimed to maintain an equal sex balance. Investigators were blinded during treatment regimens and at sample collection for timepoint experiments. For ageing experiments, it was not possible for investigators to be blind to genotype as this factor needed to be known to maintain the welfare of the experimental cohort. Group sizes were determined on historical experiments where effect size and power were sufficient to detect statical differences.

The genotypes, transgenes and alleles used for this study were as follows: $Lgr5^{CreER}$ (ref. 39), $Gls2^{CreER}$ (ref.13), $Glul^{CreER}$ (ref. 13), $Cyp1a2^{CreER}$ (ref. 13), $Igfbp2^{CreER}$ (ref. 40), $R26^{CreER}$ (ref. 41), $Villin^{CreER}$ (ref. 42), $Ctnnb1^{ex3}$ (ref. 43), $Apc^{fl/fl}$ (ref. 9), $Apc^{fl/fl}$-hypomorphic[31], $Rnf43^{fl/fl};Znrf3^{fl/fl}$ (ref. 44), $R26^{LSL-Myc}$ (ref. 45), $R26^{LSL-tdTomato}$ (ref. 46), $Braf^{V600E}$ (refs. 47,48), $Pten^{fl/fl}$ (ref. 49), $Yap^{fl/fl}$ (obtained from the International Mouse Phenotyping Consortium Knockout Mouse Project Repository) and $Taz^{fl/fl}$ (ref. 50).

The $Igfbp2$-knockout mouse line was generated at the CRUK Scotland Institute by CRISPR gene-editing technology. A CRISPR project was designed to the $Igfpb2$ gene (ENSMUSG00000039323/GRCm39: CM000994.3). Two CRISPR guides were identified, which cut in the intronic sequence surrounding exon 2 (ENSMUSE00001244511) and were demonstrated to have high cutting efficiency in vitro: TAACTCTGTAGGTGGTAACG, TACTAGCCGCTTGGTGTTGA. Alt-R CRISPR–Cas9 crRNA, Alt-R CRISPR–Cas9 tracrRNA and Alt-R spCas9 nuclease were purchased from Integrated DNA Technologies. To generate the electroporation solution, the crRNA–tracrRNA duplex and spCas9 protein were combined in Optimem (Thermo Fisher Life Technologies). Approximately 7 h after in vitro fertilization, the one-cell stage embryos of 5–6-week-old C57BL/6J mice (Charles River) were introduced into electroporation solution and electroporated using a NEPA21 electroporator (Nepa Gene). The following day, two-cell embryos were transferred into the oviducts of pseudopregnant CD1 females (Charles River). Genotyping of subsequent pups was performed from ear samples by PCR using the primers F: AGACTGGGATGTGGAAGCAG and R: ACTTCTCAGTTCTCCAGGGC followed by Sanger sequencing.

Mice were maintained on a mixed C57BL/6 background. Genetic recombination was induced in both male and female mice, 2–4 months of age, with either an adeno-associated virus expressing $Cre$ under the control of the $TBG$ promoter (AAV8.TBG.Cre; AAV.TBG. PI.Cre.rBG was a gift from J. M. Wilson (Addgene plasmid #107787) to achieve temporal-specific and hepatocyte-specific Cre-mediated recombination of floxed alleles, or tamoxifen to activate $Rosa26^{CreER}$ (whole body), $Villin^{CreER}$ (intestinal specific) $Lgr5^{CreER}$, $Glul^{CreER}$, $Cyp1a2^{CreER}$, $Igfbp2^{CreER}$ and $Gls2^{CreER}$ (only male mice were used for the $Gls2^{CreER}$ experiments as female mice did not recombine as efficiently as male mice when treated with tamoxifen). AAV8.TBG.Cre was administered via intravenous (i.v.) tail-vein injections at a volume of 100 µl and a concentration of either $6.4 \times 10^8$ GC ml$^{-1}$ (ref. 51) or $2 \times 10^{11}$ GC ml$^{-1}$. Samples were excluded if there was evidence of a failed injection and impaired genetic recombination. At $6.4 \times 10^8$ GC ml$^{-1}$, the AAV8.TBG. Cre tropism is different between sexes; therefore, these experiments were performed on male mice. An exception to this was in the treatment of $Braf^{V600E/+};Rnf43^{fl/fl};Znrf3^{fl/fl}$ mice with dabrafenib and LGK974; here, equal numbers of male and female mice were used per experimental group. Tamoxifen (T5648, Sigma-Aldrich) was administered via intraperitoneal (i.p.) injection at 3-mg, 2-mg or 0.5-mg doses.

The PORCN-O-acyltransferase inhibitor LGK974 (205851, MedKoo Biosciences) was administered twice daily via oral gavage at 5 mg kg$^{-1}$ in a vehicle composed of 0.5% Tween-80 and 0.5% methylcellulose. Rapamycin (R-5000, LC Laboratories) was administered once daily via i.p. injection at 10 mg kg$^{-1}$ in a vehicle composed of 5% ethanol, 5% polyethylene glycol 400 and 5% Tween-80 in PBS. The BRAF inhibitor dabrafenib (A-1219, Active Biochem) was administered once daily via oral gavage at 30 mg kg$^{-1}$ in a vehicle composed of 0.5% hydroxypropyl methylcellulose and 0.1% Tween-80. To assay cell proliferation, the nucleotide analogue BrdU (RPN201, Amersham Biosciences) was administered 2 h before sampling via i.p. injection of 250 µl.

The AAV8.U6.shRNA-mIgfbp2, AAV8.CMV.Igfbp2 and AAV8.TBG. myrAkt were designed and constructed by a commercial vendor Vector-Builder. For the AAV8.U6.shRNA-mIgfbp2, the shRNA targeted the following sequence in the 3′ untranslated region (UTR) of the mouse $Igfbp2$ gene GAACCTCCCTTGCTTCTGTTA (catalogue number: P230413-1010twp; lot number: 230420AAVN02). Knockdown was confirmed using IGFBP2 IHC on livers treated with AAV8.U6.shRNA-mIgfbp2. The corresponding control AAV8.U6.shRNA-scramble was also purchased from VectorBuilder (catalogue number: AAV8LP(VB010000-0023jze)-C; lot number: 220329AAVJ07). Both AAV.U6.shRNAs were administered via i.v. tail-vein injections at $2 \times 10^{11}$ GC ml$^{-1}$. For the AAV8. CMV.Igfbp2 virus, the gene encoding mouse IGFBP2 was packaged in to an AAV8.CMV vector (catalogue number: scAAV8MP(VB241111-1375dkc); lot number: 241122AAVS06). IGFBP2 expression was confirmed using IGFBP2 IHC on livers treated with AAV8.CMV.Igfbp2. For the AAV8.TBG. myrAkt virus, mouse AKT1 with a Src myristoylation sequence at the beginning[52] was packaged in to an AAV8.TBG vector (catalogue number: AAV8L(VB250225-1147vdk); lot number: 250311AAVG01). For the AAV8. CMV.Igfbp2 and AAV8.TBG.myrAkt experiments, two control AAV8. CMV.eGFP and AAV8.TBG.eGFP viruses were used (catalogue number: scAAV8CP(VB010000-9304aud)-f and AAV8C(VB010000-9287ffw)-b; lot number: 241114AAVP24 and 241012AAVA01).

### Human tumour data

TCGA-LIHC data were downloaded from cBioPortal (https://cbioportal-datahub.s3.amazonaws.com/lihc_tcga_pan_can_atlas_2018.tar.gz). For Extended Data Fig. 10, three additional cohorts of HCC were used. The AMC[53] and INSERM[54] data were both downloaded from cBioPortal (https://cbioportal-datahub.s3.amazonaws.com/lihc_amc_prv.tar. gz and https://cbioportal-datahub.s3.amazonaws.com/hcc_inserm_ fr_2015.tar.gz). The LINC-JP data were downloaded from the ICGC data portal (https://dcc.icgc.org/releases/release_28/Projects/LINC-JP). Mismatch-repair deficient tumours were identified with SigMA[55] and removed from the analysis. To stratify HCC samples based on $CTNNB1$ mutation status, only activating mutations were considered. Activating mutations were defined as missense mutations at hotspots (protein position 32, 33, 34, 35, 36, 37, 41, 45, 335, 383 and 387) and in-frame indels at hotspots (protein position 23–71) according to Chang et al.[56]. Expression data were obtained from the batch-normalized RSEM (RNA-seq by

Expectation-Maximization) values from TCGA-LIHC after $\log_2$ transformation with a pseudo-count of 1.

## Human tumour data: GSVA analysis

For Extended Data Fig. 2g, GSVA scores were calculated with the R package GSVA (https://bioconductor.org/packages/release/bioc/html/GSVA.html) on $\log_2$-transformed RSEM values after removing lowly expressed genes (mean $\log_2(RSEM + 1) < 5$).

## Modelling *CTNNB1* mutations

To investigate whether the observed *CTNNB1* mutation hotspots are driven by the underlying mutational processes in HCC, we modelled the mutation frequencies using the trinucleotide mutational spectrum. Here only missense single-nucleotide variants were considered. We first calculated the 96-dimensional (six substitution subtypes: C > A, C > G, C > T, T > A, T > C and T > G, each flanked by one of the four types on the 5′ and 3′ sides) trinucleotide mutational spectrum of all HCC samples with at least 50 single-nucleotide variants and took the mean. This mean spectrum represents the exome-wide mutation probability $P_i^{exome}$ for each mutation type $i$ from 1 to 96 in HCC. We then performed a renormalization using the trinucleotide frequency in the whole exome and in the gene *CTNNB1* to obtain $P_i^{CTNNB1}$, which represents the local mutation probability at *CTNNB1* in HCC. Last, we calculated the predicted missense mutation frequency at each protein position as $P(\text{missense}) = \sum_i P(\text{missense}|i)P_i^{CTNNB1}$, where $P(\text{missense}|i)$ is the probability of a mutation of type $i$ generating a missense mutation at this particular protein position and can be calculated using the codon table. The predicted mutation frequencies were normalized so that their sum over all protein positions was the same as that of the observed frequencies.

## Apoptosis gene signature in HCC

TCGA-LIHC RNA sequencing (RNA-seq) count data and corresponding clinical information were downloaded using recount3 package (v1.6)[57]. TCGA mutational data were downloaded using GenomicDataCommons[58] R package (v1.12.0). Counts were normalized by applying the variance-stabilizing transformation function from DESeq2 (v1.36)[59]. Single-sample gene set enrichment analysis was performed using R package corto (v1.2)[60] with the Hallmark gene set[61] 'Apoptosis', downloaded using msigdbr (v7.5.1)[62]. Binned *MYC* expression was divided into two equal bins of the same number of patients. *CTNNB1* mutation status was converted to binary: mutated or not. Data were visualized using a combination of ggplot2[63] and cowplot[64] packages in R.

## Immunofluorescence and IHC

Livers were collected and fixed in 10% neutral buffered formalin either overnight at room temperature or overnight at 4 °C. Fixed tissue was processed for paraffin embedding, and tissue blocks were cut into 5-µm sections. IHC and immunofluorescence were performed on formalin-fixed paraffin-embedded sections according to standard staining protocols. The primary antibodies used for IHC and immunofluorescence were directed to the following antigens: β-catenin (1:50, 610154, BD Biosciences), glutamine synthetase (1:300, ab73593, Abcam (immunofluorescence); 1:800; HPA007316, Sigma-Aldrich (IHC)), BrdU (1:400, ab6326, Abcam (immunofluorescence); 1:250, 347580, BD Biosciences (IHC)), HNF4α (1:300, PP-H1415-00, Perseus Proteomics), E-cadherin (1:300, 610181, BD Biosciences), Ki67 (1:1,000, 12202, Cell Signaling Technology), IGFBP2 (1:1,000, PA5-81409, Invitrogen), RFP (1:10,000 600-401-379, Rockland), cleaved caspase 3 (1:500, 9661, Cell Signaling Technology), cleaved PARP (1:1,000, ab32064, Abcam), cyclin D1 (1:150, 55506, Cell Signaling Technology), peEF2 (1:100, 2331, Cell Signaling Technology), p4E-BP1 Thr37/46 (1:250, 2855, Cell Signaling Technology), pS6(Ser235/236) (1:75, 4858, Cell Signaling Technology), ribosomal protein pS6(Ser240/244) (1:1,000, 5364, Cell Signaling Technology), SOX9 (1:500, AB5535, Millipore) and MYC (1:800, ab32072, Abcam).

Representative brightfield images were acquired using an Olympus BX53 microscope and Olympus cellSens imaging software (v1.7.1). Immunofluorescent images were acquired using the Zeiss LSM 710 confocal microscope and the ZEN Black image acquisition software. Images were further processed using Fiji/ImageJ software (v1.53t)[65].

## Image analysis

Immunofluorescent images were acquired in up to four fluorescent channels at ×20 magnification on an Opera Phenix high-content imaging system (Perkin Elmer) and subsequently analysed using the Columbus software (v2.9.1.532; Perkin Elmer). Forty images were taken per liver section. DAPI-stained nuclei were identified based on pixel intensity using method 'B'. Nuclear size (more than 40 µm²) and morphology (roundness of more than 0.6) were then determined. Illumination correction and background normalization were performed using the sliding parabola module. Nuclei were then assigned as positive or negative based on the mean pixel intensity in the corresponding channel in either the nucleus (HNF4α, BrdU) or a 6-µm-thick region surrounding the nucleus (GLUL). The establishment and optimization of analysis algorithms was performed blind. IHC images were acquired using a SCN400F slide scanner (Leica Microsystems) at ×20 magnification. Scanned images were analysed using HALO image analysis software (v2.0.1145; Indica Labs). Liver sections were selected using the manual annotation tool and an image classifier to segment tissue and vasculature. Cell quantification and area quantification algorithms were then used to identify positive cells and staining. For the IGFBP2 IHC analysis, ten circular regions with a radius of 190 µm were used to segment areas around the central vein and portal node for each biological replicate. β-Catenin and RFP scoring was performed manually using images from the SCN400F slide scanner with the Leica Aperio ImageScope software (v12.4.3.5008). For β-catenin and RFP mosaic liver analysis, the following criteria were used: (1) nuclear β-catenin-positive and RFP-positive cells were scored in an average of 29 FOVs at ×10 magnification; (2) when scoring lesions, only clusters of five or more adjacent positive cells were quantified. *Igfbp2* scoring in tumours was performed manually using the Olympus BX53 microscope. Tumours containing cells positive for *Igfbp2* RNA scope probe were quantified as positive and tumours not containing cells positive for *Igfbp2* RNA scope probe were quantified as negative.

## RNA in situ hybridization

In situ hybridization for *Igfbp2* (405958) and *Notum* (428988) mRNA (all from Advanced Cell Diagnostics) was performed using RNAscope 2.5 LS Reagent Kit–BROWN (322100, Advanced Cell Diagnostics) on a BOND RX autostainer (Leica Biosystems) according to the manufacturer's instructions. Negative (dapβ, 312038) and positive (mm-*Ppib*, 313918) control probes (both from Advanced Cell Diagnostics) were included in each run to ensure staining specificity and RNA integrity.

## RNA isolation and sequencing

RNA was isolated from whole-liver tissue using the RNeasy Mini Kit (74104, Qiagen) according to the manufacturer's instructions. RNA concentrations were determined using a NanoDrop 200c spectrophotometer (Thermo Scientific), and quality was assessed using RNA ScreenTape on an Agilent 2200 Tapestation (Agilent Technologies). A total of 2 µg RNA was purified via poly(A) selection. RNA-seq libraries were generated using an Illumina TruSeq RNA sample prep kit and sequenced on an Illumina NextSeq 500 platform using the NextSeq 500/550 75-cycle High-Output kit (2 × 36 cycles, paired-end reads, single index), with the exception of samples from *Rosa26^CreER* models for which libraries were prepared using the Illumina TruSeq Stranded mRNA kit before sequencing on an Illumina Novaseq 6000 platform (2 × 150 cycles, paired-end, dual index). Raw sequence quality was assessed using the FastQC algorithm (v0.11.8). Sequences were subsequently trimmed to remove adaptor sequences and low-quality base calls,

defined by a Phred score of less than 20, using the Trim Galore tool (v0.6.4). The trimmed sequences were aligned to the mouse genome build GRCm38.98 using HISAT2 (v2.1.0), with raw counts per gene subsequently determined using FeatureCounts (v1.6.4). When comparing across groups, data were normalized as a block using quantile normalization. Differential expression analysis was performed using the R package DESeq2 (v1.22.2), which uses a negative binomial generalized linear model, with significance assessed using a Wald test and Benjamini–Hochberg multiple testing correction. Reactome pathway enrichment was performed using the R package ReactomePA (v1.36.0). Gene set analysis was performed using R packages GSA (v1.03.1) when comparing against WT, and GSVA (v1.40.1) for intergroup comparison of multiple genotypes. Liver zonation gene lists were derived from analysis in Halpern et al.[28]. Gene expression data across nine layers were filtered to include genes with non-zero expression, and then split into four zonation clusters using Euclidean distance and 'complete' hierarchical clustering.

### NanoString GeoMX digital spatial profiler

Mouse liver sections from four biological replicates per genotype were analysed using the digital spatial profiling procedure[66]. Formalin-fixed paraffin-embedded tissue sections were treated with 0.1 µg ml⁻¹ proteinase K (AM2546, Thermo Fisher Scientific) followed by heat-mediated epitope retrieval and incubated overnight with RNA oligo probes (mouse whole transcriptome atlas, Nanostring, GeoMx NGS RNA WTA Mm). Morphological markers were then detected using immunofluorescence, an anti-glutamine synthetase antibody (1:300, ab73593, Abcam) conjugated to 594 Alexa fluorescent protein (Fluorescent Protein Labeling Kits, A10239, Thermo Fisher), 647-anti-BrdU antibody (1:300, ab220075, Abcam) and the nuclear stain (SYTO13) were used. Slides were imaged at ×20 magnification using the GeoMx digital spatial profiler with the integrated software suite. Images were then used to select 2,500–300,000 µm² regions of interest (ROIs) on which the instrument focuses UV light (385 nm) to cleave the UV-sensitive probes with the subsequent release of the hybridized barcodes. For the day 4 *Ctnnb1*[ex3/WT] samples, 32 ROIs (8 per replicate) per each condition (GLUL-high, GLUL-low, GLUL-adj., PN-adj. and BrdU[pos]) were selected for UV-mediated cleavage and probe collection. For the day 60 *Ctnnb1*[ex3/WT];*R26*[LSL-MYC] samples, 34 ROIs for single clones, 49 ROIs for lesions and 15 ROIs for normal GLUL[pos] central veins were selected with further segmentation to GLUL[pos] regions for UV-mediated cleavage and probe collection. Libraries were prepared using GeoMx Seq Code primers (NanoString) and 1× PCR Master Mix (NanoString) and AMPure XP purification. Library quality was checked using an Agilent Bioanalyzer. The libraries were run on an Illumina NovaSeq sequencing system (GeneWiz/Azenta). The FASTQ files from sequenced samples were converted into Digital Count Conversion (DCC) files using the GeoMx NGS pipeline on NanoString's DND platform. The DCC files were uploaded onto the GeoMx digital spatial profiler analysis suite (NanoString), where they underwent quality control, filtering and Q3 normalization. Normalized GeoMx data were analysed using R base functions and packages described above.

### Ribosome profiling

Livers were collected and harvested essentially as previously described[67]. Livers were dissected on surfaces cleaned and treated with RNase Zap to reduce RNase exposure. Livers were rapidly dissected and snap frozen as 5 × 5 × 5 mm fragments in liquid nitrogen. Livers were ground to powder using a mortar and pestle with the CryoGrinder system (OPS Diagnostics) to ensure the samples were kept frozen. Roughly 200 mg tissue aliquots were stored in 1.5-ml tubes at −80 °C until required.

Ground liver tissue was poured directly from a 1.5-ml tube on dry ice into 900 µl ice cold lysis buffer (15 mM Tris-HCl (pH 7.5), 15 mM MgCl₂, 150 mM NaCl, 1% Triton X-100, 0.05% Tween 20, 2% *n*-dodecyl β-D-maltoside (89903, Thermo Fisher), 0.5 mM DDT, 100 µg ml⁻¹

cycloheximide, 1× cOmplete, Mini, EDTA-free protease inhibitor cocktail (11836170001, Merck), 1× protease inhibitor cocktail (P9599-5ml, Sigma), 5 mM sodium fluoride, 500 U ml⁻¹ RiboLock RNase inhibitor (EO0381, Thermo Fisher Scientific) and 25 U ml⁻¹ Turbo DNase (AM2239, Thermo Fisher Scientific) on ice and immediately mixed together and placed on ice for 10 min, while being inverted every 2 min to maximize the dispersal and exposure of ground tissue to lysis buffer. Lysates were centrifuged at 4 °C for 5 min at 16,000*g* and 800 µl supernatant pipetted into a fresh 1.5-ml tube on ice. For cytoplasmic RNA sample, RNA was extracted from 40 µl lysate with 1 ml TRIzol, as per the manufacturers' instructions. Lysate (600 µl) was digested with 1 µl Ambion RNase I cloned, 100 U µl⁻¹ (AM2295, Thermo Fisher Scientific), 1 µl MNase (1:20 diln; O247S, New England Biolabs) with 3.6 µl 0.25 M CaCl₂, at 22 °C for 15 min at 650 rpm in a thermomixer. The digestion was stopped with 5 µl SUPERaseIn RNase inhibitor (20 U µl⁻¹; AM2696, Thermo Fisher Scientific) and 26 µl 0.5 M EGTA (to stop RNase 1 and MNase, respectively). The samples were loaded onto a 10–50% sucrose gradient, containing 15 mM Tris-HCl (pH 7.5), 15 mM MgCl₂, 150 mM NaCl and 100 µg ml⁻¹ cycloheximide, prepared with a BioComp gradient station and cooled to 4 °C for at least 1 h. Samples were centrifuged in a Beckman XPN-90 Ultracentrifuge with an SW40ti rotor at 38,000 rpm for 2 h at 4 °C. One-ml fractions were collected with a BioComp gradient station and Gilson FC 203B fraction collector. Fractions pertaining to the 80S peak were extracted with acid phenol chloroform, followed by two chloroform washes, and RNA was precipitated with 2 µl glycogen (10901393001, Roche), 300 mM NaOac (pH 5.2) and an equal volume of isopropanol overnight at −20 °C.

RNA was pelleted by centrifugation at 12,000*g* for 45 min at 4 °C. The supernatant was removed with a pipette, and the pellet was washed twice with 70% ethanol and dissolved in 10 µl RNase-free water. RNA was diluted with 10 µl 2× TBE-urea sample buffer (LC6876, Thermo Fisher Scientific), heated at 80 °C for 90 s, placed immediately on ice, and then loaded onto a pre-run 15% TBE-urea gel (EC68852BOX, Thermo Fisher Scientific) and ran at 200 V for 1 h alongside custom 28-nt (AGCGUGUAC UCCGAAGAGGAUCCAACGU) and 34-nt (GCAUUAACGCGAACUCGGCC UACAAUAGUGACGU) RNA markers. The gel was stained with 1× SYBR gold (S11494, Thermo Fisher Scientific) and imaged on a Typhoon FLA 7000. An image was printed to size to allow bands, inclusive of 28-nt and exclusive of 34-nt markers, to be cut from the gel, placed into a 1.5-ml RNA low-binding microcentrifuge tube, and crushed with a RNase-free disposable pestle. RNA was eluted from the crushed gel pieces in 500 µl extraction buffer (300 mM NaOAc pH 5.2, 1 mM EDTA and 0.25% SDS) overnight at 16 °C at 600 rpm in a thermomixer. Gel pieces were filtered out with a Costar Spin-X centrifuge tube filter (0.45 µm; 8163, Scientific Laboratory Supplies) and RNA was precipitated with 2 µl glycogen and 500 µl isopropanol overnight at −20 °C.

Precipitated RNA was again pelleted, washed and dissolved in 13.5 µl RNase-free water, as above, and underwent rRNA depletion as follows. To this, 13.5 µl RNA was added, 5 µl hybridization buffer (10 mM Tris-HCl pH 7.5, 1 mM EDTA and 2 M NaCl), 1 µl RNasin plus ribonuclease inhibitor (N2615, Promega) and 0.5 µl biotinylated DNA oligo pool (Supplementary Table 1; 100 µM total DNA with rRNA_depl_1 and rRNA_depl_2 at a 3:1 molar ratio compared with all other oligos) that had been denatured at 95 °C for 3 min and then placed immediately on ice. This mix was incubated at 68 °C for 10 min at 1,250 rpm in a thermomixer and then allowed to cool slowly to room temperature by turning off the thermomixer. rRNA was depleted with 160 µl Dynabeads MyOne Streptavidin C1 (65001, Thermo Scientific) as per the manufacturer's instructions and depleted RNA was precipitated with 2 µl glycogen and ethanol.

Precipitated RNA was again pelleted and washed as above and then dissolved in 43 µl RNase-free water and heated at 80 °C for 90 s and immediately placed on ice. RNA then underwent 5′ phosphorylation and 3′ dephosphorylation with 1 µl T4 PNK (M0201S, NEB), 5 µl 10× PNK buffer and 1 µl SUPERaseIn RNase inhibitor (20 U µl⁻¹) at 37 °C for 35 min,

with 5 µl 10 mM ATP added for the final 20 min. RNA was extracted with acid phenol–chloroform and precipitated with isopropanol as above.

Purified RNA was quantified on a qubit with the high-sensitivity RNA kit (Q32852) and 5 ng was input into the Bioo Scientific Nextflex small RNA v3 kit (NOVA-5132-06), using the alternative step F bead clean-up, 14 PCR cycles and gel-extraction option.

Cytoplasmic RNA samples were run on an Agilent TapeStation to check RNA integrity. RNA concentration and sample purity were measured on a NanoDrop spectrophotometer. rRNA was depleted from 1 µg cytoplasmic RNA with the RiboCop v2, and sequencing libraries were prepared from the rRNA-depleted RNA with the Corall Total RNA-Seq Library Prep Kit v1 (095.96, Lexogen) using 13 PCR cycles.

Final cytoplasmic RNA and RPF libraries were quantified on an Agilent TapeStation, with a high-sensitivity D1000 ScreenTape (5067-5584, Agilent), and sequenced single-end on an Illumina NextSeq500 instrument with a 75 cycles high-output kit.

## Polysome profiles

For undigested polysome profiles, liver tissue was collected, harvested and lysed exactly as for ribosome profiling, except that 600 µl of lysate was loaded straight onto a 10–50% sucrose gradient and centrifuged in a Beckman XPN-90 Ultracentrifuge with a SW40ti rotor at 38,000 rpm for 2 h at 4 °C and run on a BioComp gradient station.

## Bioinformatic processing of ribosome profiles

Ribosome footprinting analysis was performed using the Bushell laboratory's RiboSeq GitHub pipeline (https://github.com/Bushell-lab/Ribo-seq). All versions of scripts used in this publication can be found on Zenodo[68] (https://zenodo.org/records/17224880). All R scripts were carried out using R (v4.3.1). Basic explanations of what these scripts do is written below.

Raw fastq files were quality control checked with fastQC and have been uploaded to the Gene Expression Omnibus database with the accession number: GSE275864. Cutadapt (v1.18)[69] was used to remove adaptors, trim 3′ bases with Phred scores < 20 and discard reads fewer than 30 and more than 50 bases after trimming. UMI-tools (v1.0.1)[70] was used to extract unique molecular indexes (UMIs; 4 nt of random sequence at the start and end of every RPF read) from the reads and appended to the read name. Reads were aligned with BBmap (v38.18), first to remove reads that aligned to either rRNA or tRNA sequences. Non-rRNA or tRNA reads were then aligned to a filtered protein-coding FASTA (see below), containing the most abundant transcript per gene (calculated from cytoplasmic RNA samples). The resulting BAM files were sorted and indexed with samtools (v1.9) and deduplicated using UMI-tools with the directional method. The number of reads with the 5′ end at each position of every transcript was then counted. Protein-coding-aligned read lengths peaked at 33–34 nt and lengths 30–38 were used in this analysis, which showed strong coding sequence (CDS) enrichment and periodicity. The offset for each read length was determined to be 12 nt for read lengths 30–31 nt, 13 nt for 32–35 nt and 14 nt for 36–38 nt. These offsets were applied to collapse all reads into a single frame. Total counts across the entire CDS, excluding the first 20 and last 10 codons, were then summed together and used as input to DESeq2 (ref. 59).

The paired cytoplasmic RNA samples were processed as for the RPFs above but with the following exceptions: the UMIs were 12 nt at the start of each read only. No maximum read length was set when trimming reads with Cutadapt. Reads were aligned to the filtered protein-coding transcriptome with Bowtie2 (v2.3.5.1)[71], using the parameters recommended for use with RSEM: --sensitive --dpad 0 --gbar 99999999 --mp 1,1 --np 1 --score-min L,0,-0.1. Both gene-level and isoform-level quantification was performed using RSEM (v1.3.3)[72]. The isoform quantification was used to determine the most abundant transcript per gene (see below), but differential expression was measured at the gene level with DESeq2 (ref. 59).

The gencode.vM27.pc_transcripts.fa file was downloaded from https://www.gencodegenes.org/mouse/release_M27.html and filtered to include only transcripts that had been manually annotated by HAVANA and that have a 5′ UTR, a 3′ UTR, a CDS equally divisible by three, an nUG start codon and a stop codon. All PAR_Y transcripts were also removed. The cytoplasmic RNA-seq data were then aligned to this filtered FASTA and the most abundant transcript per gene was determined, based on the mean transcripts per million (TPMs) across all samples from the RSEM output. The RPF reads were then aligned to a FASTA containing only the most abundant transcript for each gene.

DESeq2 (ref. 59) was used to test for differential expression. This was carried out separately on either the RPF or cytoplasmic RNA samples to calculate $log_2$FCs and plot principal component analyses and also to test whether the change in RPFs could be explained by the change in cytoplasmic RNA, as previously described[73]. TE down groups were mRNAs with adjusted $P < 0.1$ and $log_2$FC $< 0$, TE up groups were mRNAs with adjusted $P < 0.1$ and $log_2$FC $> 0$, no change groups were mRNAs with adjusted $P > 0.9$ and not significant (NS) mRNAs were those with $0.1 \geq$ adjusted $P \leq 0.9$.

Gene Ontology over-representation analysis was performed with the R package EnrichR[74] using the genes in the groups identified as being upregulated or downregulated at the translational level (TE up or TE down).

## Statistical analyses

A priori based on historical datasets used to ensure the smallest sample size that could give a significant difference was chosen in accordance with the 3Rs. Statistical analysis was performed with GraphPad Prism (v7.0.4) for Windows (GraphPad Software; www.graphpad.com). Normal distribution of data was determined using the D'Agostino and Pearson omnibus normality test. For non-parametric data, or where the sample size ($n$) was too small to determine normal distribution, data significance was analysed using a one-tailed or two-tailed Mann–Whitney test; for parametric data, data significance was analysed using a one-tailed or two-tailed unpaired Student's $t$-test. Paired data were analysed using a pairwise Wilcoxon test. For comparison between more than two groups, a one-way ANOVA, Kruskal–Wallis or two-way ANOVA was used with either Tukey's, Dunn's or Sidak's multiple comparisons post-hoc tests. To analyse differences in the distribution of data, a Kolmogorov–Smirnov cumulative frequency test was used. Statistical comparisons of survival data were performed using the 'log-rank' (Mantel–Cox) test. $P \leq 0.05$ was considered significant. For individual value plots, data are displayed as mean ± s.d., unless stated otherwise. Statistical tests and corresponding $P$ values are indicated in the figure legends and figures, respectively. For all histological analysis, the samples were randomized and the researchers were blinded to the genotype or treatment.

## Reporting summary

Further information on research design is available in the Nature Portfolio Reporting Summary linked to this article.

## Data availability

The RNA-seq and spatial transcriptomic data generated in this study are publicly available through the Gene Expression Omnibus with the following accession codes; GSE230644, GSE230110, GSE230137, GSE230144 and GSE275864. All other data are available from the corresponding author on reasonable request. The TCGA-LIHC data were downloaded from cBioPortal (https://cbioportal-datahub.s3.amazonaws.com/lihc_tcga_pan_can_atlas_2018.tar.gz). The AMC[53] and INSERM[54] data were both downloaded from cBioPortal (https://cbioportal-datahub.s3.amazonaws.com/lihc_amc_prv.tar.gz and https://cbioportal-datahub.s3.amazonaws.com/hcc_inserm_fr_2015.tar.gz). The LINC-JP data were downloaded from the ICGC data portal

(https://dcc.icgc.org/releases/release_28/Projects/LINC-JP). Ribosome footprinting analysis was performed using the Bushell laboratory's RiboSeq GitHub pipeline (https://github.com/Bushell-lab/Ribo-seq). Source data are provided with this paper.

## Code availability

All versions of scripts used in this publication can be found on Zenodo[68] (https://zenodo.org/records/17224880).

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

**Acknowledgements** We thank the Core Services and Advanced Technologies at the Cancer Research UK Scotland Institute (funded by Cancer Research UK core funding A17196 and A31287)—particularly the Biological Services Unit, the Transgenic Technology service, Bioinformatics service, the Histology Service and the Molecular Technologies service—for technical support; C. Winchester at the Cancer Research UK Scotland Institute for advising on manuscript preparation; and members of the Sansom laboratory and SPECIFICANCER Cancer Research UK Grand Challenge consortium for discussions of the data and manuscript. O.J.S. and his laboratory members and S.M. were supported by Cancer Research UK (A29055, A28223, A21139, DRCQQR-May21\100002 and CTRQQR-2021\100006). A.R. and K.G. were supported by the SPECIFICANCER Cancer Research UK Grand Challenge Award (A29055 to O.J.S.). C.A.F. was supported by the Rosetta Cancer Research UK Grand Challenge Award (A25045 to O.J.S.). R.A.R. and N.S. were supported by Cancer Research UK core funding to the Scotland Institute (A17196 and A31287) and the Colorectal Cancer and WNT signalling group (A21139 and DRCQQR-May21\100002 to O.J.S.). N.V. was supported by the Wellcome Trust (201487 to O.J.S.). M.L.M. was supported by a CRUK Accelerator Award (A28223 to O.J.S.). H.J., D.C.G. and P.J.P. were supported by the SPECIFICANCER Cancer Research UK Grand Challenge and an award from the Mark Foundation for Cancer Research to the SPECIFICANCER team. J.I. was supported by a Sigrid Juselius Foundation Senior Fellowship. M.M. and T.G.B. were funded by the Wellcome Trust and the CRUK Scotland Centre (WT107492Z and CTRQQR-2021\100006) and CRUK Accelerator (A26813). The human HCC results shown here are in whole or in part based on data generated by the TCGA Research Network (https://www.cancer.gov/tcga). H.Z. is supported by NIH R01 grants (CA251928 and AA028791), the Simmons Comprehensive Cancer Center, and the Emerging Leader Award from the Mark Foundation for Cancer Research (no. 21-003-ELA). S.H.W. is funded by a Chief Scientist Office Early Postdoctoral Fellowship (EPD/22/12) and a Research Incentive Grant (RIG012508) from The Carnegie Trust for the Universities of Scotland. L.B. is funded by Cancer Research UK (C52499/A27948, CTRQQR-2021\100006 and PRCBTP-May24/100001) and UKRI MRC (MR/Z506199/1). The mouse and viral diagrams were obtained from Servier Medical Art (https://smart.servier.com/smart_image/mouse-3/ and https://smart.servier.com/smart_image/adenovirus-molecule/) and are licensed under a CC BY 4.0 licence. Other diagrams were generated using BioRender (http://biorender.com).

**Author contributions** A.R., T.G.B. and O.J.S. designed and interpreted the results of all experiments. A.R., K.G., R.A.R., C.A.F., N.V., M.L.M., A.H., M.M., S.M. and S.H.W. performed the experiments and analysed the results. H.J., D.C.G. and H.H. analysed publicly available human cancer datasets. A.R., H.L. and N.B.J. designed and performed the spatial transcriptomic experiments. J.A.W., J.M. and M.B. designed, performed and analysed the ribosome sequencing experiments. K.G. processed and analysed the RNA-seq and spatial transcriptomic data. B.C. and C.N. performed ISH and IHC. S.B., E.A. and D.S. produced the *Igfbp2*-knockout mice. R.G., L.B., N.B., H.C., H.Z., J.I., C.J.M., M.B., N.B.J., P.J.P. and T.G.B. provided advice and reagents. A.R., N.S. and O.J.S. wrote the paper.

**Competing interests** O.J.S. receives funding from Novartis, Cancer Research Technology (Cancer Research Horizons), Boehringer Ingelheim and AstraZeneca for other unrelated projects. H.Z. is a cofounder of Quotient Therapeutics and Jumble Therapeutics, advises for Newlimit, Alnylam Pharmaceuticals and receives funding from Chroma Medicines for other unrelated projects. H.C. is the head of PharmaResearch and Early Development at Roche; full disclosure is available at https://www.uu.nl/staff/JCClevers. M.B. has consulted for BIoNTech. All other authors declare no competing interests.

**Additional information**
**Correspondence and requests for materials** should be addressed to Owen J. Sansom.

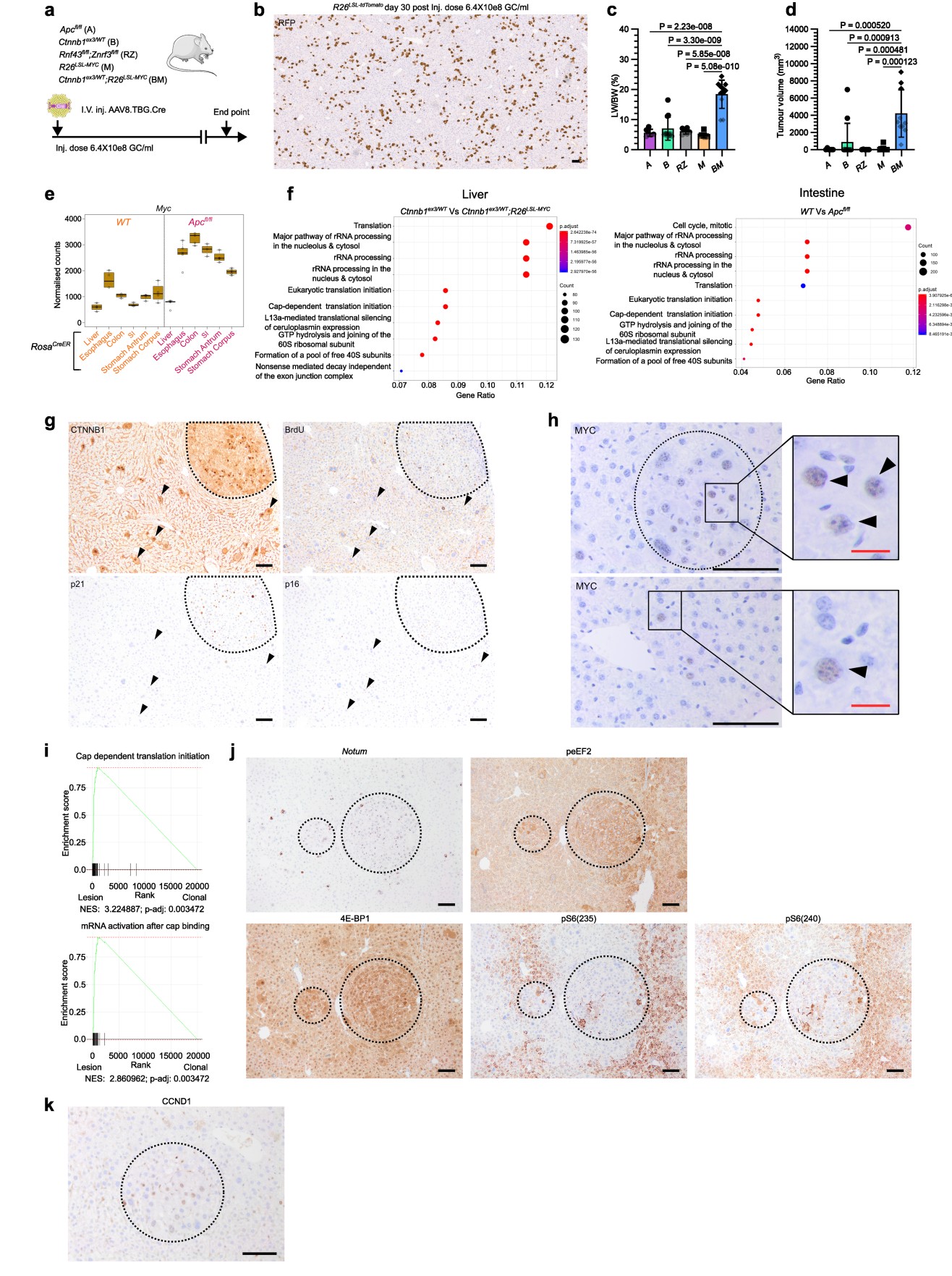

**Extended Data Fig. 1** | See next page for caption.

**Extended Data Fig. 1 | Elevated MYC expression is required for Wnt-driven cancer in the liver and single mutant clones do not express senescence markers or the translational machinery required for tumorigenesis.**
**a**, Mouse model recombining alleles at a low clonal density in the hepatocyte epithelium. The illustrations of the mouse and adenovirus were adapted from Medical Art Servier (https://servier.com) under a CC BY 4.0 licence.
**b**, Representative image of RFP IHC; $R26^{LSL\text{-}tdTomato}$ liver 30 days post administration of AAV8.TBG.Cre (GC/ml = genome copy per ml), n = 3. **c**, Liver-to-body weight ratio (LW/BW). Biological replicates: $Apc^{fl/fl}$ (A) n = 6, $Ctnnb1^{ex3/WT}$ (B) n = 12, $Rnf43^{fl/fl}$; $Znrf3^{fl/fl}$ (RZ) n = 6, $R26^{LSL\text{-}MYC}$ (M) n = 8, $Ctnnb1^{ex3/WT}$; $R26^{LSL\text{-}MYC}$ (BM) n = 11. Bars are mean ± s.d. One-way ANOVA with Tukey's multiple comparisons test. **d**, Tumour scoring. Biological replicates: A n = 6, B n = 12, RZ n = 6, M n = 9, BM n = 10. Bars are mean ± s.d. One-way ANOVA with Tukey's multiple comparisons test. **e**, Whole-tissue normalised RNA-Seq read counts for $Myc$ in indicated tissues from wild-type and $Apc^{fl/fl}$ mice. Box plots: centre line = median, upper (25th percentile) and lower quartiles (75th percentile) (box limits), and 1.5× interquartile range (whiskers) Biological replicates, moving left to right on the x-axis: n = 4,4,4,3,4,4,5,5,5,3,4,5. **f**, Top reactome pathways enriched in genes differentially upregulated in: AAV8.TBG.Cre-treated livers 10 days post induction ($2 \times 10^{11}$ GC/ml) BM versus B; and $Villin^{CreER} Apc^{fl/fl}$ versus $Villin^{CreER}$ (WT) intestines at day 4 post induction with tamoxifen. Dots are coloured according to their adjusted P-values, with the size of the dots representing the number of differentially expressed. Statistical significance was determined with the enrichPathway() function in R, using a two-sided hypergeometric model to determine the probability of geneset overlap occurring by chance. Multiple testing correction was applied using the Benjamini-Hochberg method. Biological replicates: liver, n = 3; intestinal, n = 16 $WT$ and n = 15 $Apc^{fl/fl}$. **g-k**, $Ctnnb1^{ex/WT}$; $R26^{LSL\text{-}MYC}$ liver, 60 days post AAV8.TBG.Cre ($6.4 \times 10^8$ GC/ml). **g**, Representative images of CTNNB1, BrdU, p21 and p16 IHC on serial sections, n = 3. Arrowheads represent single mutant clones and dashed black lines indicate a lesion. **h**, Representative images of MYC IHC, black arrowheads highlight MYC^pos nuclei **i**, Spatial transcriptomics GSEA. Normalised Enrichment Score (NES) was calculated by normalising to the mean enrichment of random samples, and two-sided permutation testing with a Benjamini-Hochberg test was applied, p-adj = p adjusted value. **j-k**, In situ hybridisation for $Notum$ and IHC for peEF2, p4E-BP1 (Thr37/46), pS6(Ser235/236), pS6(Ser240/244), and CCND1. Dashed outline highlights lesion. Representative images of n = 4 per group. All black scale bars = 100 μm. All red scale bars = 20 μm.

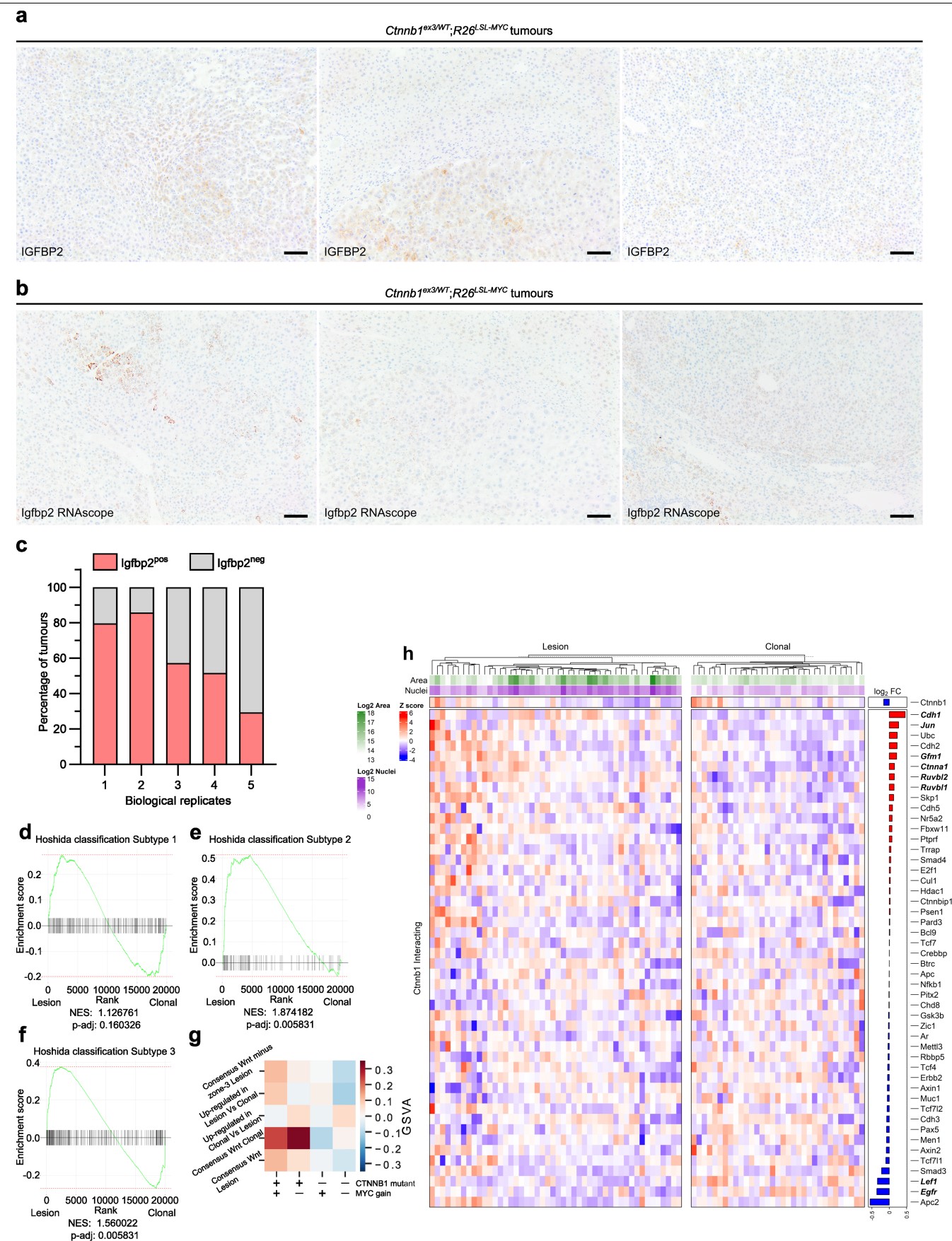

**Extended Data Fig. 2 |** See next page for caption.

**Extended Data Fig. 2 | *Ctnnb1^ex3/WT^;R26^LSL-MYC^* Tumours express *Igfbp2*.**
End point *Ctnnb1^ex3/WT^;R26^LSL-MYC^* liver tumours. **a** and **b**, Images of IGFBP2 immunohistochemistry and images of in situ hybridisation for *Igfbp2*, n = 3. Scale bars, 100 μm. **c**, Tumor scoring for *Igfbp2* positive and negative tumours, Biological replicates n = 5. **d-f**, GSEA of Hoshida[22] subtypes, comparing single clones to lesions in day-60 spatial transcriptomic data. For all GSEA plots the NES was calculated by normalising to the mean enrichment of random samples, and two-sided permutation testing was applied to determine statistical significance. Benjamini-Hochberg multiple test correction was applied to the resulting, p-adj = p adjusted value. **g**, Gene set variation analysis (GSVA) on TCGA-LIHC samples stratified by *CTNNB1* activating mutation status and *MYC* copy gain status. Samples with Genomic identification of significant targets in cancer (GISTIC) copy number 1 (gain) or 2 (high level amplification) are considered to have copy gains. Mean GSVA scores for samples within each group are shown. **h**, Heatmap showing differential gene expression of factors that regulate CTNNB1 activity at a protein and transcriptional level from the spatial transcriptomics assay. Selected regions of interest: n = 34 clonal; n = 49 lesions. Significantly changed genes highlighted in bold text. Biological replicates: n = 4 mouse livers. Black scale bars, 100 μm. Red scale bars, 20 μm. NES = normalised enrichment score, p-adj = p adjusted value.

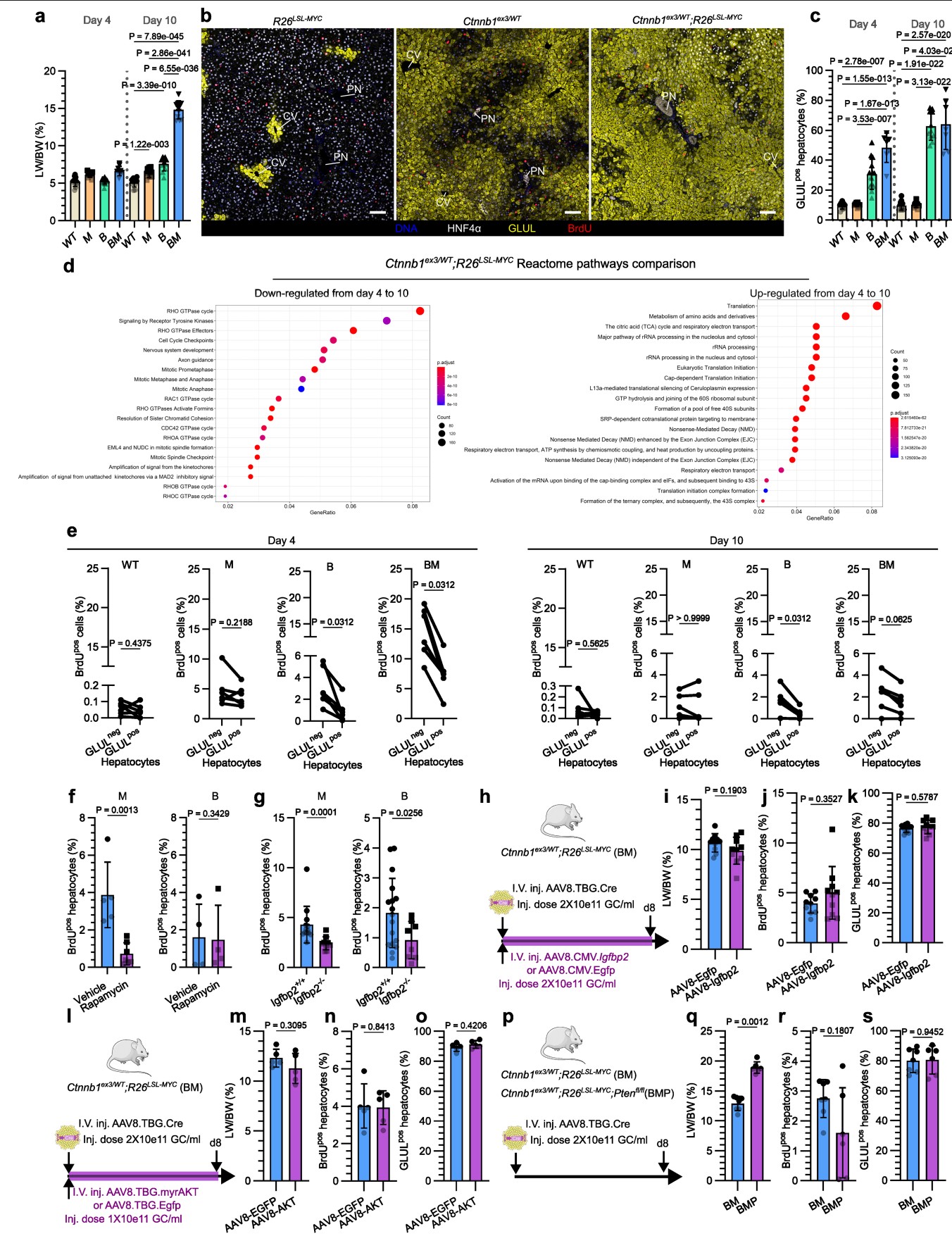

**Extended Data Fig. 3 |** See next page for caption.

**Extended Data Fig. 3 | Transient proliferation of zone-1 and -2 GLUL[neg] hepatocytes following β-catenin and MYC activation. a-e**, AAV8.TBG.Cre ($2 \times 10^{11}$ GC/ml) treated livers sampled 4 and 10 days post administration **a**, LW/BW, Biological replicates: Day 4−WT n = 10, $R26^{LSL-MYC}$ (M) n = 11, $Ctnnb1^{ex3/WT}$ (B) n = 14, $Ctnnb1^{ex3/WT};R26^{LSL-MYC}$ (BM) n = 9; Day 10−WT n = 10, M n = 11, B n = 10, BM n = 11. Bars are mean ± s.d. One-way ANOVA with Holm-Sidak's multiple comparisons test. **b**, Confocal IF staining for BrdU (red), GLUL (yellow), and HNF4α (white) 10 days post induction. Nuclei counterstained with DAPI (blue), n = 6. Scale bars, 100μm. **c**, Quantification of GLUL[pos] hepatocytes. Biological replicates: Day 4−WT n = 10, M n = 11, B n = 14, BM n = 7; Day 10−WT n = 10, M n = 10, B n = 11, BM n = 7. Bars are mean ± s.d. One-way ANOVA with Holm-Sidak's multiple comparisons test. **d**, Whole liver RNA-Seq, top reactome pathways enriched in downregulated and upregulated genes between day-4 and -10 BM livers. Dots are coloured according to their adjusted P-values. The enrichPathway() function in R, using a two-sided hypergeometric model determined the probability of geneset overlap occurring by chance. Multiple testing correction was applied using the Benjamini-Hochberg method. Biological replicates: Day 4, n = 5; Day 10, n = 3. **e**, Quantification of BrdU[pos] hepatocytes, stratified into GLUL-positive and -negative subsets. Two-sided Wilcoxon matched-pairs test. Biological replicates: n = 6 per condition.

**f**, Quantification of BrdU[pos] hepatocytes in Day-4 vehicle- and rapamycin-treated mice. Biological replicates: M vehicle n = 5, rapamycin n = 7; B vehicle n = 4, rapamycin n = 4. Bars are mean ± s.d. One tailed Mann−Whitney test. **g**, Quantification of BrdU[pos] hepatocytes in *Igfbp2* knockout (*Igfbp2*[-/-]) mice. Biological replicates: M *Igfbp2*[+/+] n = 13, *Igfbp2*[-/-] n = 9; B *Igfbp2*[+/+] n = 17, *Igfbp2*[-/-] n = 8. Bars are mean ± s.d. One tailed t-test and Mann−Whitney test. Data from Fig. 1c included in *Igfbp2*[+/+] groups. **h**, Acute BM activation combined with viral-vector delivery of an *Igfbp2* expressing transgene. **i-k**, LW/BW and quantification of BrdU[pos] and GLUL[pos] hepatocytes 8 days after AAV8-vector administration. Biological replicates: n = 10. Bars are mean ± s.d. two tailed Mann−Whitney test. **l**, Acute BM activation combined with viral-vector delivery of a Akt transgene containing a myristoylation sequence. **m-o**, LW/BW and quantification of BrdU[pos] and GLUL[pos] hepatocytes 8 days after AAV8-vector administration. Biological replicates: n = 5. Bars are mean ± s.d. two tailed Mann−Whitney test. **p**, Acute BM activation combined with PTEN loss. The illustrations of the mouse and adenovirus in panels **h,l,p** were adapted from Medical Art Servier (https://servier.com) under a CC BY 4.0 licence. **q-s**, LW/BW and quantification of BrdU[pos] and GLUL[pos] hepatocytes 8 days after AAV8-vector administration. Biological replicates: BM n = 7, BMP n = 6. Bars are mean ± s.d. two tailed Mann−Whitney test.

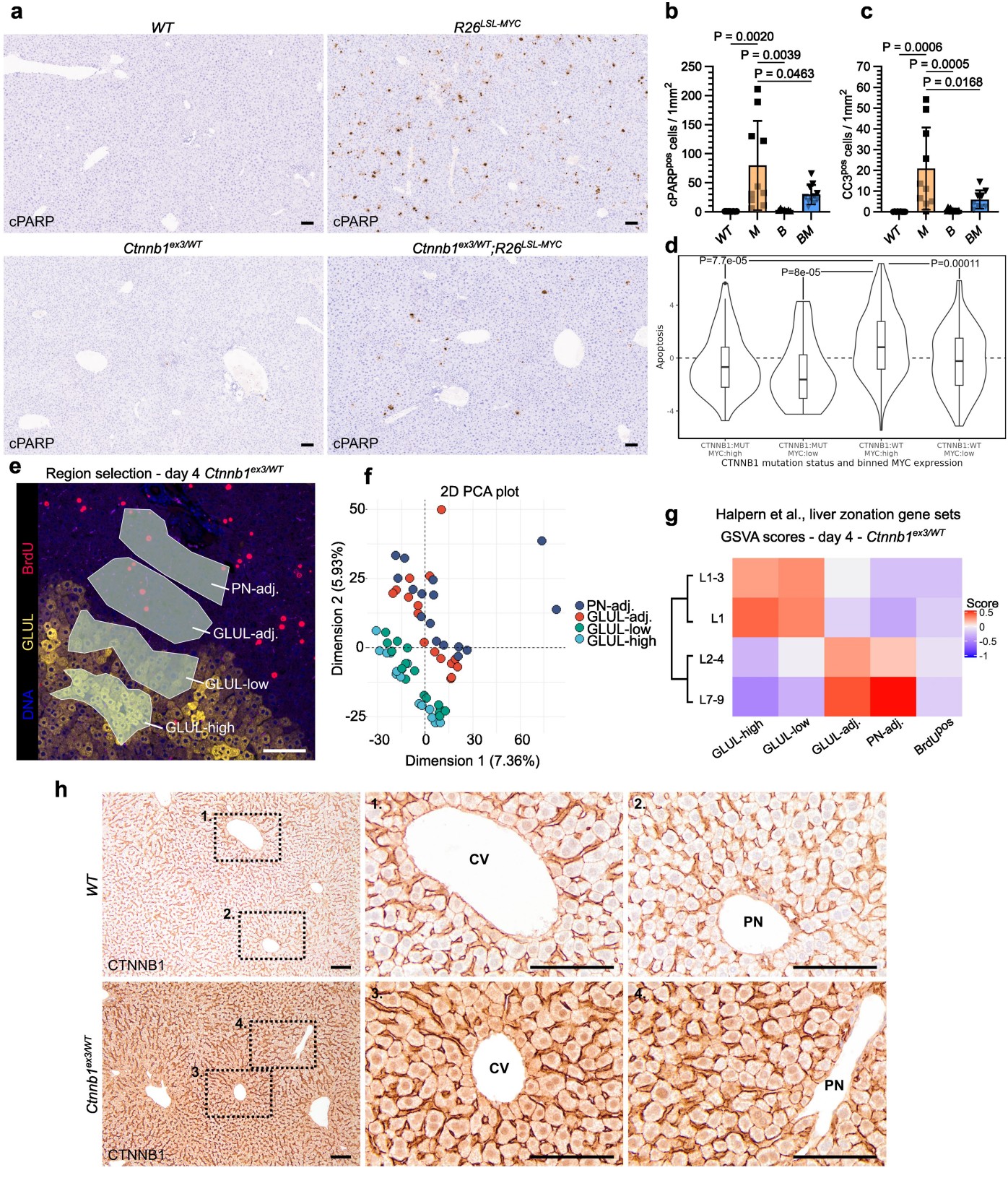

**Extended Data Fig. 4** | See next page for caption.

**Extended Data Fig. 4 | $R26^{LSL-MYC}$ liver is apoptotic and $Ctnnb1^{ex3/WT}$ mutation uniformly activates the Wnt pathway across the liver lobule. a**, Representative images of cPARP immunohistochemistry, n = 7. **b**, **c**, Quantification of apoptotic cells by IHC for cleaved PARP (cPARP) and cleaved caspase-3 (CC3) 10 days post AAV8.TBG.Cre induction. **b**, Biological replicates: $WT$ n = 8, $R26^{LSL-MYC}$ ($M$) n = 10, $Ctnnb1^{ex3/WT}$ ($B$) n = 7, $Ctnnb1^{ex3/WT}$;$R26^{LSL-MYC}$ ($BM$) n = 11. Bars are mean ± s.d. One-way ANOVA with Tukey's multiple comparisons test. **c**, Biological replicates: $WT$ n = 9, $B$ n = 11, $M$ n = 10, $BM$ n = 9. Bars are mean ± s.d. One-way ANOVA with Tukey's multiple comparisons test. **d**, Apoptosis gene set enrichment in differentially expressed genes among TCGA-LIHC binned by $CTNNB1$ mutation status and $MYC$ expression. A two sided, pairwise Wilcoxon test was used to compare data to the MYChigh/CTNNB1:WT group. Boxplots represent the median (central line) and 25th and 75th percentiles of the data (box). Whiskers represent the maximum and minimum of non-outlier values within 1.5× the interquartile range. Data beyond the whiskers are outliers. N = 372. **e-g**, NanoString GeoMx digital spatial profiling of $Ctnnb1^{ex3/WT}$ day-4 liver. **e**, Representative image of immunofluorescence masks used to define the regions selected for spatial transcriptomics. BrdU (red), GLUL (yellow), DNA (blue). PN, portal node, adj., adjacent. **f**, Principal component analysis (PCA) of spatial transcriptomics data. ROI per Biological replicate: n = 8. **g**, Gene set variance analysis (GSVA) scores for liver-zonation gene sets Lobule layers (L1–9), CV, layer 1 to PN, layer 9 according to Halpern et al.[25]. Biological replicates n = 4. **h**, Representative CTNNB1 IHC in AAV8.TBG.Cre-treated $WT$ and $Ctnnb1^{ex3/WT}$ liver at day 4 post induction, n = 3. Panels 1 - 4 are higher magnification images of the dashed areas in the left panel, highlighting hepatocytes near the PN and the CV. All scale bars, 100 µm.

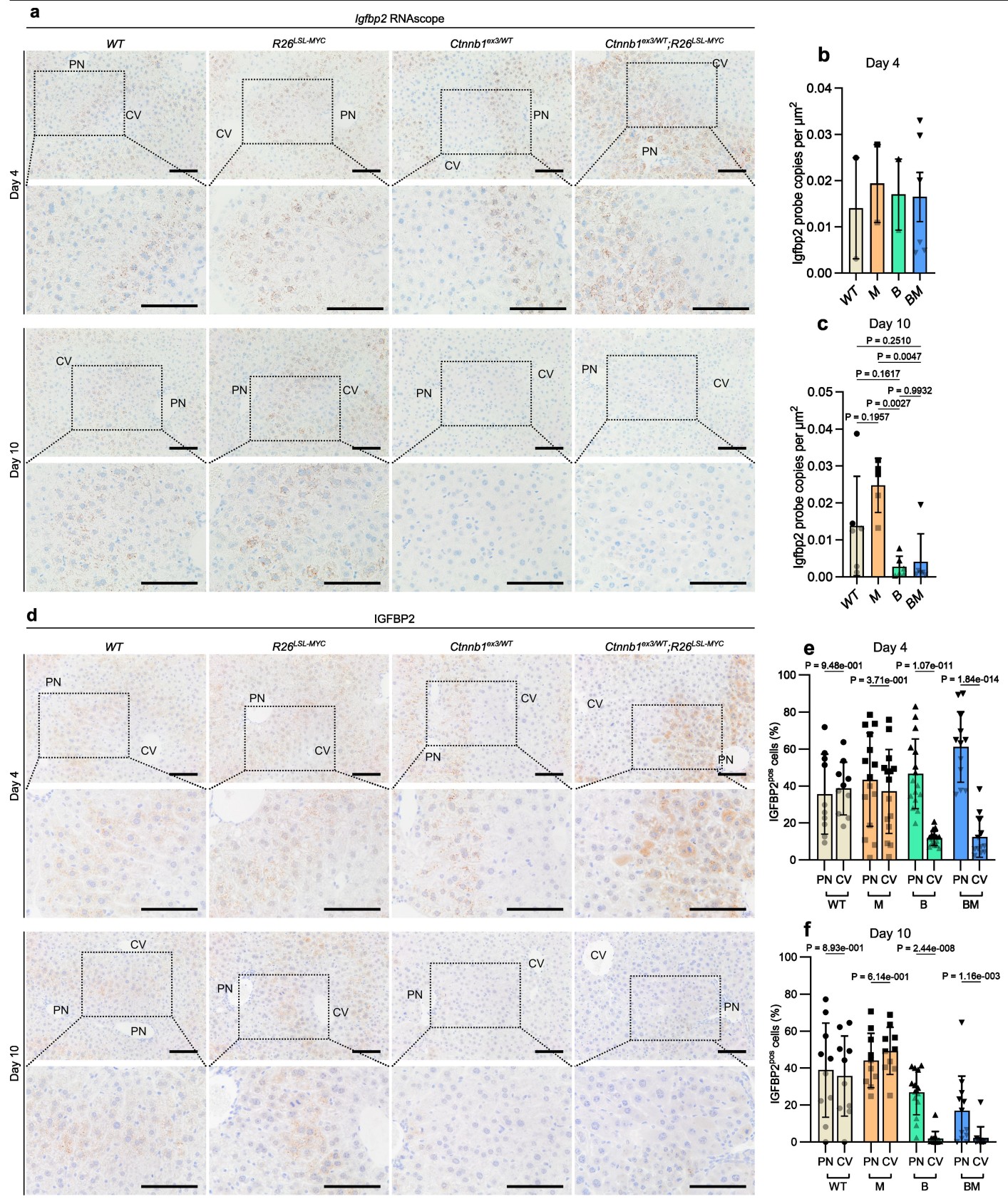

**Extended Data Fig. 5** | See next page for caption.

**Extended Data Fig. 5 | IGFBP2 expression is suppressed in day-10 *Ctnnb1^ex3/WT*-mutated liver.** AAV8.TBG.Cre-treated wild-type (*WT*), *R26^LSL-MYC^* (*M*), *Ctnnb1^ex3/WT^* (*B*), and *Ctnnb1^ex3/WT^;R26^LSL-MYC^* (*BM*) livers at days 4 and 10 post induction. **a**, Images of in situ hybridisation for *Igfbp2*. Dashed boxes indicate zoomed in regions. PN, Portal Node; CV, Central Vein. Scale bars, 100 μm. **b**, **c**, Quantification of *Igfbp2* RNAscope probe copies in livers of indicated genotypes on day 4 (**b**) and 10 (**c**) post induction. Biological replicates: day 4: *WT* n = 2, *M* n = 2, *B* n = 2, *BM* n = 6; day 10: *WT* n = 6, *M* n = 5, *B* n = 6, *BM* n = 6. Bars are mean ± s.d. One-way ANOVA with Tukey's multiple comparisons test. **d**, Images of IGFBP2 immunohistochemistry. Dashed boxes indicate zoomed in regions. PN, Portal Node; CV, Central Vein. Scale bars, 100 μm. **e**, **f**, Quantification of IGFBP2^pos^ cells in livers of indicated genotypes on day 4 (**e**) and 10 (**f**) post induction. 10 Circular regions with a Radius of 190 μm at the portal node (PN) and central vein (CV) were quantified per mouse. Biological replicates: Day 4: *WT* n = 10, *M* n = 15, *B* n = 16, *BM* n = 12; Day 10: *WT* n = 10, *M* n = 10, *B* n = 14, *BM* n = 12. Bars are mean ± s.d. One-way ANOVA with Tukey's multiple comparisons test.

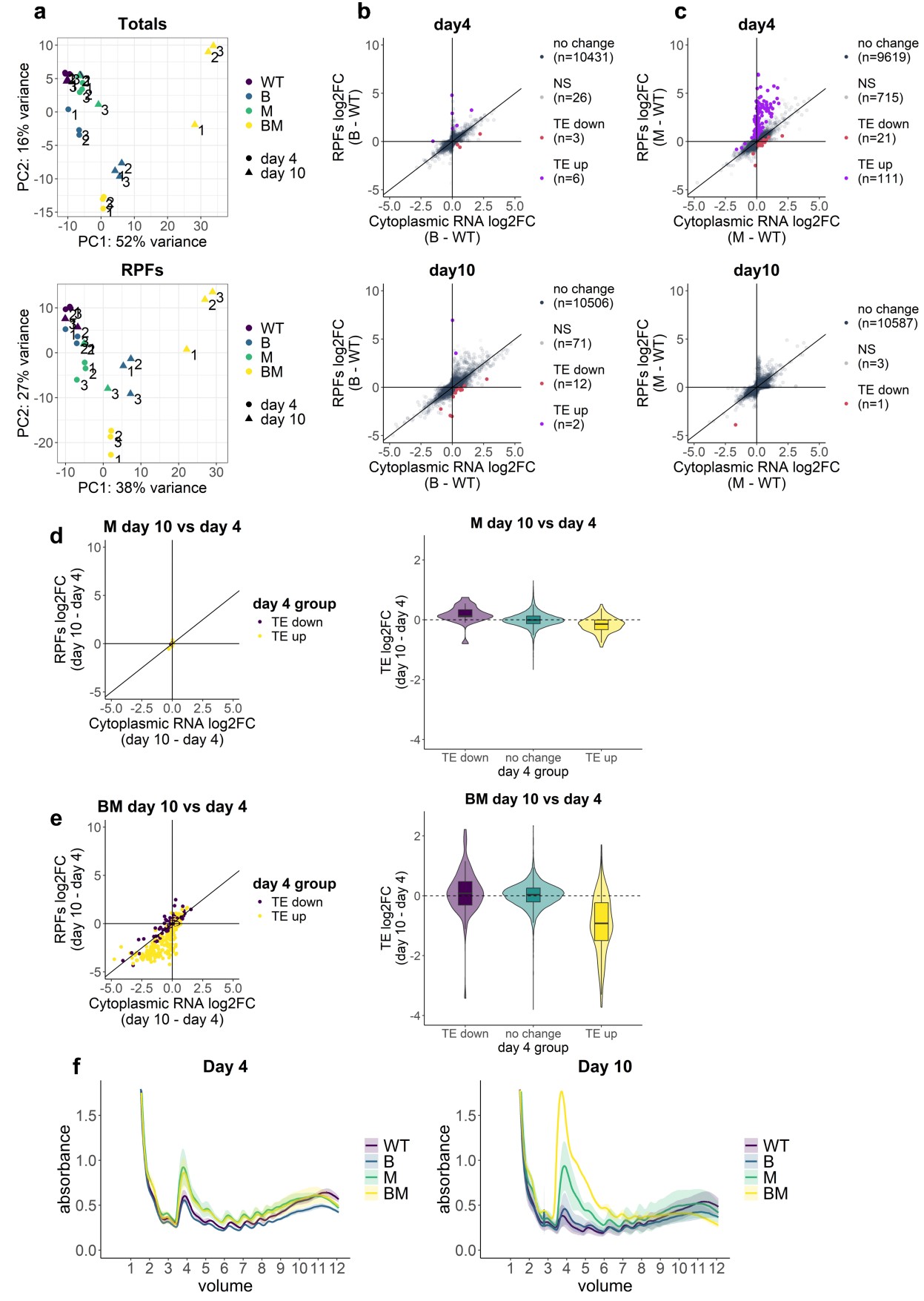

**Extended Data Fig. 6** | See next page for caption.

**Extended Data Fig. 6 | Ribosome profiling of Wnt and MYC mutated liver.**
Ribosome profiling analysis comparing AAV8.TBG.Cre-treated wild-type
(WT), *Ctnnb1*$^{ex3/WT}$ (B), *R26*$^{LSL-MYC}$ (M), and *Ctnnb1*$^{ex3/WT}$*;R26*$^{LSL-MYC}$ (BM) livers 4 and
10 days post induction. **a-e**, Ribosome sequencing (Ribo-Seq) data. Biological
replicates n = 3 per condition (time point and genotype). **a**, Principal component
analysis (PCA) of Ribo-Seq data. Totals represents all sequenced cytoplasmic
RNA, RPFs = ribosome protected fragments and represents sequenced RNA
protected by ribosomes. **b**, **c**, Scatter plot presenting translational efficiency
(TE) changes, colour scheme represents mRNAs translationally up-regulated
(TE up, purple dots, padj<0.1 and TE log2FC > 0), down regulated (TE down red
dots, padj<0.1 and TE log2FC > 0) WT livers were compared to either *Ctnnb1*$^{ex3/WT}$

(c) or *R26*$^{LSL-MYC}$ (d) livers at day 4 and 10. P values were calculated in DESeq2
using a two-sided Wald test, then multiple testing correction was applied using
the Benjamini-Hochberg method. **d**, **e**, Scatter plot presenting translational
efficiency (TE) changes and violin plots comparing day-4 and day-10 *R26*$^{LSL-MYC}$
(M) livers and *Ctnnb1*$^{ex3/WT}$*;R26*$^{LSL-MYC}$ (BM) livers, coloured by TE groups from
day 4 in panel c and day 4 in Fig. 1j. Boxplots represent the median (central line)
and 25th and 75th percentiles of the data (box). Whiskers represent the
maximum and minimum of non-outlier values within 1.5× the interquartile
range. Data beyond the whiskers are outliers. **f**, Polysome profiles. Line and
shaded areas represent mean and SD. Biological replicates n = 2.

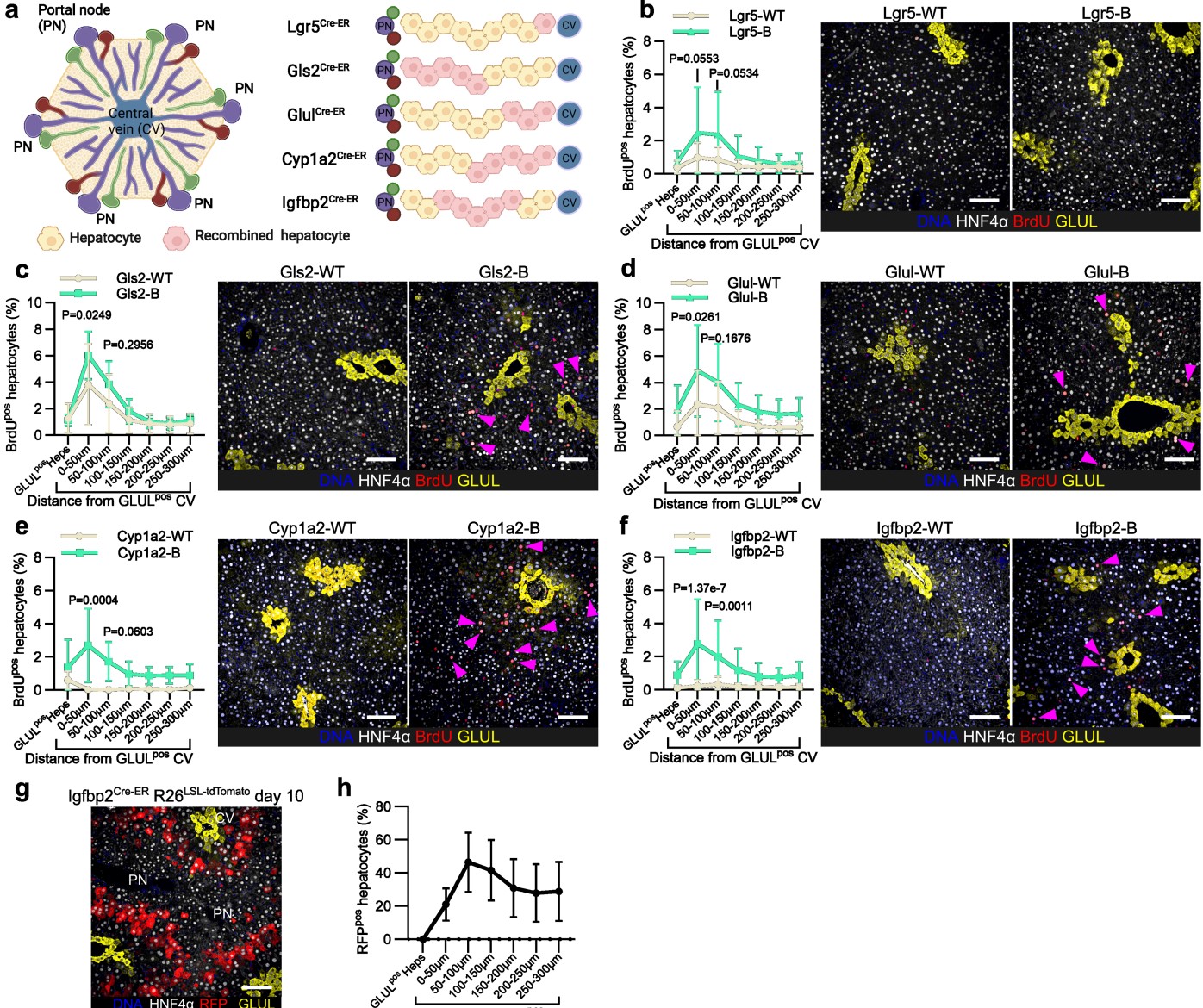

**Extended Data Fig. 7 | *Lgr5*^pos zone-3 hepatocytes are not permissive to aberrant growth driven by *Ctnnb1^ex3/WT*. a**, Diagram of the hepatic lobule and the regions various zonal specific Cre recombinases act in hepatocytes (red). The diagram was created in BioRender. Raven, A. (2025) (https://BioRender. com/fz1pkat). **b-f**, Quantification of BrdU^pos hepatocytes in relation to their distance from the GLUL^pos CV hepatocytes in liver sections of the indicated genotypes (WT = Wildtype, B = Ctnnb1^ex3/WT) and confocal immunofluorescence images of livers (magenta arrowheads highlight BrdU positive hepatocytes). All confocal immunofluorescence staining for BrdU (red), GLUL (yellow), and HNF4α (white) in liver sections. Nuclei were counterstained with DAPI (blue). Scale bars, 100 μm. **b**, *Lgr5*^CreER mice 4 days post induction. Data are mean ± s.d. Biological replicates: n = 9. Two-way ANOVA with Sidak's multiple comparison test. **c**, *Gls2*^CreER mice 4 days post induction. Data are mean ± s.d. Biological

replicates: *Gls2*-WT n = 5, *Gls2*-B n = 8. Two-way ANOVA with Sidak's multiple comparison test. **d**, *Glul*^CreER mice 4 days post induction. Data are mean ± s.d. Biological replicates: *Glul*-WT n = 8, *Glul*-B n = 11. Two-way ANOVA with Sidak's multiple comparison test. **e**, *Cyp1a2*^CreER mice 4 days post induction. Data are mean ± s.d. Biological replicates: *Cyp1a2*-WT n = 4, *Cyp1a2*-B n = 10. Two-way ANOVA with Sidak's multiple comparison test. **f**, *Igfbp2*^CreER mice 4 days post induction. Data are mean ± s.d. Biological replicates: *Igfbp2*-WT n = 13, *Igfbp2*-B n = 9. Two-way ANOVA with Sidak's multiple comparison test. **g**, Representative confocal IF staining for tdTomato (red), GLUL (yellow), and HNF4α (white) in liver sections from *Igfbp2*^Cre−ER *R26*^LSL-tdTomato mice on day 10 post induction. CV, central vein; PN, portal node. **h**, Quantification of RFP^pos hepatocytes in relation to their distance from the GLUL^pos CV in liver sections from *Igfbp2*^CreER *R26*^LSL-tdTomato mice 10 days post induction. Data are mean ± s.d. n = 11.

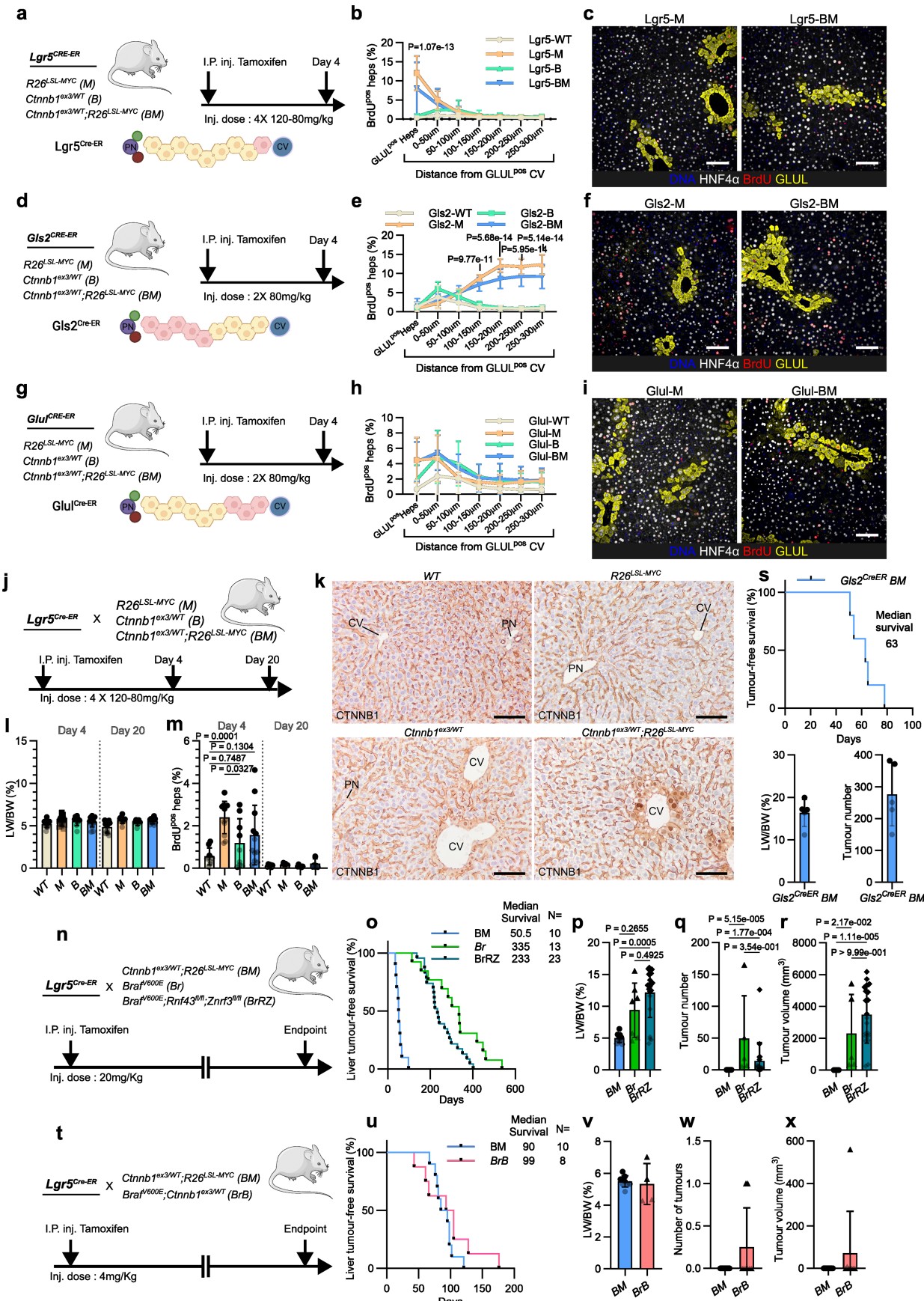

**Extended Data Fig. 8** | See next page for caption.

**Extended Data Fig. 8 | _Lgr5_<sup>pos</sup> zone-3 hepatocytes are not permissive to tumorigenesis driven by _Ctnnb1_<sup>ex3/WT</sup>;_R26_<sup>LSL_MYC</sup> mutations. a-i**, Schematics of mouse models of acute Wnt and MYC activation in relation to the promoter and creER used to induce recombination. Quantification of BrdU<sup>pos</sup> hepatocytes in relation to their distance from the GLUL<sup>pos</sup> CV zone in liver sections of the indicated genotypes (WT = Wildtype, M = _R26_<sup>LSL-MYC</sup>, B = _Ctnnb1_<sup>ex3/WT</sup>, BM = _Ctnnb1_<sup>ex3/WT</sup>; _R26_<sup>LSL-MYC</sup>) and confocal IF images of livers; BrdU (red), GLUL (yellow), and HNF4α (white) in liver sections. Nuclei counterstained with DAPI (blue). **a, b, c**, _Lgr5_<sup>CreER</sup> mice 4 days post induction. Data are mean ± s.d. Biological replicates: _Lgr5_-WT n = 9, _Lgr5_-B n = 9, _Lgr5_-M n = 12, _Lgr5_-BM n = 12. Two-way ANOVA with Tukeys multiple comparison test. **d, e, f**, _Gls2_<sup>CreER</sup> mice 4 days post induction. Data are mean ± s.d. Biological replicates: _Gls2_-WT n = 5, _Gls2_-M n = 4, _Gls2_-B n = 8, _Gls2_-BM n = 4. Two-way ANOVA with Sidak's multiple comparison test. **g, h, i**, _Glul_<sup>CreER</sup> mice 4 days post induction. Data are mean ± s.d. Biological replicates: _Glul_-WT n = 8, _Glul_-M n = 9, _Glul_-B n = 11, _Glul_-BM n = 12. Two-way ANOVA with Tukey's multiple comparison test. The schematics in panels **a, d, g** were created in BioRender. Raven, A. (2025) (https://BioRender.com/fz1pkat). **j**, CV _Lgr5_-specific model of acute Wnt and MYC activation at various time points. **k**, Representative images of CTNNB1 IHC 20 days post induction. Biological replicates n = 3. **l**, LW/BW at day 4 and day 20. Bars are mean ± s.d. Biological replicates: day 4: WT n = 12, M n = 16, B n = 10, BM n = 15;

day 10: WT n = 13, M n = 12, B n = 7, _BM_ n = 12. **m**, Quantification of BrdU<sup>pos</sup> hepatocytes at day-4 and day-20. Bars are mean ± s.d. One-way ANOVA with Tukey's multiple comparisons test. Biological replicates: day 4: WT n = 9, M n = 12, B n = 9, BM n = 12; day 10: WT n = 6, M n = 6, B n = 6, BM n = 7. **n**, CV _Lgr5_-specific model of Wnt and _Braf_<sup>V600E</sup> activation. **o-r**. Tumour free survival curve (mice had either liver or intestinal tumours), LW/BW ratios and tumour scoring. Bars are mean ± s.d. (**p**) Biological replicates: BM n = 10, _Braf_<sup>V600E</sup> (Br) n = 7, and _Braf_<sup>V600E</sup>;_Rnf43_<sup>fl/fl</sup>;_Znrf3_<sup>fl/fl</sup> (BrRZ) n = 22. (**q** and **r**) Biological replicates: BM n = 10, _Br_ n = 5, and BrRZ n = 19. One-way Kruskal-Wallis test with Dunn's multiple compraison test. **s**, Tumour free survival curve, LW/BW ratios and tumour scoring from aged _Gls2_<sup>Cre-ER</sup>;_Ctnnb1_<sup>ex3/WT</sup>;_R26_<sup>LSL-MYC</sup> livers. Biological replicates n = 5. Bars are mean ± s.d. **t**, CV _Lgr5_-specific model of _Ctnnb1_<sup>ex3/WT</sup>; _R26_<sup>LSL-MYC</sup> (BM) and _Braf_<sup>V600E</sup>;_Ctnnb1_<sup>ex3/WT</sup> (BrB) activation. The illustration of the mouse in panels **a, d, g, j, n, t** were adapted from Medical Art Servier (https://servier.com) under a CC BY 4.0 licence. **u-x**. Tumour free survival curve (mice had either liver or intestinal tumours), LW/BW ratios and tumour scoring. Bars are mean ± s.d. (**v**) Biological replicates: BM n = 9, BrB n = 4, and. (**w** and **x**) Biological replicates: BM n = 9, BrB n = 8. All Scale bars = 100 μm. Endpoint = abdominal swelling or additional signs of morbidity. Data from b, e, h is also plotted in Extended Data Fig. 7b–d.

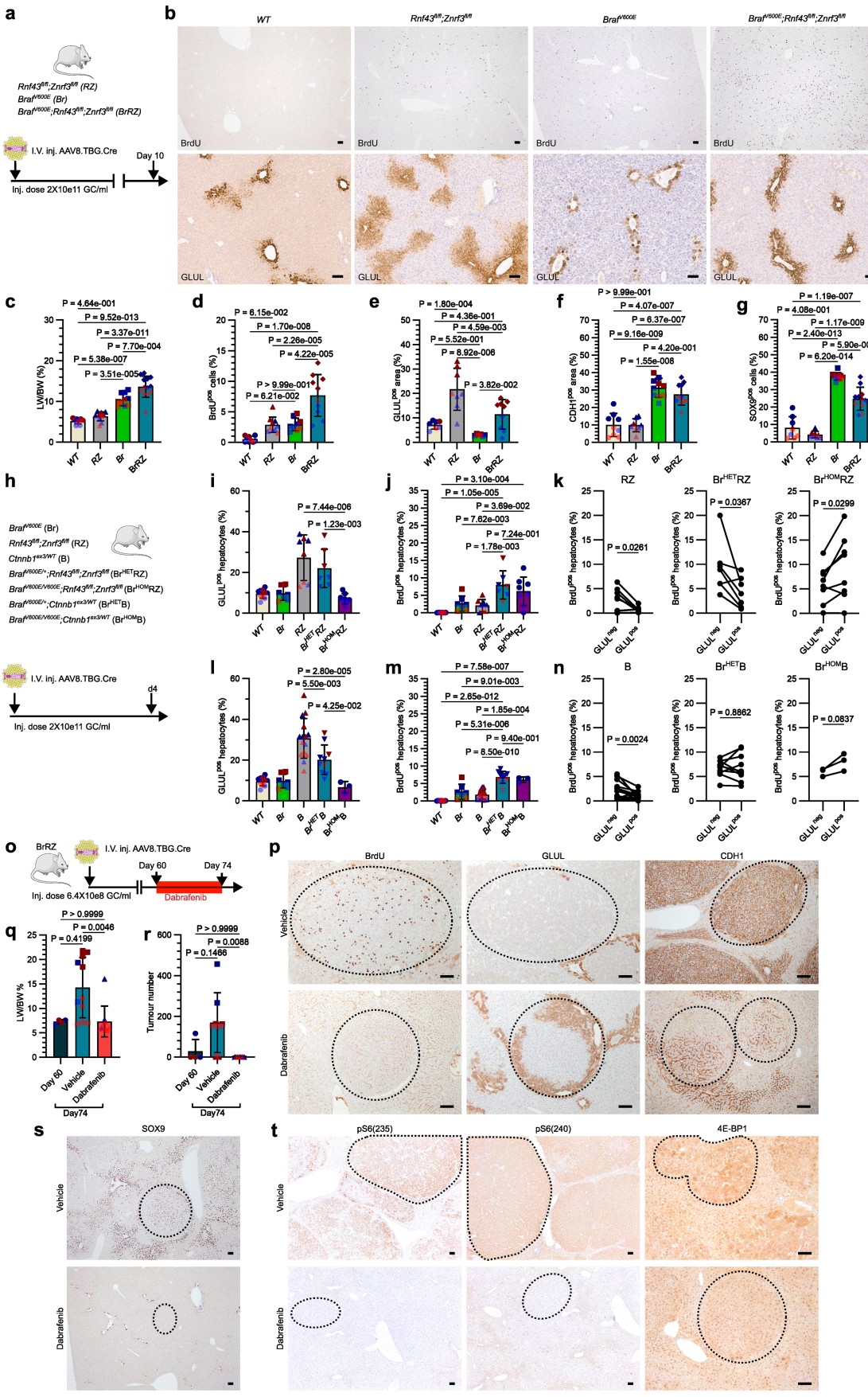

**Extended Data Fig. 9** | See next page for caption.

**Extended Data Fig. 9 | MAPK activity modulates liver zonation by promoting zone-1 and suppressing zone-3, inhibiting MAPK signalling in established tumours switches liver zonation resulting in a wnt high state and reduced mTOR activity. a**, Schematic representing liver-specific mouse model of acute Wnt (*Rnf43$^{fl/fl}$;Znrf3$^{fl/fl}$*), and *Braf$^{V600E}$* activation. **b**, Representative images of BrdU and GLUL IHC of day-10 livers, n = 6. **c**, LW/BW ratios. **d-g**, BrdU, GLUL, CDH1, and SOX9 histo-scoring of day-10 livers. Bars are mean ± s.d. One-way ANOVA with Tukey's multiple comparisons test. Biological replicates: (**c**) *WT* n = 9, *RZ* n = 10, *Br* n = 10, *BrRZ* n = 13; (**d**) *WT* n = 9, *RZ* n = 10, *Br* n = 9, *BrRZ* n = 10; (**e**) *WT* n = 7, *RZ* n = 8, *Br* n = 6, *BrRZ* n = 9; (**f**) *WT* n = 9, *RZ* n = 8, *Br* n = 10, *BrRZ* n = 11; (**g**) *WT* n = 8, *RZ* n = 8, *Br* n = 10, *BrRZ* n = 11. **h**, Schematic representing model of acute Wnt and *Braf$^{V600E}$* activation. **i-k**, quantification of BrdU and GLUL in hepatocytes 4 days after recombination of *Rnf43$^{fl/fl}$;Znrf3$^{fl/fl}$*, *Braf$^{V600E}$* or a combination of these alleles. Bars are mean ± s.d. Biological replicates: *WT* n = 10, *RZ* n = 8, *Br* n = 7, *Br$^{HET}$RZ* n = 7 and *Br$^{Hom}$RZ* n = 8. (**i**, **j**) One-way ANOVA with Tukey's multiple comparisons test. (**k**) two-sided, paired, t-test. **l-n**, quantification of BrdU and GLUL expression in hepatocytes 4 days after recombination of *Ctnnb1$^{ex3/WT}$*, *Braf$^{V600E}$* or a combination of these alleles. Bars are mean ± s.d. Biological replicates: *WT* n = 10, *B* n = 15, *Br* n = 7, *Br$^{HET}$B* n = 10 and *Br$^{Hom}$B* n = 3. (**l**, **m**) One-way ANOVA with Tukey's multiple comparisons test. (**n**) Two-sided, paired, t-test. WT and *Ctnnb1$^{ex3/WT}$* data (**i**, **j**, **l**, **m**) also plotted in Fig. 3c and Extended Data Fig. 4c,e. WT and *Braf$^{V600E}$* data repeated in figure panels i, j, l and m. **o**, Model of short-term dabrafenib treatment in established BrRZ tumours. The illustrations of the mouse and adenovirus in panels **a**,**h**,**o** were adapted from Medical Art Servier (https://servier.com) under a CC BY 4.0 licence. **p**, BrdU, GLUL and CDH1 IHC. **q**, Liver-to-body weight ratios. Biological replicates: day 60 n = 4; Vehicle, n = 11; dabrafenib, n = 10. **r**, Macroscopic tumour scoring. Biological replicates: day 60 n = 4; Vehicle, day 74 n = 8; dabrafenib, day 74 n = 7. Bars are mean ± s.d. One-way ANOVA and Dunn's multiple comparisons test. **s-t**, SOX9, pS6(Ser235/236), pS6(Ser240/244) and p4E-BP1 (Thr37/46) IHC. Dashed line, highlights tumour boundary. Males, blue points; Females, red points. All scale bars, 100 μm. The illustrations of the mouse and adenovirus were adapted from Medical Art Servier (https://servier.com) under a CC BY 4.0 licence.

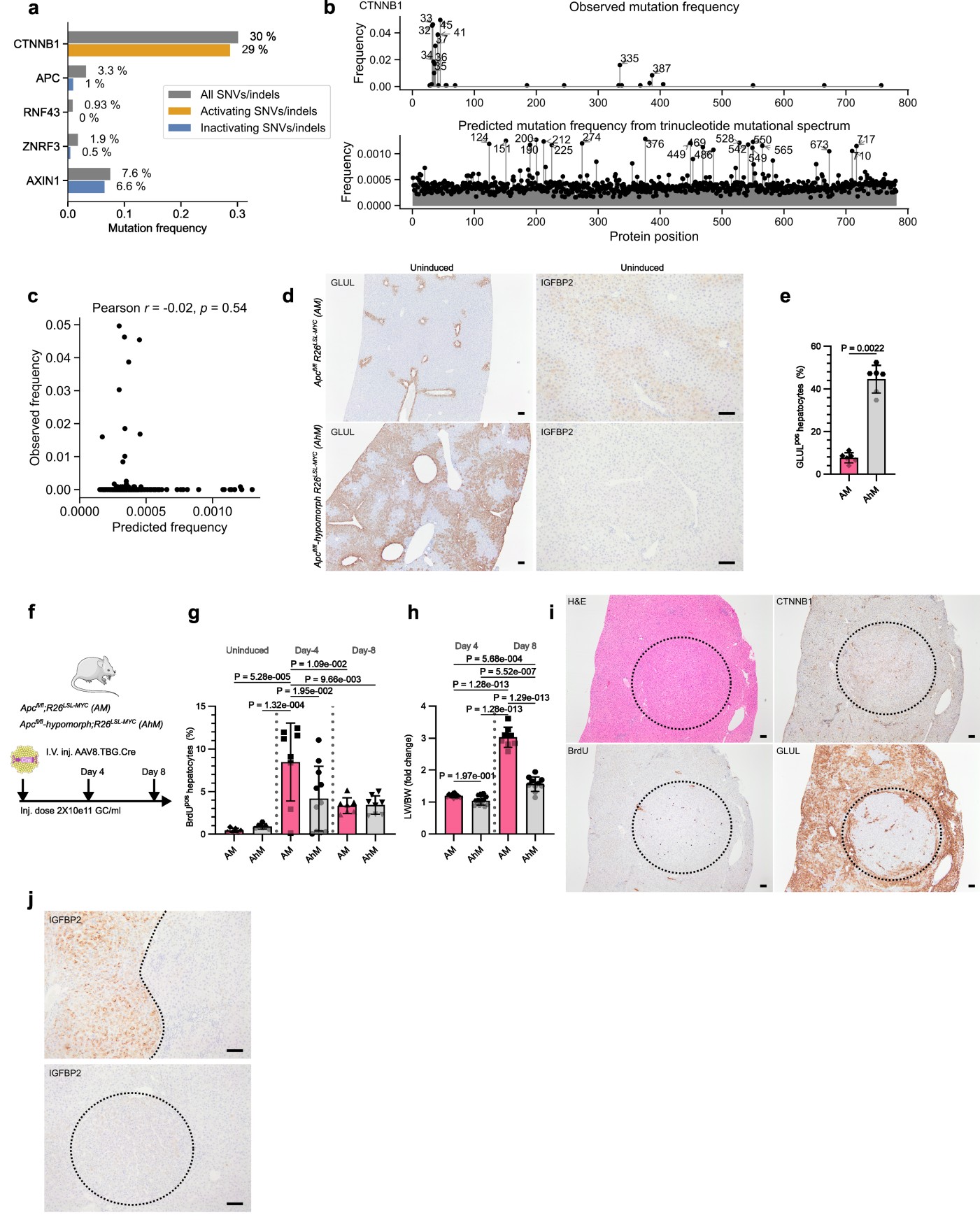

**Extended Data Fig. 10** | See next page for caption.

**Extended Data Fig. 10 | *CTNNB1* mutational modelling and Wnt-driven growth in an *Apc*-hypomorphic liver.** Modelling *CTNNB1* mutations using trinucleotide mutational spectrum. **a**, Mutation frequency of *CTNNB1, APC, RNF43, ZNRF3,* and *AXIN1* in HCC. In total 1,189 HCC samples were analysed by combining multiple cohorts (Materials and Methods). For *CTNNB1*, activating mutations are defined as missense mutations or in-frame indels at hotspots. For tumour suppressor genes, inactivating mutations are defined as nonsense mutations, splice-site mutations, and frame-shift indels. **b**, Observed *CTNNB1* mutation frequencies were compared to the predicted mutation frequencies at different protein positions. Prediction is based on the trinucleotide mutational spectrum in HCC (Materials and Methods). Only missense single-nucleotide variants (SNVs) were considered. **c**, The lack of correlation between observed and predicted mutation frequencies indicates that the hotspot *CTNNB1* mutations are mainly driven by selection advantage, rather than the underlying mutagenic processes. A two-sided p-value is associated with the Pearson correlation coefficient. **d**, GLUL and IGFBP2 IHC in uninduced livers (no administration of AAV8.TBG.Cre). **e**, Quantification of GLUL$^{pos}$ hepatocytes in uninduced mice n = 6 per genotype. Bars are mean ± s.d. Two-tailed Mann–Whitney test. **f**, Schematic of liver-specific mouse model to acutely recombine hypomorphic-*Apc*$^{fl/fl}$ and *R26*$^{LSL-MYC}$. The illustrations of the mouse and adenovirus were adapted from Medical Art Servier (https://servier.com) under a CC BY 4.0 licence. **g**, Quantification of BrdU$^{pos}$ hepatocytes. Bars are mean ± s.d. One-way ANOVA and Holm-Sidak's multiple comparisons. Biological replicates: uninduced: n = 6; Day 4: AM n = 8, AhM n = 11; Day 8: AM n = 7, AhM n = 8. **h**, Fold change in liver-to-body weight ratios. Bars are mean ± s.d. One-way ANOVA and Tukey's multiple comparisons test. Biological replicates: Day 4: AM n = 11, AhM n = 14; Day 8: AM n = 8, AhM n = 10. **i**–**j**, Aged uninduced *Apc*$^{fl/fl}$-hypomorphic liver. Dashed line, tumour boundary. GLUL, CTNNB1, BrdU and IGFBP2 immunohistochemistry, and haematoxylin and eosin (H&E) stain, n = 7. All scale bars, 100 μm.

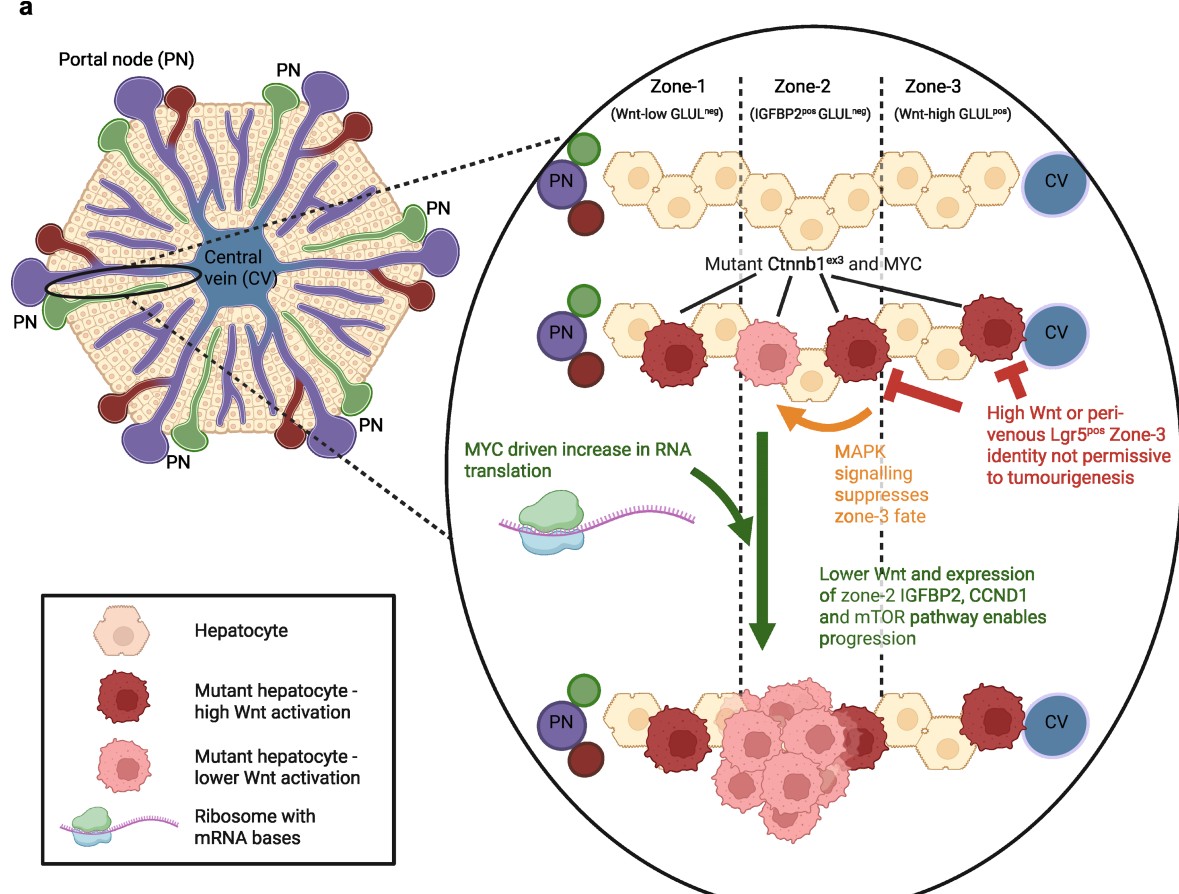

**Extended Data Fig. 11 | Graphical model of mutant-β-catenin and MYC driven tumorigenesis. a**, Mutant hepatocytes in the hepatic lobule need to avoid a high Wnt, zone-3 fate, as this is not permissive to tumorigenesis; reduced Wnt and IGFBP2 expression synergise with MYC driven RNA translation to support proliferation. PN = portal node (a vasculature structure composed from the portal vein (purple), hepatic artery (red) and bile duct (green)), CV = central vein (blue). The graphical model was created in BioRender. Raven, A. (2025) (https://BioRender.com/yzpsf78).

# Reporting Summary

## Statistics

For all statistical analyses, confirm that the following items are present in the figure legend, table legend, main text, or Methods section.

| n/a | Confirmed | |
|---|---|---|
| ☐ | ☒ | The exact sample size (*n*) for each experimental group/condition, given as a discrete number and unit of measurement |
| ☐ | ☒ | A statement on whether measurements were taken from distinct samples or whether the same sample was measured repeatedly |
| ☐ | ☒ | The statistical test(s) used AND whether they are one- or two-sided *Only common tests should be described solely by name; describe more complex techniques in the Methods section.* |
| ☐ | ☒ | A description of all covariates tested |
| ☐ | ☒ | A description of any assumptions or corrections, such as tests of normality and adjustment for multiple comparisons |
| ☐ | ☒ | A full description of the statistical parameters including central tendency (e.g. means) or other basic estimates (e.g. regression coefficient) AND variation (e.g. standard deviation) or associated estimates of uncertainty (e.g. confidence intervals) |
| ☐ | ☒ | For null hypothesis testing, the test statistic (e.g. *F*, *t*, *r*) with confidence intervals, effect sizes, degrees of freedom and *P* value noted *Give P values as exact values whenever suitable.* |
| ☒ | ☐ | For Bayesian analysis, information on the choice of priors and Markov chain Monte Carlo settings |
| ☐ | ☒ | For hierarchical and complex designs, identification of the appropriate level for tests and full reporting of outcomes |
| ☐ | ☒ | Estimates of effect sizes (e.g. Cohen's *d*, Pearson's *r*), indicating how they were calculated |

*Our web collection on statistics for biologists contains articles on many of the points above.*

## Software and code

Policy information about availability of computer code

| Data collection | Olympus cellSens imaging software (version 1.7.1), HALO image analysis software (version 2.0.1145, Indica Labs),  Columbus software (version 2.9.1.532), ZEN Black image acquisition software (version 2009), GeoMx NGS pipeline on NanoString's DND platform. Leica Aperio ImageScope software (version 12.4.3.5008) |
|---|---|
| Data analysis | recount3 package (version 1.6), GenomicDataCommons R package (version 1.12.0), DESeq2 (version 1.36),  FastQC algorithm (version 0.11.8), mouse genome build GRCm38.98 using HISAT2 (version 2.1.0.),  FeatureCounts (version 1.6.4.), DESeq2 (version 1.22.2), ReactomePA (version 1.36.0), GSA (version 1.03.1), GSVA (version 1.40.1), R (version 4.3.1), Cutadapt (version 1.18), UMI-tools (version 1.0.1), BBmap (version 38.18), samtools (version 1.9), Bowtie2 (version 2.3.5.1), RSEM (version 1.3.3), EnrichR, GraphPad Prism (version 7.0.4). Fiji/ImageJ software (version 1.53t), |

For manuscripts utilizing custom algorithms or software that are central to the research but not yet described in published literature, software must be made available to editors and reviewers. We strongly encourage code deposition in a community repository (e.g. GitHub). See the Nature Portfolio guidelines for submitting code & software for further information.

## Data

Policy information about availability of data

All manuscripts must include a data availability statement. This statement should provide the following information, where applicable:
- Accession codes, unique identifiers, or web links for publicly available datasets
- A description of any restrictions on data availability
- For clinical datasets or third party data, please ensure that the statement adheres to our policy

The RNA sequencing and spatial transcriptomic data generated in this study are publicly available through the Gene Expression Omnibus (GEO) with the following accession codes; GSE230644, GSE230110, GSE230137, GSE230144, GSE275864.

## Research involving human participants, their data, or biological material

Policy information about studies with human participants or human data. See also policy information about sex, gender (identity/presentation), and sexual orientation and race, ethnicity and racism.

| | |
|---|---|
| Reporting on sex and gender | Not applicable |
| Reporting on race, ethnicity, or other socially relevant groupings | Not applicable |
| Population characteristics | Not applicable |
| Recruitment | Not applicable |
| Ethics oversight | Not applicable |

Note that full information on the approval of the study protocol must also be provided in the manuscript.

# Field-specific reporting

Please select the one below that is the best fit for your research. If you are not sure, read the appropriate sections before making your selection.

☒ Life sciences          ☐ Behavioural & social sciences          ☐ Ecological, evolutionary & environmental sciences

For a reference copy of the document with all sections, see nature.com/documents/nr-reporting-summary-flat.pdf

# Life sciences study design

All studies must disclose on these points even when the disclosure is negative.

| | |
|---|---|
| Sample size | A priori based on historical data sets were used to ensure the smallest sample size that could give a significant difference was chosen in accordance with the 3Rs |
| Data exclusions | Regarding animal models that used an inducible Cre-lox genetic system, samples were excluded if there was evidence of a failed injection of inducing agents and evidence of impaired genetic recombination. |
| Replication | Multiple biological replicates were used to verify reproducibility of experiments. All attempts at replication were successful. All experiments were replicated at least twice. |
| Randomization | For all histological analysis the samples were randomized. For genetic studies, animals were assigned to groups according to their genotype. Treatment groups were randomly assigned however steps were taken during group assignment to avoid separating males in to singly housed cages. Selection of groups also aimed to maintain an equal sex balance. |
| Blinding | For all histological analysis the samples researchers were blinded to the genotype or treatment. Investigators were blinded during treatment regimens and at sample collection for timepoint experiments. For aging experiments it was not possible for investigators to be blind to genotype as this factor needed to be known to maintain the welfare of the experimental cohort |

# Reporting for specific materials, systems and methods

We require information from authors about some types of materials, experimental systems and methods used in many studies. Here, indicate whether each material, system or method listed is relevant to your study. If you are not sure if a list item applies to your research, read the appropriate section before selecting a response.

## Materials & experimental systems

| n/a | Involved in the study |
|---|---|
| ☐ | ☒ Antibodies |
| ☒ | ☐ Eukaryotic cell lines |
| ☒ | ☐ Palaeontology and archaeology |
| ☐ | ☒ Animals and other organisms |
| ☒ | ☐ Clinical data |
| ☒ | ☐ Dual use research of concern |
| ☒ | ☐ Plants |

## Methods

| n/a | Involved in the study |
|---|---|
| ☒ | ☐ ChIP-seq |
| ☒ | ☐ Flow cytometry |
| ☒ | ☐ MRI-based neuroimaging |

## Antibodies

| | |
|---|---|
| Antibodies used | β-catenin (1:50, 610154, BD Biosciences), glutamine synthetase (1:300, ab73593, Abcam (IF); 1:800; HPA007316, Sigma-Aldrich (IHC)), BrdU (1:400, ab6326, Abcam (IF); 1:250, 347580, BD Biosciences (IHC)), HNF4α (1:300, PP-H1415-00, Perseus Proteomics), E-cadherin (1:300, 610181, BD Biosciences), Ki67 (1:1000, 12202, Cell Signaling Technology), IGFBP2 (1:1000, PA5-81409, Invitrogen), RFP (1:1000,0 600-401-379, Rockland), cleaved caspase 3 (1:500, 9661, Cell Signaling Technology), cleaved PARP (1:1000, ab32064, Abcam), cyclin D1 (1:150, 55506, Cell Signaling Technology), peEF2 (1:100, 2331, Cell Signaling Technology), p4E-BP1 Thr37/46 (1:250, 2855, Cell Signaling Technology), pS6(Ser235/236) (1:75, 4858, Cell Signaling Technology), ribosomal protein pS6(Ser240/244) (1:1000, 5364, Cell Signaling Technology), SOX9 (1:500, AB5535, Millipore) cMYC (1:800, ab32072, Abcam) |
| Validation | All antibodys were selected on the manufacturers recommendations regarding target species. All antibodies were optimised and validated on control tissue with known expression of target antigen; location and intensity of signal where assessed to confirm correct antibody binding. |

## Animals and other research organisms

Policy information about studies involving animals; ARRIVE guidelines recommended for reporting animal research, and Sex and Gender in Research

| | |
|---|---|
| Laboratory animals | Mus Musculus with a mixed C57BL/6 background were used, experiments were started on mice aged 2–4 months. |
| Wild animals | Study did not involve wild animals |
| Reporting on sex | Mouse sex was determined at weaning by inspecting the genitalia. Experiments using the Gls2CreER only included males as there were differences in in genetic recombination between males and females. Experiments using 6.4×108 GC/ml only used males as the AAV8.TBG.Cre tropism is different between sexes. An exception to this was in the treatment of BrafV600E/+;Rnf43fl/fl;Znrf3fl/fl mice with dabrafenib and LGK974; here, equal numbers of males and females were used per experimental group. In all other animal experiments equal numbers of males and females were used in each experimental group. |
| Field-collected samples | Study did not involve samples collected from the field |
| Ethics oversight | All mouse experiments were performed according to UK Home Office regulations (project licence 70/8646 and PP3908577) following approval by the University of Glasgow Animal Welfare and Ethical Review Body. |

Note that full information on the approval of the study protocol must also be provided in the manuscript.

## Plants

| | |
|---|---|
| Seed stocks | *Report on the source of all seed stocks or other plant material used. If applicable, state the seed stock centre and catalogue number. If plant specimens were collected from the field, describe the collection location, date and sampling procedures.* |
| Novel plant genotypes | *Describe the methods by which all novel plant genotypes were produced. This includes those generated by transgenic approaches, gene editing, chemical/radiation-based mutagenesis and hybridization. For transgenic lines, describe the transformation method, the number of independent lines analyzed and the generation upon which experiments were performed. For gene-edited lines, describe the editor used, the endogenous sequence targeted for editing, the targeting guide RNA sequence (if applicable) and how the editor was applied.* |
| Authentication | *Describe any authentication procedures for each seed stock used or novel genotype generated. Describe any experiments used to assess the effect of a mutation and, where applicable, how potential secondary effects (e.g. second site T-DNA insertions, mosiacism, off-target gene editing) were examined.* |

