## [Peer Review File · Nature]

Hepatic zonation determines tumourigenic potential of mutant β -catenin

Corresponding Author: Professor Owen Sansom

Version 0:

Reviewer comments:

Referee #1

(Remarks to the Author)
Key results

In this manuscript, Raven et al contribute to the growing body of literature explaining the context in which oncogenic mutations can actually induce tumor formation. Using an extensive series of mouse models and AAV8 hepatocyte transductions (in vivo), they show that hyperactivated b-catenin, together with Myc, does not generally induce tumor formation in zone 3 hepatocytes, but can do so from other liver zones. Overall, this work is thorough, extensive and highly interesting.

Originality and significance:

The work is original and highly significant, and would be of interest to a broad field of hepatologists, cell biologists, and cancer biologists.

Limitations

My main concern is that the most important results and conclusion (to my eyes) that Zone 3 hepatocytes are not transformed by hyper-active b-catenin largely rests on the use of an Lgr5-cre mouse to drive expression "in Zone 3 hepatocytes". But this Lgr5-cre is highly restricted to the first layer of hepatocytes around the central vein, and mimics GS expression. It is thus more restricted than "Zone 3" which is composed of layers 1-3 (Itzkovitz nomenclature). The Lgr5 results would likely reflect layer 1 hepatocytes. There are a number of excellent conditional mouse models to address zonal questions, and whether Zone 3 (layers 1-3) would be competent to develop tumors induced by hyper-activated b-catenin (for example Wei Yet al Science. 2021).

Data & methodology:

The data are robust with a good number of replicates, including males and female and specifying their sex. The statistical analyses are sound. However, the authors show means and SEM in most graphs, while the recommended data presentation is means +/-SD. (see for example Barde & Barde Perspect Clin Res. 2012: "SEM quantifies uncertainty in estimate of the mean whereas SD indicates dispersion of the data from mean. As readers are generally interested in knowing the variability within sample, descriptive data should be precisely summarized with SD.")

It is also not stated whether they used paired or unpaired t-tests (I assume unpaired?) and whether/how they compared males vs males and females vs females in their different graphs (since there are quite obvious sex differences).

Conclusions: Do you find that the conclusions and data interpretation are robust, valid and reliable?

Yes, with the inclusion of additional experiments suggested.

Suggested improvements: Please list additional experiments or data that could help strengthening the work in a revision.

Minor:

Lines 124-126 describe upregulation of Igfbp2, but the referenced panels show CTNNB1 staining? With the additional quantification of Igfbp2 RNA scope (suggested below) the authors could instead reference Extended Fig 4 (perhaps this was the original intended Fig callout).

Lines 186-188 / Fig 3d,e: The authors claim that Wnt pathway activation is low in lesions compared to surrounding single-positive transduced hepatocytes. But in the absence of a transduction reporter (co-expression of GFP?) it is impossible to know whether there are also transduced low-Ctnnb1-expressing single hepatocytes.

Lines 192-193, the authors state that the downregulation of Lef1 and Sfrp2 corroborate the downregulation of canonical Wnt signaling. Are these cherry-picked genes? Could the authors instead test for regulation of Wnt activity with GSEA?

Fig 3: Panels d,e: The quantifications do not appear to support the conclusion that the lesions increased in size between days 30 and 60. In d there are an equal number of large lesions (50+) at Days 30 and 60. E is a cumulative distribution which does not tell us about absolute numbers. Based on d, the data instead suggest that there are more medium-sized tumors (16-34) at day 60 than 30. But these differences do not appear to have been statistically tested.

Extended data Fig 3:

- Are the differences in d significant?
- I suggest the authors reorder the legend in f to match the visual order in e. Especially as this also follows the top-to-bottom order in the PCA.
- Panel g: could the authors include a staining of control liver tissue?

Extended data Fig 4:

- RNA scope cannot be eye-balled since there can be unspecific hazy background which looks like signal but is not. Signal punctae should be quantified and graphs presented to support the conclusion that Igfbp2 RNA is down-regulated. Ideally the authors would quantify by zone and describe Igfbp2 regulation by zone.

References: Does this manuscript reference previous literature appropriately? If not, what references should be included or excluded?

I did not identify any missing references.

Clarity and context: Is the abstract clear, accessible? Are abstract, introduction and conclusions appropriate?

Yes

Inflammatory material: Does the manuscript contain any language that is inappropriate or potentially libelous?

No

Referee #2

(Remarks to the Author)

The manuscript by Raven et al. demonstrates that hepatocyte identity and zonal position determines the tumorigenic potential of mutant beta-catenin (encoded by CTNNB1). CTNNB1 is one of the most frequently mutated genes in hepatocellular carcinoma (HCC), a tumor type with high mortality rate. By performing in vivo experiments in mice, the main findings of the study are: i) overexpression of MYC and activation of Ctnnb1 in hepatocytes induces tumor formation; ii) overexpression of MYC and activation of Ctnnb1 in hepatocytes of zone 3 does not lead to tumor formation; iii) activation of Braf and Ctnnb1 in hepatocytes of zone 3 does lead to tumor formation; iv) Wnt signaling is decreased in MYC-Ctnnb1 lesions; v) MYC-CTNNB1 tumors are dependent on Igfbp2-mTor-Ccnd1 pathway; vi) Apc-deficient tumors present higher Wnt activation but lower tumorigenesis. These last results suggest that a "certain" Wnt activation is required (potentially representing an example of a pathway following the goldilocks principle).

The study is conceptually novel (the fact that MYC-CTNNB1 can induce tumors from zone 2 and 3 hepatocytes but not from zone 1 is exciting) but lacks mechanistic depth. The main limitation of the study is that it does not provide an explanation as to why zone 1-2 hepatocytes can be transformed by MYC-CTNNB1 and zone 3 hepatocytes cannot get transformed by MYC-CTNNB1. The proposed mechanism (increased zone 2 Igfbp2 signaling and reduction of Wnt signaling) are interesting observations but do not explain completely the observed phenotypes and the mechanism is not clear.

There are many outstanding unanswered questions.

Why can MYC-CTNNB1 transform zone 1-2 hepatocytes but not zone 3 hepatocytes? Is it a cell-intrinsic mechanism? Could it be mediated by the immune system? Could it be rescued by overexpression or ablation of one or several differentially expressed genes? Can MYC alone transform hepatocytes in zone 3? Can MYC-CTNNB1 transform hepatocytes in zones 1-2 (by using the right genetic tools and not just by "elimination")? Can MYC alone transform hepatocytes in zones 1-2? Do MYC levels affect the phenotypes? In a previous study, it has been shown that MYC can lead to (PMID: 31623618) or modulate (PMID: 32814112) Wnt activation. Is overactivation of Wnt signaling deleterious for hepatocytes? Why Braf-CTNNB1 forms tumors from zone 3 while also having CTNNB1? Are epigenetic mechanisms involved?

Other comments:

- In Fig. 1, while it is expected that a high dose of AAV8-Cre will lead to recombination in all hepatocytes, it would be important to include the control with LSL-tom. In addition, stainings for MYC and CTNNB1 should be included to understand

whether CTNNB1 levels/localization and MYC levels can somehow affect the observed effects.

- In Fig. 1, activation of CTNNB1, alone or with MYC, leads to an expansion of Glul-positive cells, suggesting that somehow activation of CTNNB1 is functional. Could it be that there is something in zone 3 hepatocytes that prevents tumorigenesis induced by MYC? Could it be that something in zones 1 and 2 is preventing activation of zone 3 genes while enabling other beta-catenin tumorigenic effects?
- In Fig. 2, MYC-CTNNB1 is unable to expand Glul-positive cells. Why is that? Could the use of Axin2-Cre or Glul-Cre also lead to the same result (lack of tumors with MYC-CTNNB1)?
- In Fig. 2, there seems to be more zone 1-2 proliferating hepatocytes in mice with MYC overexpression than without. Why is that? Is MYC expressed in those cells? Is it a tamoxifen effect?
- I think it would be important to show CTNNB1-Braf mouse experiments in Fig. 2.
- In Fig. 3F, how is MYC expression? Is it different between clones and lesions?
- The relevance of Igfbp2 is not clear since its expression is already down at day 10. In addition, according to the data posted in GSE230644, its expression doesn't seem significantly higher than in the single clones (it actually seems lower than in the single clones).
- Lef1, Smad3, etc... cannot be found in the GSE230644 dataset. How were the genes in Fig. S8 selected? And what are the statistics?
- HCC tumors have been classified as Hoshida S1, S2, and S3. S1 presents canonical Wnt activation while S3 presents liver-specific Wnt activation. How are these signatures enriched in the different samples and comparisons?

Referee #3

(Remarks to the Author)

In this study, the tumorigenic potential of Wnt pathway-activating mutations is investigated in hepatocytes of different identities and zonal position. The authors show that the combined introduction of Ctnnb1 mutations and c-Myc overexpression in hepatocytes induces tumorigenesis via induction of transient proliferation, mild Wnt pathway activation and an upregulation of processes associated with hepatocytes of the midlobular zone (zone-2), including mTOR signaling. A main conclusion of the presented work is that hepatocytes within the Wnt-high pericentral region (zone-3) are non-permissive for malignant transformation upon introduction of Ctnnb1-activating mutations.

Although this study presents a number of interesting insights in how Ctnnb1 mutations cooperate with other mutations to facilitate HCC development, the main conclusions seem partly based on indirect evidence and therefore remain insufficiently substantiated. In addition, a mechanistic understanding of the proposed model appears limited, and how the results translate to human HCC remains unclear on several points. Some of the findings contrast with previous work but explanations for these discrepancies are lacking.

Main points of concern:

1. Fig. 1: A model is proposed in which transient proliferation induced by Ctnnb1 and c-Myc mutations occurs "predominantly in GLUL-neg zone-1 and -2 hepatocytes" (page 5, line 110), after which cells adopt a zone-3 phenotype. Extended data Fig 2d however shows that also in zone-3 hepatocytes proliferation is increased by one or more orders of magnitude compared to wild-type liver. The statement that hepatocyte expansion at day 4 in this model mainly relies on zone-2 hepatocytes therefore does not seem justified.
2. The authors zoom in on expression of Igfbp2 as a marker of zone-2 hepatocytes and confirm that its expression is increased around day 4 when proliferation is high and lost around day 10 when the proliferative burst has reduced. Based on previous work showing that proliferative zone-2 hepatocytes display increased mTOR activity, the authors perform treatment of mice with rapamycin and observe reduced overall proliferation (at what day of measurement?) after induction of mutant Ctnnb1/c-Myc (Fig 1j). The authors use these results to conclude that the induced proliferative burst by mutant Ctnnb1/c-Myc is mediated by zone-2-derived cells. Evidence that Rapamycin selectively impairs proliferation of zone-2 hepatocytes and NOT zone-3 hepatocytes would be needed to substantiate this statement.
3. Using an Lgr5-Cre model, the authors aim to uncover susceptibility of zone-3, GLUL-pos hepatocytes for mutant Ctnnb1- and c-myc-mediated transformation. Introduction of mutant c-Myc, but not mutant bcat, induces proliferation of Lgr5-Cre cells, and the authors conclude that these cells therefore are not permissive for Wnt-driven growth. It is difficult to see how, based on these results, specific conclusions can be drawn for zone-3 cells, since no direct comparison is made with zone-2 cells. In particular, because in the AAV model mutant bcat also induced lower levels of proliferation when compared to mutant c-Myc, thus indicating that also zone-2 cells are likely less responsive to mutant bcat.
4. The Braf/ZR-mutant model seems somewhat out of place and presentation of the data is confusing. First, Lgr5-Cre models are applied. At what day are livers harvested and compared between the different mutations in this model (fig 2d)? Information on number and volume of tumors is lacking for this model, complicating interpretation of the data. Furthermore, what is the relevance of the Braf/RZ model for human HCC? Even though the current study indicates that Braf/RZ mutations are highly tumorigenic, BRAF mutations have very low prevalence in human HCC (0.3%) and are also not listed in Extended fig 10a.
5. In Fig. 3, the authors use a low titer AAV model to study clonal outgrowth of mutant cells and tumorigenesis. The formation of larger lesions was accompanied with a decrease in the nuclear accumulation of beta-catenin and concomitant low Wnt pathway activation, in comparison to neighboring single mutant cells with high levels of nuclear beta-catenin. These findings

are potentially interesting and hold relevance for the understanding of early tumorigenesis. The nature of the induced CTNNB1-activating mutations (loss of phospho-sites in degron) however would rather predict uniform effects, due to impaired β -catenin degradation in each hepatocyte. Insight in the mechanism (transcription? translation? nuclear exclusion?) by which nuclear β -catenin is reduced in these tumorigenic lesions would greatly strengthen the work.

6. Fig. 3: mTor signaling, Igfbp2 and Cnnd1 appear upregulated in Wnt-low lesions, but how these processes are linked to Wnt pathway activation remains unknown. Previously, upregulated mTOR signaling was shown to downregulate Wnt signaling, although the underlying mechanism appears irrelevant for CTNNB1-mutant hepatocytes (Zeng 2018, PNAS, PMID: 30297426). Furthermore, upregulation of cyclin D1 in Wnt-low hepatocytes is surprising, in view of its status as a canonical Wnt target gene (Shtutman 1999, PNAS, PMID: 10318916). These points should be discussed.

7. A recent study proposed a direct correlation between AXIN2 mRNA levels and KI67 staining in the liver, suggesting that Wnt pathway activity and proliferation are directly correlated in hepatocytes (Sun 2021, Cell Stem Cell, PMID: 34129813). Furthermore, CTNNB1 mutations were linked to a non-proliferative HCC subtype marked by high Wnt pathway activation (Rebouissou 2020, Journal of Hepatology, PMID: 31954487). Both these articles challenge the data presented in this study, where CTNNB1 mutations lead to Wnt-low lesions with high proliferation. It will be important to discuss these discrepancies in the manuscript.

8. Human HCC mainly arises in conditions of viral infection and/or chronic inflammation of the liver. Contrary to the findings presented here, liver injury models combined with lineage tracing studies indicate that zone-3-derived hepatocytes are highly susceptible to transformation (e.g. Kurosaki et al, JHEP 2021). Thus, inflammatory conditions may affect tumorigenic potential of zone-3 hepatocytes. How do the authors interpret their results against models of carcinogen-induced tumorigenesis, and the role of chronic inflammation in human HCC development?

Other points:

- Fig. 1: Only images of liver tissues at day 4 are shown (Fig. 1d), images at day 10 are missing. These will be important to help interpret the data (e.g. of Fig 1f).

- Fig. 1: After proliferation, a marked expansion of zone-3 transcripts was observed (e.g. Fig 1f). Levels of apoptosis induced by c-Myc were suppressed by introduction of Ctnnb1 mutations (Ext data Fig 3a,b). How do levels of apoptotic cells correspond with liver zones? Potentially, Ctnnb1 mutations protect from apoptosis in zone-3 hepatocytes as well. This information will help to understand the relative contribution of proliferating cells from different zones to the expanded zone-3 and to tumorigenesis.

- How data presented for 9 layers (e.g. fig 1f,g) correspond to the main regions of interest, zone-1 to -3, needs better explanation.

- Extended fig 5: what day does 'endpoint' represent for all mouse genotypes?

Minor points

- Line 117: suppressed = suppressed

Version 1:

Reviewer comments:

Referee #1

(Remarks to the Author)

The authors have not yet corrected data presentation from means +/- SEM to means +/- SD.

For comment 1.7: There is a discrepancy between the images and the quantifications. Both the Igfbp RNA and protein levels appear increased in a zoned fashion at Day 4, in Ctnnb1^{ex3}/WT; R26LSL-Lyc mice, but this is not reflected in the quantifications. Might this be because whole areas were quantified, rather than taking into account zonation, as so elegantly done in the rest of the manuscript..?

Referee #2

(Remarks to the Author)

The revised manuscript includes new data that strengthens the conclusions of the study. However, there are still several outstanding concerns.

The most important finding in the study is summarized in Fig. 2i-k, which shows that Ctnnb1-MYC expression doesn't lead to tumors from zone 3 Glul-positive Lgr5-positive hepatocytes but generates tumors from Gsl2-positive hepatocytes. Moreover,

Braf activation can generate tumors from Glul-positive Lgr5-positive hepatocytes, indicating that there is some restriction that prevents Glul-positive Lgr5-positive hepatocytes from generating tumors with Wnt-pathway activation.

Regarding the mechanism, the authors propose that Glul-positive Lgr5-positive hepatocytes are too differentiated to become transformed by Wnt pathway activation and that some level of dedifferentiation needs to occur to undergo transformation.

This potential mechanism is supported by several pieces of evidence:

- 1) Braf activation is able to restrict zone 3 differentiation and lead to tumor formation from zone 3 Glul-positive Lgr5-positive hepatocytes, potentially through Cdh1-mediated Wnt suppression.
- 2) An intermediate level of Wnt pathway activation from zone 1-2 hepatocytes (measured by Glul and Ctnnb1 staining and spatial transcriptomics) is required for tumorigenesis.
- 3) Ctnnb1-MYC-driven lesions are enriched in translation, YAP, and Igfbp2, and their inhibition reduces tumorigenesis.
- 4) Overactivation of Wnt signaling by Apc deletion is less conducive to tumor formation in combination with MYC overexpression than Ctnnb1 activation, which leads to less Wnt activation than Apc deletion.

While the data is in general compelling, additional data is needed to strengthen the mechanism.

Main concerns:

- 1) It would be important to provide additional examples of tumors that can arise from zone 3 Glul-positive Lgr5-positive hepatocytes, ideally rescuing the restriction that exists for Ctnnb1-MYC tumors. Does Igfbp2 expression enable tumor formation from zone 3 Glul-positive Lgr5-positive hepatocytes?
- 2) The fact that translation, YAP, and Igfbp2 are required for Ctnnb1-MYC tumor formation from zone 1/2 hepatocytes does not demonstrate that zone-1/2 factors are needed for Ctnnb1-MYC tumor formation from zone 3 Glul-positive Lgr5-positive hepatocytes. The text should reflect this limitation. Also, overexpression of Igfbp2 from zone 3 Glul-positive Lgr5-positive hepatocytes could help establish this point.
- 3) It would be important to show, for Figures 3 and 5, how many of the lesions and endpoint tumors present intermediate or high levels of Wnt activation.
- 4) What happens to hepatocytes with overactivated Wnt signaling? Do they become terminally differentiated or undergo cellular senescence? It is not very clear what happens to zone 3 Glul-positive Lgr5-positive hepatocytes after Ctnnb1-MYC expression or zone 1-2 hepatocytes after Wnt overactivation. They seem to be polyploid, which has been associated with senescence. Either way, those cells should be characterized better. It could also be that zone 3 Glul-positive Lgr5-positive hepatocytes and zone 1-2 hepatocytes have a different fate upon Wnt overactivation.
- 5) Could it be that Ctnnb1-MYC tumors downregulated liver-specific Wnt signaling but upregulate canonical Wnt signaling?

The study starts by studying the effects of MYC, Ctnnb1, and combine Ctnnb1-MYC expression effects on cell proliferation. This part of the study (most of figure 1 and 2) is not too robust and is quite confusing, distracting from the main message of the study. The authors could consider restructuring the paper to focus the message on the most significant findings.

- 1) For example, Ctnnb1 activation in zone 3 Glul-positive Lgr5-positive hepatocytes do not lead to BrdU incorporation while Ctnnb1-MYC expression promotes proliferation. However, that proliferation does not end up on tumor formation. The conclusion from this section of the study, "Together these data suggest that an extreme zone-3 Lgr5pos hepatocyte fate is not permissive to oncogenic Wnt-driven growth" is not accurate since there is proliferation with Ctnnb1-MYC expression.
- 2) It is not clear why Ctnnb1 activation from different models leads to the same pattern of proliferation. Is it due to tamoxifen? Soluble factors? Background signal?

Minor concerns:

- 1) It would be important to highlight that Lgr5-CreER-Ctnnb1-MYC mice die to intestinal tumors, both in the text and the graph. Maybe plot "liver tumor free survival"?
- 2) "Further to this, it appears that the heightened level of oncogenic Wnt-pathway activation, generated by Apc loss, is also not compatible with tumorigenesis in the liver, consistent with the relative absence of APC mutations in HCC compared, for example, with colorectal cancer." Maybe say "less compatible" since the Apc model does end up with tumors.
- 3) "Over time (days 4–10 post induction), two notable features became sequentially apparent: firstly, the increase in hepatocyte proliferation was transient, occurring predominantly in GLULneg zone-1 -2 hepatocytes around day 4 and diminishing by day 10 post induction (Fig. 1d,e and Extended Data Fig. 2d,e). Secondly, this transient proliferative response was succeeded by marked expansion of the zone-3 GLUL-expressing domain along the liver-lobule axis, underpinned by a significant increase in the levels of zone-3 gene transcripts (Fig. 1f and Extended Data Fig. 2b,c)." Is this expansion of pre-existent Glul-positive cells or late zone-3 induction of zone 1-2 hepatocytes? Please, clarify.

Referee #3

(Remarks to the Author)

The authors have added a convincing and impressive set of data with which they have greatly improved the manuscript. They included analysis of different inducible and zonally restricted Cre mouse models that strongly substantiate the observation that zone 3 hepatocytes are non-permissive for oncogenic transformation. Furthermore, they now provide an in-depth analysis of how Igfbp2 and mTOR signaling promote growth in Ctnnb1/Myc-mutant hepatocytes. Also, new results

were added that potentially provide an explanation of how Ctnnb1 target expression may be reduced in lesions versus clones. Finally, how the experimental setup and results support the main model and how these findings relate to existing literature is now better explained and discussed.

Taken together, the authors have satisfactorily addressed my concerns. I think the revised manuscript has been greatly strengthened and presents convincing support for an original and important novel concept that will be of interest to researchers working in diverse areas, including liver homeostasis, cancer biology, cancer genetics and cell biology.

Version 2:

Reviewer comments:

Referee #1

(Remarks to the Author)

I thank the authors for their revision and additional data, and congratulate them on important and interesting work!

Referee #2

(Remarks to the Author)

The authors have addressed the main concerns from the reviewers. The clarity of the paper could be further improved.

Response to referees' comments

Referee #1 (Remarks to the Author):

Key results

In this manuscript, Raven et al contribute to the growing body of literature explaining the context in which oncogenic mutations can actually induce tumor formation. Using an extensive series of mouse models and AAV8 hepatocyte transductions (in vivo), they show that hyperactivated b-catenin, together with Myc, does not generally induce tumor formation in zone 3 hepatocytes, but can do so from other liver zones. Overall, this work is thorough, extensive and highly interesting.

Originality and significance:

The work is original and highly significant, and would be of interest to a broad field of hepatologists, cell biologists, and cancer biologists.

Limitations

1.1)

My main concern is that the most important results and conclusion (to my eyes) that Zone 3 hepatocytes are not transformed by hyper-active b-catenin largely rests on the use of an Lgr5-cre mouse to drive expression "in Zone 3 hepatocytes". But this Lgr5-cre is highly restricted to the first layer of hepatocytes around the central vein, and mimics GS expression. It is thus more restricted than "Zone 3" which is composed of layers 1-3 (Iitzkovitz nomenclature). The Lgr5 results would likely reflect layer 1 hepatocytes. There are a number of excellent conditional mouse models to address zonal questions, and whether Zone 3 (layers 1-3) would be competent to develop tumors induced by hyper-activated b-catenin (for example Wei Yet al Science. 2021).

We thank the reviewer for this insightful question and agree a greater focus on which spatially defined hepatocytes are permissive to oncogenic wnt is an important question. To resolve which "layers" of zone3 hepatocytes may be competent or not, we have used additional, zonally restricted, Cres to explore if all of zone 3 is resistant to mutant-B-catenin or if this effect is limited to the Lgr5^{pos} zone-3 hepatocytes. To address this important question we crossed the mutant Ctnnb1^{ex3/WT} and R26^{LSL-MYC} alleles to three different inducible Cres from Wei, Y, et al. Science. 2021 (this includes: Cyp1a2^{CreER}, Glul^{CreER}, Glis2^{CreER}) and an additional inducible Cre from Lin Y, H, et al. Cell Stem Cell 2023 (Igfbp2^{CreER}). In an acute setting, induction of the R26^{LSL-MYC} allele with tamoxifen resulted in proliferation in all zones of the liver (Extended Data Fig. 2a-i). However, focusing on the Ctnnb1ex3/WT allele alone, we found that zone-3 hepatocytes adjacent to GLUL^{pos}, hepatocytes (i.e. layer2-3) did proliferate (Fig2. b-f) . Demonstrating that hepatocytes in zone-3 respond differently to oncogenic-Wnt signalling depending on their location in zone-3. The extreme region of Zone-3 around the central vein where the Lgr5pos, Glul^{pos} hepatocytes reside are not permissive to oncogenic-wnt driven growth, whereas the inner layers of zone 3 (2-3) are permissive and this coincides with the beginning of Igfbp2 expression as shown with the reporter scoring (Fig2. g,h).

Igfbp2^{Cre-ER} R26^{LSL-tdTomato} day 10

In a tumourigenic model, where we allow the mice to age, the *Gls2*CreER *Ctnnb1*^{ex3}/WT R26^{LSL}-MYC livers formed tumours 60 days post induction. Unfortunately the *Glul*CreER *Ctnnb1*^{ex3}/WT R26^{LSL}-MYC mouse had off target affects not related to the liver, which meant they could not be aged past 11days. The rapid formation of tumours in the *Gls2*CreER model supports our hypothesis that *Ctnnb1*^{ex3}/WT;R26^{LSL}-MYC tumours develop from the *Igfbp2* expressing region of the liver.

Data & methodology:

The data are robust with a good number of replicates, including males and female and specifying their sex. The statistical analyses are sound. However, the authors show means and SEM in most graphs, while the recommended data presentation is means +/-SD. (see for example Barde & Barde Perspect Clin Res. 2012: "SEM quantifies uncertainty in estimate of the mean whereas SD indicates dispersion of the data from mean. As readers are generally interested in knowing the variability within sample, descriptive data should be precisely summarized with SD."). It is also not stated whether they used

paired or unpaired t-tests (I assume unpaired?) and whether/how they compared males vs males and females vs females in their different graphs (since there are quite obvious sex differences).

We apologise to the reviewer for the lack of details. We have stated in the methods section where paired and unpaired t-tests were used. Also in the methods we have stated where only males were analysed (the low dose AAV8.tbq.cre models and the Gls2creER model). Where males and females were analysed together we endeavoured to use equal numbers of each sex. We did not compare the sexes separately in these instances.

Conclusions: Do you find that the conclusions and data interpretation are robust, valid and reliable? Yes, with the inclusion of additional experiments suggested.

Suggested improvements: Please list additional experiments or data that could help strengthening the work in a revision.

Minor:

1.2)

Lines 124-126 describe upregulation of Igfbp2, but the referenced panels show CTNNB1 staining? With the additional quantification of Igfbp2 RNA scope (suggested below) the authors could instead reference Extended Fig 4 (perhaps this was the original intended Fig callout).

We apologise for the lack of clarity. Lines 123-124 were solely referring to the extended data figure, highlighting that Wnt pathway activation shown through intensity of nuclear CTNNB1 staining was uniform across the hepatic lobule, but there were clear transcriptional differences between regions in the hepatic lobule. The comment about Igfbp2 expression was in relation to the spatial transcriptomics volcano plot. We have now separated out the references and made it clearer in the text. The new text is as follows:

“Although Wnt-pathway activation through nuclear accumulation of β -Catenin was uniform across the liver lobule, there was a clear transcriptional separation between GLULpos and GLULneg regions (Extended Data Fig. 3f,g,h), with the latter region showing significant expression of Igfbp2 (Fig. 1g).”

1.3)

Lines 186-188 / Fig 3d,e: The authors claim that Wnt pathway activation is low in lesions compared to surrounding single-positive transduced hepatocytes. But in the absence of a transduction reporter (co-expression of GFP?) it is impossible to know whether there are also transduced low-Ctnnb1-expressing single hepatocytes.

We apologise this point was not clearly explained in the text. All mutant hepatocytes have abnormal levels of cytoplasmic and nuclear B-Catenin through recombination of the Ctnnb1^{ex3/WT} mutant allele. A CTNNB1 low mutant hepatocyte from a lesion still has higher levels of CTNNB1 in both the cytoplasm and nucleus compared to a WT hepatocyte and is therefore without a reporter. We have now made this point clearer in the text. The new text is as follows:

“Although all Ctnnb1^{ex3/WT};R26^{LSL-MYC} mutant hepatocytes had elevated CTNNB1 expression, a key feature of the Ctnnb1^{ex3/WT};R26^{LSL-MYC} mutant lesions was their reduced Wnt-pathway activation, as indicated by decreased nuclear CTNNB1 positivity, when compared to neighbouring single mutant Ctnnb1^{ex3/WT};R26^{LSL-MYC} hepatocytes.”

1.4)

Lines 192-193, the authors state that the downregulation of Lef1 and Sfrp2 corroborate the downregulation of canonical Wnt signaling. Are these cherry-picked genes? Could the authors instead test for regulation of Wnt activity with GSEA?

The reviewer is correct that using two Wnt targets alone is not sufficient to state down-regulation of canonical Wnt signalling. A more accurate description, would be that a subset of canonical Wnt target genes are down-regulated, of which *lef1* and *sfrp2* were the main ones. We have compared the lesions and clones using a range of Wnt signatures from Wnt driven cancer models and from a ‘core’ gene set generated from consensus Wnt-activation in multiple tissues – most of the lesions are negatively enriched for Wnt associated genes when compared to the single clones. Most striking was the negative enrichment of LEF1 target genes in the lesions – reflecting the decrease in *Lef1* gene expression in the lesions. Interestingly, Lef1 activity restricts tumour growth in Apc-driven colorectal adenomas and loss of Lef1 in Apc-deficient intestine promotes a MYC-driven proliferation phenotype (Heino et al., 2021, PMID: 34788095). Together, the decrease in nuclear CTNNB1 the GSEA plots for Wnt-driven transcription indicate that Wnt signalling is reduced in the mutant-lesions when compared to the mutant-clones. In regards to lines 192-193, we have re-written this part of the manuscript and have removed this comment from the text to avoid confusion. However we are happy to include the data in the manuscript at the reviewer’s request.

1.5)

Fig 3: Panels d,e: The quantifications do not appear to support the conclusion that the lesions increased in size between days 30 and 60. In d there are an equal number of large lesions (50+) at Days 30 and 60. E is a cumulative distribution which does not tell us about absolute numbers. Based on d, the data instead suggest that there are more medium-sized tumors (16-34) at day 60 than 30. But these differences do not appear to have been statistically tested.

We thank the reviewer for their comment. We have altered the text to better describe the data. The new text is as follows (lines213-216):

“The size distribution of these clusters (hereafter lesions) changed between day 30 and 60, there was a trend towards increased amounts of mid-sized lesions (16-25 cells in size), and there was significantly less smaller lesions (5-10 cells in size) (Fig. 3d,e).”

1.6)

Extended data Fig 3:

- Are the differences in d significant?
- I suggest the authors reorder the legend in f to match the visual order in e. Especially as this also follows the top-to-bottom order in the PCA.
- Panel g: could the authors include a staining of control liver tissue?

Apologies for the oversight, pairwise Wilcoxon significance tests have now been included to Extended Data Fig.3 d. We have also made the key in the PCA plot easier to interpret with a change to the order of the key (Extended Data Fig. 3f). Additional control images, stained in the same immunohistochemistry batch and imaged using the same light conditions have now been added to Extended Data Fig. 3h.

1.7)

Extended data Fig 4:

- RNA scope cannot be eye-balled since there can be unspecific hazy background which looks like signal but is not. Signal punctae should be quantified and graphs presented to support the conclusion that Igfbp2 RNA is down-regulated. Ideally the authors would quantify by zone and describe Igfbp2 regulation by zone.

We thank the reviewer for this comment. We have now quantified both the RNAscope probes and the IGFBP2 immunohistochemistry to show that *Ctnnb1*^{ex3/WT} recombination throughout the liver suppresses Igfbp2 mRNA and protein expression at day 10. We have also included higher magnification images to enable staining and its location in the hepatic lobule to be more visible to the reader.

References: Does this manuscript reference previous literature appropriately? If not, what references should be included or excluded?

I did not identify any missing references.

Clarity and context: Is the abstract clear, accessible? Are abstract, introduction and conclusions appropriate?

Yes

Inflammatory material: Does the manuscript contain any language that is inappropriate or potentially libelous?

No

Referee #2 (Remarks to the Author):

The manuscript by Raven et al. demonstrates that hepatocyte identity and zonal position determines the tumorigenic potential of mutant beta-catenin (encoded by CTNNB1). CTNNB1 is one of the most frequently mutated genes in hepatocellular carcinoma (HCC), a tumor type with high mortality rate. By performing in vivo experiments in mice, the main findings of the study are: i) overexpression of MYC and activation of *Ctnnb1* in hepatocytes induces tumor formation; ii) overexpression of MYC and activation of *Ctnnb1* in hepatocytes of zone 3 does not lead to tumor formation; iii) activation of *Braf* and *Ctnnb1* in hepatocytes of zone 3 does lead to tumor formation; iv) Wnt signaling is decreased in MYC-*Ctnnb1* lesions; v) MYC-CTNNB1 tumors are dependent on *Igfbp2*-mTor-Ccnd1 pathway; vi) *Apc*-deficient tumors present higher Wnt activation but lower tumorigenesis. These last results suggest that a “certain” Wnt activation is required (potentially representing an example of a pathway following the goldilocks principle).

The study is conceptually novel (the fact that MYC-CTNNB1 can induce tumors from zone 2 and 3 hepatocytes but not from zone 1 is exciting) but lacks mechanistic depth. The main limitation of the study is that it does not provide an explanation as to why zone 1-2 hepatocytes can be transformed by MYC-CTNNB1 and zone 3 hepatocytes cannot get transformed by MYC-CTNNB1. The proposed mechanism (increased zone 2 *Igfbp2* signaling and reduction of Wnt signaling) are interesting observations but do not explain completely the observed phenotypes and the mechanism is not clear. There are many outstanding unanswered questions.

2.1)

Why can MYC-CTNNB1 transform zone 1-2 hepatocytes but not zone 3 hepatocytes? Is it a cell-intrinsic mechanism? Could it be mediated by the immune system? Could it be rescued by overexpression or ablation of one or several differentially expressed genes? Can MYC alone transform hepatocytes in zone 3? Can MYC-CTNNB1 transform hepatocytes in zones 1-2 (by using the right genetic tools and not just by “elimination”)? Can MYC alone transform hepatocytes in zones 1-2? Do MYC levels affect the phenotypes? In a previous study, it has been shown that MYC can lead to (PMID: 31623618) or modulate (PMID: 32814112) Wnt activation. Is overactivation of Wnt signaling deleterious for hepatocytes? Why *Braf*-CTNNB1 forms tumors from zone 3 while also having CTNNB1? Are epigenetic mechanisms involved?

We thank the reviewer for their positive response. To address the above statement we have focused on explaining the mechanism in more detail. In the acute recombination model, which results in pan-hepatocellular activation of *Ctnnb1*^{ex/WT} and *R26*^{LSL-MYC} alleles, we have ablated the zone-2 factor *Igfbp2* to suppress the proliferative response at day 4 (Fig1. I and Extended Data Fig. 2f,g). This result compliments the previous findings that mTOR inhibition with rapamycin also suppresses the proliferative phenotype at day 4 and explains why there are zonal differences in the proliferative response.

As Igfbp2 and mTOR are involved in the proliferative response at day 4 and mTOR activity drives RNA translation we performed ribosome profiling on day 4 and day 10 livers. Ribosome sequencing revealed a unique RNA translation programme in the day 4 *Cttnb1^{ex3/WT};R26^{LSL-MYC}* livers, which was enriched for mRNA encoding proteins involved in cell cycle and growth. This enhanced RNA translation was downregulated at day 10 when the majority of hepatocytes had differentiated to an extreme zone-3 fate (Fig. 1j-l and Extended Data Fig. 5).

Furthermore, we have validated findings from the acute model by targeting *Cttnb1^{ex3/WT};R26^{LSL-MYC}* mutant clones in the tumorigenesis assay with a shRNA against Igfbp2. Treatment between day-25 and day-60 suppressed lesion formation (Fig. 4.f-h).

The requirement for Igfbp2 and mTOR to drive an oncogenic, pro-growth RNA translation programme in mutant clones progressing to lesions was further supported by a strong enrichment of RNA translation gene sets in the lesions when compared to the single mutant clones.

In summary, the manuscript now describes a mechanism in which peri-venous, zone-3, $GLUL^{pos}$, $Lgr5^{pos}$ hepatocytes are not permissive to Wnt-driven tumorigenesis. For Wnt-driven tumourigenesis to occur an oncogenic RNA translation needs to be activated through *Igfbp2* and mTOR activity. As *Igfbp2* is a zone 2 factor, and its expression is suppressed by Wnt pathway activation – mutant clones that have lower levels of Wnt signalling or regions of the hepatic lobule where *Igfbp2* is expressed (zone2) are more likely to proliferate.

As outlined above we have further explained the mechanism. In regard to the additional questions asked by the reviewer, we have tried to answer some of them below.

The immune system does not appear to have a major role, the *Ctnnb1^{ex3/WT};R26^{LSL-MYC}* mutated liver does not have an influx of CD45^{pos} leukocytes. Additional analysis of whole liver RNA Seq showed a decrease in the amount of transcripts associated with neutrophils and monocytes and an increase towards more lymphocytes in the *Ctnnb1^{ex3/WT}* mutant livers. However, histo-scoring for CD8^{pos} cells did not reveal any changes when compared to WT liver. In human, *CTNNB1*-mutant HCC are characterised by an immune exclusion phenotype (Ruiz de Galarreta et al., 2019, PMID: 31186238, Sia et al., 2017, PMID: 2862457, and Xiao et al., 2020, PMID: 33039961), it therefore seems unlikely there is a role for the immune system to promote growth in *Ctnnb1^{ex3}*-driven HCC. Together these data suggest there is not a major role for the immune system in Wnt-driven tumourigenesis. However further detailed studies, beyond the scope of this manuscript, would be needed to validate this conclusion.

In this model the MYC transgene alone is a poor driver of HCC. The low dose AAV8.TBG.Cre administered to the livers recombined in all zones of the hepatic lobule, therefore any zonal susceptibility to MYC-driven cancer would have been observed with eventual tumourigenesis. It is worth noting that the level of MYC does influence tumourigenesis. All experiments in this manuscript that used the *R26^{LSL-MYC}* allele were homozygous for the MYC transgene, however using a heterozygote *R26^{LSL-MYC}* allele extends survival in the *Ctnnb1^{ex3/WT};R26^{LSL-MYC}* liver cancer model. Please see data below from Muller et al. another manuscript currently under submission at Nature (cohort 5 = *Ctnnb1^{ex3/WT};R26^{LSL-MYC}*-homozygote and cohort 3 = *Ctnnb1^{ex3/WT};R26^{LSL-MYC}*-heterozygote).

Adpated from Muller et al

Activation of the MYC transgene used in these models did not appear to increase Wnt signalling. We looked at Wnt gene targets that show consensus upregulation when the Wnt pathway is activated across multiple tissue types. At day 4 and 10 a subset of them, particularly the genes that negatively regulate Wnt signalling (Axin2, Notum, Rnf43, Znf3) had increased expression in the Ctnnb1^{ex3}/WT mutant livers. The MYC transgene did not appear to increase the expression of these Wnt target genes.

Braf^{V600E} combined with Ctnnb1^{ex3}/WT can still form tumours as the Braf mutation suppresses the Ctnnb1^{ex3}/WT-induced zone-3 phenotype. Reducing GLUL expression in the AAV8.TBG.cre high dose acute recombination model (please see graph below).

Finally, we have identified an RNA translation mechanism, however we have not identified the role of epigenetics in regulating Igfbp2 expression in the hepatic lobule and how oncogenic Wnt activation alters these epigenetic mechanisms. Interestingly, Yap and Taz loss prevented the progression of single mutant clones to the Wnt low lesions. Yap and Taz (hippo) signalling has been linked to cell fate in the liver and antagonising Wnt pathway activation in HCC (Yimlamai et al., 2014, PMID: 24906150 and Fitamant et al., 2015, PMID: 25772357), it may also have an important role in regulating the epigenetic changes required to modulate Wnt activation and enable progression of mutant clones to lesions. It is highly likely epigenetics has multiple roles in the evolution of Wnt-driven liver cancer but we believe this question would be better addressed in a more focused follow up study that can address a question of that scope.

Other comments:

2.2)

- In Fig. 1, while it is expected that a high dose of AAV8-Cre will lead to recombination in all hepatocytes, it would be important to include the control with LSL-tom. In addition, stainings for MYC and CTNNB1 should be included to understand whether CTNNB1 levels/localization and MYC levels can somehow affect the observed effects.

We thank the reviewer for their feedback. As expected, high dose AAV8.TBG.Cre (2×10^{11} GC/ml dosing) recombines throughout the liver resulting in pan-hepatocellular nuclear CTNNB1 accumulation in *Ctnnb1*^{ex3/WT} livers, MYC expression in hepatocytes that have the R26^{LSL-MYC} transgene and tdTomato expression in livers that have the R26^{LSL-tdTomato} reporter. As the AAV8.TBG.Cre model has been widely published and to save space in the manuscript we have only included the CTNNB1 immunohistochemistry in Extended Data Fig3 because it is relevant to interpretation of the data. However, we are happy to include the additional stains if the reviewer feels it is necessary.

2.3)

- In Fig. 1, activation of CTNNB1, alone or with MYC, leads to an expansion of Glul-positive cells, suggesting that somehow activation of CTNNB1 is functional. Could it be that there is something in zone 3 hepatocytes that prevents tumorigenesis induced by MYC? Could it be that something in zones 1 and 2 is preventing activation of zone 3 genes while enabling other beta-catenin tumorigenic effects? We thank the reviewer for their comment. In the acute model (high dose AAV8.TBG.Cre) all hepatocytes eventually up-regulate a zone 3 fate, this includes pre-induction zone 1 and 2 hepatocytes – they are not resistant to the zonal differentiation effects of *Ctnnb1*^{ex3/WT}. When this occurs there is no longer the necessary IGFBP2 and mTOR activity present in the hepatic lobule to activate the proliferative RNA translation required for oncogenic growth. MYC alone increases RNA translation but needs the additional IGFBP2/mTOR signalling axis to drive excessive growth, which is absent in the GLULpos extreme zone 3 cell state (please see data above in point 2.1).

2.4)

- In Fig. 2, MYC-CTNNB1 is unable to expand Glul-positive cells. Why is that? Could the use of Axin2-Cre or Glul-Cre also lead to the same result (lack of tumors with MYC-CTNNB1)?
 - In Fig. 2, there seems to be more zone 1-2 proliferating hepatocytes in mice with MYC overexpression than without. Why is that? Is MYC expressed in those cells? Is it a tamoxifen effect?

We thank the reviewer for their interesting question. Generally at day 4 in the tamoxifen treated livers there is an increase in proliferation stimulated by the administration of tamoxifen, this occurs in mutant and WT livers. This tamoxifen associated proliferation is not seen at later time points and likely

contributes to the proliferation seen in the R26^{LSL-MYC} mutant livers (please see below and point 1.1 in reviewer 1 section).

To investigate the impact of different zonally restricted Cres we crossed the mutant *Ctnnb1*^{ex3/WT} and R26^{LSL-MYC} alleles to three different inducible Cres from Wei, Y, et al. *Science*. 2021 (this includes: *Cyp1a2*CreER, *Glul*CreER, *Gls2*CreER) and an additional inducible Cre from Lin Y, H, et al. *Cell Stem Cell* 2023 (*Igfbp2*CreER). As MYC activation drives proliferation in all regions of the hepatic lobule (Extended Data Fig. 2e and Extended Data Fig. 6a-i), we focused solely on the *Ctnnb1*^{ex3/WT} mutant allele. We found that zone-3 hepatocytes adjacent to GLUL^{pos}, hepatocytes (i.e. layer2-3) did proliferate (Fig2. b-f). Demonstrating that hepatocytes in zone-3 respond differently to oncogenic-Wnt signalling depending on their location in zone-3. The extreme region of Zone-3 around the central vein where the *Igr5*^{pos}, *Glul*^{pos} hepatocytes reside are not permissive to oncogenic-wnt driven growth, whereas the inner layers of zone 3 (layers 2-3) are permissive and this coincides with the beginning of *Igfbp2* expression as shown with the reporter scoring (Fig2. g,h). This aligns with the mechanism laid out in point 2.1.

Igfbp2^{Cre-ER} R26^{LSL-tdTomato} day 10

Finally, in a tumourigenic model, where we allow the mice to age, the *Gls2*^{CreER} *Ctnnb1*^{ex3/WT} R26^{LSL-MYC} livers formed tumours 60 days post induction.

2.5)

- I think it would be important to show CTNNB1-Braf mouse experiments in Fig. 2.

We thank the reviewer for the for their comment. *Lgr5Cre BrafV600E;Ctnnb1ex3/WT* livers develop tumours. This has now been included in Extended Data Fig. 6 .

2.6)

- In Fig. 3F, how is MYC expression? Is it different between clones and lesions?

We thank the reviewer for this constructive comment. MYC expression was detected in both *Ctnnb1^{ex/WT};R26^{LSL-MYC}* single clones and lesions using immunohistochemistry. There did not appear to be a difference in the intensity of staining between the single clones and the lesions. This data has now been included to Extended Data Fig9. a.

2.7)

- The relevance of *Igfbp2* is not clear since its expression is already down at day 10. In addition, according to the data posted in GSE230644, its expression doesn't seem significantly higher than in

We thank the reviewer for their feedback. The original analysis of the spatial transcriptomics data was performed using the single cell RNA-seq analysis software 'Seurat' (recommended by Nanostring), which did not account for the variability in the lesions – resulting in one lesion with very high *Igfbp2* expression skewing the analysis towards significance. We have removed this volcano plot from the manuscript as it does not accurately represent the data. We still conclude that *Igfbp2* is a necessary

component of $Cttnb1^{ex/WT};R26^{LSL-MYC}$ driven tumorigenesis as it is expressed in $CTNNB1;MYC$ HCC and $Cttnb1^{ex/WT};R26^{LSL-MYC}$ tumours, and knockdown of $Igfbp2$ expression suppresses proliferation in the short term and lesion formation.

2.8)

- Lef1, Smad3, etc... cannot be found in the GSE230644 dataset. How were the genes in Fig. S8 selected? And what are the statistics?

We apologise to the reviewer, the uploaded spatial data is a filtered and normalised set from the DSP software, but the genes in this figure arose from DESeq2 which takes raw counts as input and does its own internal processing/normalisation (which resulted in more genes). The submission has now been amended to also include the raw/unnormalised counts as well as the normalised counts, in case someone wants to recreate this.

The genes in the heatmap from Extended Data Fig. 8 were selected as they have been linked to B-Catenin regulation and may explain why we see different level of nuclear B-Catenin between the lesions and single clones.

Statistical significance was determined using a Wald test with Benjamini-Hochberg multiple test correction a threshold of $p < 0.05$ was used to identify significant genes.

2.9)

- HCC tumors have been classified as Hoshida S1, S2, and S3. S1 presents canonical Wnt activation while S3 presents liver-specific Wnt activation. How are these signatures enriched in the different samples and comparisons?

We thank the reviewer for their comment. The $Cttnb1^{ex3/WT};R26^{LSL-MYC}$ lesions were significantly enriched for genes in S2 and S3 subtypes when compared to the single mutant clones. Reflecting how the mutant lesions are more closely related to Wnt driven human HCC than the single clones, which have a higher degree of Wnt-driven, Lef1 associated transcription (please point 1.4) and not the RNA translation and YAP signatures that support tumorigenesis (Fig. 3). We have now added these GSEA plots to Extended Data Fig. 9e-g. We believe this difference between lesions and clones shows how our data is relevant to human HCC, reinforcing the concept that a Wnt-MYC proliferative/translatome in the lesions enables tumorigenesis.

Referee #3 (Remarks to the Author):

In this study, the tumorigenic potential of Wnt pathway-activating mutations is investigated in hepatocytes of different identities and zonal position. The authors show that the combined introduction of Ctnnb1 mutations and c-Myc overexpression in hepatocytes induces tumorigenesis via induction of transient proliferation, mild Wnt pathway activation and an upregulation of processes associated with hepatocytes of the midlobular zone (zone-2), including mTOR signaling. A main conclusion of the presented work is that hepatocytes within the Wnt-high pericentral region (zone-3) are non-permissive for malignant transformation upon introduction of Ctnnb1-activating mutations.

Although this study presents a number of interesting insights in how Ctnnb1 mutations cooperate with other mutations to facilitate HCC development, the main conclusions seem partly based on indirect evidence and therefore remain insufficiently substantiated. In addition, a mechanistic understanding of the proposed model appears limited, and how the results translate to human HCC remains unclear on several points. Some of the findings contrast with previous work but explanations for these discrepancies are lacking.

Main points of concern:

3.1)

1. Fig. 1: A model is proposed in which transient proliferation induced by Ctnnb1 and c-Myc mutations occurs “predominantly in GLUL-neg zone-1 and -2 hepatocytes” (page 5, line 110), after which cells adopt a zone-3 phenotype. Extended data Fig 2d however shows that also in zone-3 hepatocytes proliferation is increased by one or more orders of magnitude compared to wild-type liver. The statement that hepatocyte expansion at day 4 in this model mainly relies on zone-2 hepatocytes therefore does not seem justified.

We thank the reviewer for their feedback. We apologise for not fully explaining the model in the text. Independently, both Ctnnb1^{ex3/WT} and R26^{LSL-MYC} mutations induce hepatocyte proliferation, importantly the proliferation induced by the MYC transgene is not influenced by zonal regions. However, Ctnnb1^{ex3/WT} mutations do predominantly induce proliferation in GLULneg hepatocytes. When combined, these mutations drive substantial hepatocyte proliferation – this proliferative response is biased towards GLULneg hepatocytes which at day 4 are the remaining Zone -1 and -2 hepatocytes that have not yet upregulated GLUL and differentiated to an extreme zone 3 fate. However the presence of the MYC transgene still enables proliferation in some of the GLULpos hepatocytes although at a much lower frequency than the GLUL^{neg} hepatocytes.

3.2)

2. The authors zoom in on expression of Igfbp2 as a marker of zone-2 hepatocytes and confirm that its expression is increased around day 4 when proliferation is high and lost around day 10 when the proliferative burst has reduced. Based on previous work showing that proliferative zone-2 hepatocytes display increased mTor activity, the authors perform treatment of mice with rapamycin and observe reduced overall proliferation (at what day of measurement?) after induction of mutant Ctnnb1/c-Myc (Fig 1j). The authors use these results to conclude that the induced proliferative burst by mutant Ctnnb1/c-Myc is mediated by zone-2-derived cells. Evidence that Rapamycin selectively impairs

proliferation of zone-2 hepatocytes and NOT zone-3 hepatocytes would be needed to substantiate this statement.

The Rapamycin treatment experiment was analysed at day 4 when peak proliferation occurs in the acute AAV8.TBG.Cre model, this has now been explicitly stated in the figure legend. We have carried out a more detailed characterisation of the proliferation in the Rapamycin treatment experiment comparing GLULpos and GLULneg hepatocytes. Rapamycin generally suppressed growth particularly MYC-driven hepatocyte proliferation but the inhibitor did not alter the increased prevalence of proliferation in the GLULneg hepatocytes in the *Ctnnb1^{ex3/WT};R26^{LSL-MYC}* mutant liver. Rapamycin alone also did not have a significant impact on the amount of proliferation in the *Ctnnb1^{ex3/WT}* liver. However ablation of Igfbp2 using a genetically deficient Igfbp2 mouse model significantly reduced proliferation in all genotypes. In summary, mTOR inhibition impacts the MYC phenotype, which reduces MYC-induced proliferation in all regions of the hepatic lobule. Loss of the zone-2 factor Igfbp2 also reduces proliferation validating that Igfbp2 and mTOR signalling promotes growth in the *Ctnnb1^{ex3/WT};R26^{LSL-MYC}* mutant liver

We have now further characterised the impact of Igfbp2 and mTOR activity on RNA translation by performing ribosome profiling on day 4 and day 10 livers. Ribosome sequencing revealed a unique RNA translation programme in the day 4 *Ctnnb1^{ex3/WT};R26^{LSL-MYC}* livers, which was enriched for mRNA encoding proteins involved in cell cycle and growth. This enhanced RNA translation was downregulated at day 10 when the majority of hepatocytes had differentiated to an extreme zone-3 fate (Fig. 1j-l and Extended Data Fig. 5).

3.3)

3. Using an *Lgr5*-Cre model, the authors aim to uncover susceptibility of zone-3, *GLUL*-pos hepatocytes for mutant *Ctnnb1*- and *c-myc*-mediated transformation. Introduction of mutant *c-Myc*, but not mutant *bcat*, induces proliferation of *Lgr5*-Cre cells, and the authors conclude that these cells therefore are not permissive for Wnt-driven growth. It is difficult to see how, based on these results, specific conclusions can be drawn for zone-3 cells, since no direct comparison is made with zone-2 cells. In particular, because in the AAV model mutant *bcat* also induced lower levels of proliferation when compared to mutant *c-Myc*, thus indicating that also zone-2 cells are likely less responsive to mutant *bcat*.

We thank the reviewer for this important question and agree a greater focus on which hepatocytes in the hepatic lobule are permissive to oncogenic-Wnt will improve our understanding of zone-3 susceptibility. To address this point we crossed the mutant *Ctnnb1^{ex3/WT}* and *R26^{LSL-MYC}* alleles to three different inducible Cres from Wei, Y, et al. Science. 2021 (this includes: *Cyp1a2^{CreER}*, *Glul^{CreER}*, *Gls2^{CreER}*) and an additional inducible Cre from Lin Y, H, et al. Cell Stem Cell 2023 (*Igfbp2^{CreER}*).

In an acute setting, induction of the *R26^{LSL-MYC}* allele with tamoxifen resulted in proliferation in all zones of the liver (Extended Data Fig. 2a-i). However, focusing on the *Ctnnb1^{ex3/WT}* allele alone, we found that zone-3 hepatocytes adjacent to *GLUL^{pos}*, hepatocytes (i.e. layer2-3) did proliferate (Fig2. b-f). Demonstrating that hepatocytes in zone-3 respond differently to oncogenic-Wnt signalling depending on their location in zone-3. The extreme region of Zone-3 around the central vein where the *Lgr5^{pos}*, *Glul^{pos}* hepatocytes reside are not permissive to oncogenic-wnt driven growth, whereas the inner layers of zone 3 (layers 2-3) are permissive and this coincides with the beginning of *Igfbp2* expression as shown with the reporter scoring (Fig2. g,h).

Igfbp2^{Cre-ER} R26^{LSL-tdTomato} day 10

3.4)

4. The Braf/ZR-mutant model seems somewhat out of place and presentation of the data is confusing. First, Lgr5-Cre models are applied. At what day are livers harvested and compared between the different mutations in this model (fig 2d)? Information on number and volume of tumors is lacking for this model, complicating interpretation of the data.

We apologise to the reviewer for the confusing layout of the data. Our key reason for using this data is that Braf-mutation can drive tumourigenesis from the Lgr5 positive cells in the liver and the tumours that ensue do not have properties of zone-3. This combined with the opportunity to inhibit both the Wnt pathway through porcupine inhibition and BRAF mutation through Dabrafenib also allowed us mechanistically interrogate these pathways.

We have included survival curves and tumour scoring data (Extended Data Fig. 6n-r). The Lgr5^{CreER};Ctnnb1^{ex3/WT};R26^{LSL-MYC} (lgr5-BM) mice came down around day 50.5 with small intestinal tumours caused by Lgr5Cre mediated recombination in the intestinal stem cell compartment. Due to the shorter latency of the Lgr5-BM model we inspected all livers for microscopic lesions, none were detected.

We also compared the Lgr5-BM model to the Lgr5^{CreER};Ctnnb1^{ex3/WT};Braf^{FV600E} (Lgr5-BrB) model. To increase the latency of the model we used mice that had 2 copies of the Lgr5^{CreER} allele and could be induced with lower doses of tamoxifen (4mg/Kg tamoxifen). Induction of genetic recombination in this setting resulted in similar median survivals of 90 (Lgr5-BM) and 99 (Lgr5-BrB) days. Although both models developed intestinal tumours, we could only detect liver tumours in the model with mutated Braf (Lgr5-BrB mice) but not the β -catenin/MYC model (Lgr5-BM mice).

Finally, when we induced genetic recombination of mutant $Ctnnb1^{ex3/WT};R26^{LSL-MYC}$ alleles with the zone 1 and 2 restricted $Gls2^{CreER}$ we observed rapid tumourigenesis and a median survival of 63-days

Together these data clarify the differences between the different oncogenic mutations and confirm Zone-3 $Lgr5^{pos}$ hepatocytes are not permissive to $Ctnnb1^{ex3}$ driven tumorigenesis unless additional mutations are included which can suppress the zone-3 cell state.

Furthermore, what is the relevance of the *Braf*/*RZ* model for human HCC? Even though the current study indicates that *Braf*/*RZ* mutations are highly tumorigenic, *BRAF* mutations have very low prevalence in human HCC (0.3%) and are also not listed in Extended fig 10a.

The reviewer is correct *BRAF* is rarely mutated in human HCC. In this experimental setting, we have used oncogenic *Braf*^{V600E} as a tool to investigate the influence of MAPK pathway activation in liver zonation and how oncogenic perturbations to this pathway influence Wnt-driven cancer. Again *RNF43* and *ZNRF3* mutations are not as common to HCC but are a useful genetic tool to aberrantly activate the Wnt pathway in the liver. We alluded to this in the opening paragraph of point 3.4, both genetic drivers are targetable with small molecule inhibitors allowing for experiments that can temporally suppress the effects of either mutation to look at the role of Zonation phenotypes in liver cancer formation as shown in Extended Data Fig. 8.

Although *BRAF* is not regularly mutated in HCC, upstream effectors of MAPK signalling are either genetically altered (*KRAS*, *NRAS*, *MET*, *VEGFA*) (The Cancer Genome Atlas Research Network, Cell, 2017, PMID: 28622513) or epigenetically dysregulated in HCC (Calvisi et al., JCI, 2007, PMID: 17717605). It is therefore relevant to the study of zonation in cancer, particularly as Ras and Raf activation is associated with regulating zone 1, portal node hepatocyte fates (Halpern et al. Nature, 2017, PMID: 28166538 and Extended Data Fig. 7b).

We have not included *BRAF* in Extended Data Fig. 10a (now Extended Data Fig. 11a) as this figure focuses solely on Wnt mutations in HCC and is not designed to give a general overview of genetic alterations in HCC, which has been well documented in other publications.

3.5)

5. In Fig. 3, the authors use a low titer AAV model to study clonal outgrowth of mutant cells and tumorigenesis. The formation of larger lesions was accompanied with a decrease in the nuclear accumulation of beta-catenin and concomitant low Wnt pathway activation, in comparison to neighboring single mutant cells with high levels of nuclear beta-catenin. These findings are potentially interesting and hold relevance for the understanding of early tumorigenesis. The nature of the induced CTNNB1-activating mutations (loss of phospho-sites in degron) however would rather predict uniform effects, due to impaired β -catenin degradation in each hepatocyte. Insight in the mechanism (transcription? translation? nuclear exclusion?) by which nuclear β -catenin is reduced in these tumorigenic lesions would greatly strengthen the work.

We thank the reviewer for this interesting question. *Ctnnb1* transcription is lower in the lesions than the clones, although this difference is not significant (top row of the heatmap below (significant differences are indicated by bold text of gene symbols)). RNA translation genes are highly expressed in the lesions compared to single clones and we can detect factors that promote RNA translation (peEF2, 4E-BP1, pS6) in the lesions. Together this indicates the lesions have increased RNA translation (please see point 2.1 in reviewer 2 section). We have characterised the pro-growth RNA translation seen in the *Ctnnb1*^{ex3/WT};R26^{LS-MYC} livers using Ribo-Seq, when focusing on *Ctnnb1* transcripts we did not see any changes to ribosome binding. In regards to nuclear exclusion, the lesions have significantly higher expression of E-Cadherin, which can retain β -Catenin to the cell membrane and has been shown previously to suppress cancer in the colon (Huels et al., 2015, PMID: 26240067).

Interestingly, Yap and Taz loss prevented the progression of single mutant clones to the Wnt low lesions. Yap and Taz (hippo) signalling has been linked to cell fate in the liver and antagonising Wnt pathway activation in HCC (Yimlamai et al., 2014, PMID: 24906150 and Fitamant et al., 2015, PMID: 25772357).

3.6)

6. Fig. 3: mTor signaling, Igfbp2 and Cnnd1 appear upregulated in Wnt-low lesions, but how these processes are linked to Wnt pathway activation remains unknown. Previously, upregulated mTOR signaling was shown to downregulate Wnt signaling, although the underlying mechanism appears irrelevant for CTNNB1-mutant hepatocytes (Zeng 2018, PNAS, PMID: 30297426). Furthermore, upregulation of cyclin D1 in Wnt-low hepatocytes is surprising, in view of its status as a canonical Wnt target gene (Shtutman 1999, PNAS, PMID: 10318916). These points should be discussed.

We thank the reviewer for their insightful comments. We have shown a Wnt high extreme zone-3 fate is not permissive to oncogenic-Wnt-driven growth. Reduction of Wnt pathway activation creates a permissive state for the zone-2 mTor signalling to be up-regulated. As Igfbp2 is not a Wnt target this is likely to be via a mechanism not related to Wnt signalling. In a normal, homeostatic liver, CCND1 expression is restricted to zone 2 hepatocytes (Wei et al., 2021, PMID: 33632817). Regulation of its expression therefore is uncoupled from the Wnt high zone-3 region of the liver suggesting, like Myc, Ccnd1 is not a Wnt target in the liver. Ccnd1 expression can be regulated by multiple pathways, one of them being hippo signalling and Yap/Taz activity (Mizuno et al., 2012, PMID: 22286761; and Barbosa et al., 2023, PMID: 37400441). As stated in point 3.5 there is increased Yap target gene expression in the lesions when compared to the single clones and could explain why CCND1 is expressed in the lesions.

3.7)

7. A recent study proposed a direct correlation between AXIN2 mRNA levels and KI67 staining in the liver, suggesting that Wnt pathway activity and proliferation are directly correlated in hepatocytes (Sun 2021, Cell Stem Cell, PMID: 34129813). Furthermore, CTNNB1 mutations were linked to a non-proliferative HCC subtype marked by high Wnt pathway activation (Rebouissou 2020, Journal of Hepatology, PMID: 31954487). Both these articles challenge the data presented in this study, where CTNNB1 mutations lead to Wnt-low lesions with high proliferation. It will be important to discuss these discrepancies in the manuscript.

We thank the reviewer for their insight and detailed appraisal of the manuscript. The data in this report compliments the findings of Sun et al. Although the method of Wnt activation was different (Sun et al used ligand dependent perturbations to stimulate Wnt signalling, whereas we have used ligand independent oncogenic Wnt mutations), we both observed increased proliferation away from the CV when Wnt signalling was activated (Sun et al. Fig3e). This accompanied an increase in expression of the Wnt target gene axin2 across all regions of the hepatic lobule. In regards to the statement highlighting a direct correlation between axin2 and ki67 in the liver (Sun et al Fig 2h), this was only observed in liver development and injury models. In both these contexts Wnt signalling is required for liver growth and not metabolic zonation (Tan et al., 2006, PMID: 17101329, Sekine et al.,

2007, PMID: 17256747). During liver damage Wnt ligand expression changes and could stimulate pathway activation in regions outside the central vein (Hu et al., 2007, PMID: 17983805; Yang et al., 2014, PMID: 24700412).

In regards to aligning our data to HCC classifications, we have shown that although the single clones and lesions have Wnt and Myc mutations it is only the lesions, with lower Wnt activation, that align with Hoshida subtypes S2 and S3. A simple conclusion from that analysis, is that the lesions look more like cancer than the clones but it does not identify what type of HCC it will go on to form. Presumably tumours in the non-proliferative HCC subtype underwent a growth phase at some point to form a tumour mass. Indeed, Rebouissou and colleagues have previously shown that secondary *CTNNB1* mutations occur as HCC progresses. It is compelling to entertain the possibility that mutant-*CTNNB1* and oncogenic Wnt signalling have multiple roles in HCC. Possibly a lower-Wnt pro-growth state is required early to establish the tumour followed by a high-Wnt state that drives other features of Wnt signalling such as immune exclusion phenotypes.

Although we are constrained by space the following points in the discussion address these points. The text now reads:

“Similar observations were made by Sun et al¹⁰, they showed a Wnt-induced proliferative response outside of zone-3 when the pathway was stimulated with Wnt ligands.”

“During HCC development, there may be multiple levels of ‘Wnt just right’ activation that can impact growth depending on disease progression, cancer stage³⁴ or the origin of the cancer initiating cell”

3.8)

8. Human HCC mainly arises in conditions of viral infection and/or chronic inflammation of the liver. Contrary to the findings presented here, liver injury models combined with lineage tracing studies indicate that zone-3-derived hepatocytes are highly susceptible to transformation (e.g. Kurosaki et al, JHEP 2021). Thus, inflammatory conditions may affect tumorigenic potential of zone-3 hepatocytes. How do the authors interpret their results against models of carcinogen-induced tumorigenesis, and the role of chronic inflammation in human HCC development?

We thank the reviewer for their comment. We agree liver damage and inflammation will have an important impact on tumorigenesis. Some of the carcinogen based models in the mouse are difficult to interpret as a lot of the toxic compounds (e.g. DEN) used to damage the liver are metabolised in zone-3 hepatocytes and therefore mutagenesis and inflammation between the zones will be different. Also, we propose that Lgr5^{pos}, GLUL^{pos}, Zone-3 hepatocytes are specifically not susceptible to Wnt-driven tumorigenesis. However, other mutations can transform these hepatocytes such as MAPK activating mutations like *Braf*^{FV600E} or in the case of Kurosaki et al, *PIK3CA* mutations.

In regards to *CTNNB1*-mutant HCC, it is worth noting that there is an increased prevalence of *CTNNB1* mutations in MASLD-driven HCC (Wong et al., 2022, PMID: 35351523). In mouse models of steatosis (shown below) we do observe alterations to Wnt signalling (reduction in *axin2*) and increases in MAPK gene targets (*Dusp6*). This suggests a background of steatohepatitis is more permissive to Wnt mutations because it alters the Wnt signalling gradient and consequentially metabolic zonation. However, this needs to be validated with follow up studies.

Other points:

3.9)

- Fig. 1: Only images of liver tissues at day 4 are shown (Fig. 1d), images at day 10 are missing. These will be important to help interpret the data (e.g. of Fig 1f).

We thank the reviewer for their feedback. Day 10 images for $R26^{LSL-MYC}$ and $Ctnnb1^{ex3/WT}$ livers have been added to Extended Data Fig. 2b.

3.10)

- Fig. 1: After proliferation, a marked expansion of zone-3 transcripts was observed (e.g. Fig 1f). Levels of apoptosis induced by c-Myc were suppressed by introduction of *Ctnnb1* mutations (Ext data Fig 3a,b). How do levels of apoptotic cells correspond with liver zones? Potentially, *Ctnnb1* mutations protect from apoptosis in zone-3 hepatocytes as well. This information will help to understand the relative contribution of proliferating cells from different zones to the expanded zone-3 and to tumorigenesis.

Apoptosis occurs uniformly across the hepatic lobule when the $R26^{LSL-MYC}$ allele is recombined with AAV8.TBG.Cre. To investigate if zone-3 hepatocytes respond differently, we have stained $Lgr5^{CreER}$ mice with or without the $Ctnnb1^{ex3/WT}$ $R26^{LSL-MYC}$ alleles. In this model we could detect apoptotic hepatocytes around the central vein in the $R26^{LSL-MYC}$ and $Ctnnb1^{ex3/WT}$ $R26^{LSL-MYC}$ livers (blue arrowheads), with a trend towards increased apoptosis in the $Lgr5^{CreER}$ $R26^{LSL-MYC}$ only mice.

3.11)

- How data presented for 9 layers (e.g. fig 1f,g) correspond to the main regions of interest, zone-1 to -3, needs better explanation.

We thank the reviewer for their feedback and have now included a schematic in the figure to better explain how the layers relate to the regions of interest.

3.12)

- Extended fig 5: what day does 'endpoint' represent for all mouse genotypes?

We apologise to the reviewer for this vague term and have now included time points and a definition of endpoint in the figure legend.

3.13)

Minor points

- Line 117: suppressed = suppressed

We thank the reviewer for pointing out this mistake and have now corrected the spelling error.

Response to referees' comments

Referee #1 (Remarks to the Author):

1.1)

The authors have not yet corrected data presentation from means \pm SEM to means \pm SD.

We apologise for this oversight and can now confirm data presentation has been changed from mean \pm s.e.m. to the requested mean \pm s.d.

1.2)

For comment 1.7: There is a discrepancy between the images and the quantifications. Both the Igfbp RNA and protein levels appear increased in a zoned fashion at Day 4, in *Ctnnb1^{ex3/WT}; R26^{LSL-Lyc}* mice, but this is not reflected in the quantifications. Might this be because whole areas were quantified, rather than taking into account zonation, as so elegantly done in the rest of the manuscript..?

We thank the reviewer for their comment and have re-run our image analysis to include specified regions. Using circles with a radius of 190 μ m, we have measured the number of IGFBP2 positive cells in proximity to either the portal node (PN) or central vein (CV). Interestingly, when *Ctnnb1* is mutated IGFBP2 expression shifts at day-4 to a more portal node position which compensates for a reduction in IGFBP2 expression in the zone-2 region directly adjacent to zone-3 (please see graphs below). At day-10 IGFBP2 expression is generally suppressed with reduced levels detectable at the portal node region and near complete loss at the central vein region. This fits with a model whereby hyper-activation of Wnt signalling results in a progressive shift of zone-3 across the hepatic lobule as hepatocytes lose their zone-1 and -2 identity and progress to a zone-3 fate. This data has now been added to Extended Data Fig.6e and f.

Figure: Quantification of IGFBP2^{pos} cells in livers of indicated genotypes (WT=wild type, M=R26^{LSL-MYC}, B=Ctnnb1^{ex3/WT}, BM= Ctnnb1^{ex3/WT};R26^{LSL-MYC}) on day 4 and 10 post induction. 10 Circular regions with a Radius of 190 μ m at the portal node (PN) and central vein (CV) were quantified per mouse. Biological replicates: Day 4: WT n=10, M n=15, B n=16, BM n=12; Day 10: WT n=11, M n=10, B n=14, BM n=12. Data are mean \pm s.d. One-way ANOVA with Tukey's multiple comparisons test.

Referee #2 (Remarks to the Author):

The revised manuscript includes new data that strengthens the conclusions of the study. However, there are still several outstanding concerns.

The most important finding in the study is summarized in Fig. 2i-k, which shows that Ctnnb1-MYC expression doesn't lead to tumors from zone 3 Glul-positive Lgr5-positive hepatocytes but generates tumors from Gsl2-positive hepatocytes. Moreover, Braf activation can generate tumors from Glul-positive Lgr5-positive hepatocytes, indicating that there is some restriction that prevents Glul-positive Lgr5-positive hepatocytes from generating tumors with Wnt-pathway activation.

Regarding the mechanism, the authors propose that Glul-positive Lgr5-positive hepatocytes are too differentiated to become transformed by Wnt pathway activation and that some level of dedifferentiation needs to occur to undergo transformation.

This potential mechanism is supported by several pieces of evidence:

- 1) Braf activation is able to restrict zone 3 differentiation and lead to tumor formation from zone 3 Glul-positive Lgr5-positive hepatocytes, potentially through Cdh1-mediated Wnt suppression.
- 2) An intermediate level of Wnt pathway activation from zone 1-2 hepatocytes (measured by Glul and Ctnnb1 staining and spatial transcriptomics) is required for tumorigenesis.
- 3) Ctnnb1-MYC-driven lesions are enriched in translation, YAP, and Igfbp2, and their inhibition reduces tumorigenesis.
- 4) Overactivation of Wnt signaling by Apc deletion is less conducive to tumor formation in combination with MYC overexpression than Ctnnb1 activation, which leads to less Wnt activation than Apc deletion.

While the data is in general compelling, additional data is needed to strengthen the mechanism.

We thank the reviewer for their constructive feedback, we have now produced additional data that both strengthens and clarifies the mechanisms required for Wnt-driven tumourigenesis to occur. Please find our additional data and comments detailed below.

Main concerns:

2.1)

1) It would be important to provide additional examples of tumors that can arise from zone 3 Glul-positive Lgr5-positive hepatocytes, ideally rescuing the restriction that exists for Ctnnb1-MYC tumors. Does Igfbp2 expression enable tumor formation from zone 3 Glul-positive Lgr5-positive hepatocytes?

We agree with the reviewer, understanding which signalling modalities or factors make zone-3 differentiated $GLUL^{pos}$, $Lgr5^{pos}$ hepatocytes permissive to Wnt-driven cancer is an important question. We have come to the conclusion that the key signalling modality that enables $Glul^{pos}$ $Lgr5^{pos}$ hepatocytes to become susceptible to Wnt-driven cancer is MAPK signalling, as it reverses zone-3 differentiation. This is evidenced in multiple ways throughout the manuscript. Firstly, we have shown using three different models ($Lgr5^{CreER}$ $Braf^{V600E}$, $Lg5^{CreER}$ $Braf^{V600E};Rnf43^{fl/fl};Znrf3^{fl/fl}$ and $Lgr5^{CreER}$ $Braf^{V600E};Ctnnb1^{ex3/WT}$) that $Lgr5$ positive zone-3 hepatocytes form tumours when MAPK signalling is aberrantly activated via a $Braf^{V600E}$ mutation. This data was presented in previous manuscript versions and is in Fig. 4 and Extended Data Fig. 9 of the current version. Furthermore, we have now generated additional data that shows progressive increases in MAPK signalling through $Braf^{V600E}$ allele copy number gain severely suppresses $GLUL$ expression and increases proliferation in the

Figure: **a**, Schematic representing liver-specific murine model of acute Wnt and $Braf^{V600E}$ activation. **b-d**, quantification of $BrdU$ and $GLUL$ expression in hepatocytes 4 days after acute recombination of $Rnf43^{fl/fl};Znrf3^{fl/fl}$, $Braf^{V600E}$ or a combination of these alleles. Data are mean \pm s.d. Biological replicates: WT n=10, RZ n=8, Br n=7, Br^{HETRZ} n=7 and Br^{HOMRZ} n=8. **(b, c)** One-way ANOVA with Tukey's multiple comparisons test. **(d)** paired t-test. **e-g**, quantification of $BrdU$ and $GLUL$ expression in hepatocytes 4 days after acute recombination of $Ctnnb1^{ex3/WT}$, $Braf^{V600E}$ or a combination of these alleles. Data are mean \pm s.d. Biological replicates: WT n=10, B n=15, Br n=7, Br^{HETB} n=10 and Br^{HOMB} n=3. **(e, f)** One-way ANOVA with Tukey's multiple comparisons test. **(g)** paired t-test. Males, blue points; Females, red points.

remaining, peri-central vein, GLUL^{pos} hepatocytes (please attached figure and data presented in Extended Data Fig. 10h-n).

Importantly even in established lesions MAPK signalling inhibition both promotes Zone-3 differentiation and suppresses mTOR activation. This is clearly observed in the *Braf^{V600E};Rnf43^{fl/fl};Znrf3^{fl/fl}* tumour model where treatment with Dabrafenib (Braf-inhibitor) switches the zonal marks in the lesions from zone-1/2 to zone-3 and suppresses mTOR activity

Figure: **a**, Schematic representing a Model of short-term dabrafenib treatment in established *Braf^{V600E};Rnf43^{fl/fl};Znrf3^{fl/fl}* tumours. **b**, BrdU, GLUL and CDH1 immunohistochemistry. **c**, Liver-to-body weight ratios. **d**, Macroscopic tumour scoring. **e,f**, SOX9, pS6(Ser235/236), pS6(Ser240/244) and p4E-BP1 (Thr37/46) immunohistochemistry. Scale bars, 100 μm. Dashed line, highlights tumour boundary. Males, blue points; Females, red points. Data are mean ± s.d. One-way ANOVA and Dunn's multiple comparisons test. Biological replicates: day 60 n=4; Vehicle, day 74 n=8; dabrafenib, day 74 n=7.

with a reduction in pS6 and 4E-BP1 staining (please see attached figure and data presented in Extended Data Fig. 11a-f). This aligns strongly with the data from the *Ctnnb1^{ex3/WT};R26^{LSL-MYC}* model where an mTOR driven proliferative translome is critical in supporting growth in zone-1/2 and the mutant lesions. Together these data provide important mechanistic explanation for how MAPK signalling drives tumourigenesis in the Wnt-high zone-3 hepatocyte population.

Importantly we also detected endogenous MAPK activation in the proliferative lesions that develop in the *Ctnnb1^{ex3/WT};R26^{LSL-MYC}* model. This is evidenced by the spatial transcriptomic data that compares proliferative lesions to single mutant-clones, in this setting we observed a significant enrichment of genes upregulated in a *Braf^{V600E}* mutant liver also present in the mutant lesions. Conversely, genes suppressed by *Braf^{V600E}* activation were significantly enriched in the single clones and not the proliferative lesions (please see attached figure below and data presented in Fig. 1k,l).

Figure: Representative image of immunofluorescence mask used to define the regions selected for spatial transcriptomics. BrdU (red), GLUL (yellow), DNA (blue). Scale bar, 100 μ m. Gene set enrichment analysis (GSEA) from the spatial transcriptomics assay; GSEA signature generated from genes up-regulated and down-regulated in day-10 *Braf*^{V600E} mutated liver.

We have also explored if additional signalling components can overcome the restrictions of zone-3 differentiation – via activation of Akt or artificial overexpression of IGFBP2. We have attempted this using various AAV8 delivered transgenes or genetically engineered mouse models. Firstly, we used an AAV8.CMV.Igfbp2 virus to over-express IGFBP2 in the *Ctnnb1*^{ex3/WT};*R26*^{LSL-MYC} model. Addition of IGFBP2 did not promote proliferation in the terminally differentiated zone-3 hepatocytes at day-8. Indicating IGFBP2 is required but not sufficient alone to reverse differentiation and induce proliferation (please see attached figure below and data presented in Extended Data Fig 4h-k).

Figure: **a**, IGFBP2 immunohistochemistry in AAV8.CMV.Igfbp2 treated livers or an AAV8.CMV.Egfp control. **b**, Schematic representing the liver-specific murine model of acute Wnt and MYC activation combined with viral-vector delivery of an *Igfbp2* expressing transgene. **c-e**, Liver-to-body weight ratios and quantification of BrdU^{pos} and GLUL^{pos} hepatocytes 8 days after AAV8-vector administration. Biological replicates: n=10. Data are mean \pm s.d. two tailed Mann-Whitney test.

To promote mTOR activity, we also stimulated Akt signalling in the differentiated zone-3 hepatocytes at day-8 using a conditional PTEN knockout allele and an AAV8 virus expressing an Akt allele with a myristoylation sequence. Although PTEN loss resulted in larger livers, both approaches did not promote proliferation in the zone-3 hepatocytes, again indicating that Akt activation alone is not sufficient to induce proliferation in the $GLUL^{pos}$ zone-3 differentiated hepatocytes (please see attached figure below and data presented in Extended Data Fig 4l-s).

Figure: **a**, Schematic representing the liver-specific murine model of acute Wnt and MYC activation combined with PTEN loss. **b-d**, Liver-to-body weight ratios and quantification of $BrdU^{pos}$ and $GLUL^{pos}$ hepatocytes 8 days after AAV8-vector administration. Biological replicates: BM $n=7$, BMP $n=6$. Data are mean \pm s.d. two tailed Mann-Whitney test. **e**, Schematic representing the liver-specific murine model of acute Wnt and MYC activation combined with viral-vector delivery of a Akt transgene with a myristoylation sequence. **f-h**, Liver-to-body weight ratios and quantification of $BrdU^{pos}$ and $GLUL^{pos}$ hepatocytes 8 days after AAV8-vector administration. Biological replicates: $n=5$. Data are mean \pm s.d. two tailed Mann-Whitney test.

In summary, MAPK signalling is sufficient to drive tumorigenesis from the Wnt-high zone-3 hepatocytes. In contrast mTOR activity and IGFBP2 are necessary for tumourigenesis, but alone they are not sufficient for oncogenic-growth in terminally differentiated zone-3 hepatocytes. Together these data clarify which additional requirements enable differentiated zone-3 hepatocytes to progress to tumorigenesis, substantially strengthening the findings of the manuscript.

2.2)

2) The fact that translation, YAP, and Igfbp2 are required for $Ctnnb1$ -MYC tumor formation from zone 1/2 hepatocytes does not demonstrate that zone-1/2 factors are needed for $Ctnnb1$ -MYC tumor formation from zone 3 $Glul$ -positive $Lgr5$ -positive hepatocytes. The text should reflect this limitation. Also, overexpression of $Igfbp2$ from 3 $Glul$ -positive $Lgr5$ -positive hepatocytes could help establish this point.

We thank the reviewer for their comment and agree this point requires clarification. We show that the key signalling modalities that determine zonal state also impact tumourigenesis. MAPK activity, which regulates peri-nodal zonation and zone-1/2 identity is required to reverse zone-3 differentiation. The data, detailed above, shows activation of MAPK signalling and the subsequent up-regulation of mTOR signalling are needed for Wnt-high zone-3 to be tumourigenic. Contrary to MAPK, high Wnt pathway activation, as seen in *Apc* mutants, strongly promote zone-3 differentiation, which again is less permissive to cancer. In the normal hepatic lobule these two signalling modalities are compartmentalised in to distinct regions. It is only in the lesions, where these signalling pathways are dysregulated, do they converge to activate mTOR and drive a proliferative translatoome that can sustain growth. In the discussion we have now clearly explained the link between zonation factors and tumourigenesis, please see text below (lines in manuscript 323-326)

“suppression of Wnt-driven differentiation to an extreme zone-3 fate by aberrant MAPK activation enables expression of the zone-2 molecular machinery, which facilitates hepatocyte growth in the liver via its engagement of the mTOR pathway”

2.3)

3) It would be important to show, for Figures 3 and 5, how many of the lesions and endpoint tumors present intermediate or high levels of Wnt activation.

We thank the reviewer for their feedback. End stage tumours in the *Ctnnb1^{ex3/WT};R26^{LSL-MYC}* model have similar levels of nuclear CTNNB1 to the day 60 lesions. Strikingly, the infrequent end-stage tumours that develop in the *Apc^{fl/fl};R26^{LSL-MYC}* model can have high levels of nuclear

Figure: Representative images of CTNNB1 immunohistochemistry of end point *Ctnnb1^{ex3/WT};R26^{LSL-MYC}* and *Apc^{fl/fl};R26^{LSL-MYC}* liver tumours. Dashed line represents tumour border, T=tumour and N=normal tissue. Scale bars, 100 μ m.

CTNNB1. However, in this setting the morphology of the cancer cells is changed, they appear smaller and less differentiated as shown in the image panel below. This suggests that the Wnt-high mutant clones that do progress to tumours escape the differentiated state via an alternative route in the *Apc^{fl/fl}* model. These images have now been added to Fig.5g.

2.4)

4) What happens to hepatocytes with overactivated Wnt signaling? Do they become terminally differentiated or undergo cellular senescence? It is not very clear what happens to zone 3 Glul-positive Lgr5-positive hepatocytes after *Ctnnb1*-MYC expression or zone 1-2 hepatocytes after

Wnt overactivation. They seem to be polyploid, which has been associated with senescence. Either way, those cells should be characterized better. It could also be that 3 Glul-positive Lgr5-positive hepatocytes and zone 1-2 hepatocytes have a different fate upon Wnt overactivation.

Figure: CTNNB1, BrdU, p21 and p16 immunohistochemistry on serial sections from a day 60 *Ctnnb1*^{ex3/WT};*R26*^{LSL-MYC} liver induced at a low clonal density with AAV8.TBG.Cre. Arrowheads represent single mutant clones and dashed black lines indicate a proliferative lesion. Scale bar, 100 μ m.

We thank the reviewer for this important question. In the *Ctnnb1*^{ex3/WT};*R26*^{LSL-MYC} mutant clone model at day-60, the single clones with over-activated Wnt signalling are terminally differentiated and not senescent. We observe a mixture of both diploid and polyploid single mutant clones at day-60 but importantly none of them express senescent markers. We have demonstrated this with immunohistochemistry (please see images above) of serial sections from day-60 *Ctnnb1*^{ex3/WT};*R26*^{LSL-MYC} mutated liver. CTNNB1^{pos} Single clones (highlighted by black arrowheads) are neither p21 nor p16 positive. We do observe p21 positivity in the lesions (highlighted by the dashed line) but this rarely co-occurs with BrdU labelling and does not extend to p16 expression. These images have now been included in the manuscript and can be found in Extended Data Fig. 1g.

2.5)

5) Could it be that *Ctnnb1*-MYC tumors downregulated liver-specific Wnt signaling but upregulate canonical Wnt signaling?

We thanks the reviewer for their question. Although not included in the manuscript we did explore a large range of Wnt associated gene signatures in the previous rebuttal letter for reviewer 1. This included gene sets identified from mutant *Apc*-colorectal adenomas, β -

Figure : Gene set enrichment analysis (GSEA) comparing various Wnt gene signatures in the day-60 from the spatial transcriptomics assay.

Catenin, Lef1 and Tcf4 ChIP-Seq and a consensus gene set generated from multiple Wnt-mutant tissues. Generally, the “canonical” Wnt gene signatures were enriched in the single clones but this was not significant for all gene sets. Please find above the figure panel presenting these GSEA plots. We therefore conclude that down-regulation of wnt signalling is not specific to liver associated Wnt signalling.

2.6)

The study starts by studying the effects of MYC, Ctnnb1, and combine Ctnnb1-MYC expression effects on cell proliferation. This part of the study (most of figure 1 and 2) is not too robust and is quite confusing, distracting from the main message of the study. The authors could consider restructuring the paper to focus the message on the most significant findings. 1) For example, Ctnnb1 activation in zone 3 Glul-positive Lgr5-positive hepatocytes do not lead to BrdU incorporation while Ctnnb1-MYC expression promotes proliferation. However,

that proliferation does not end up on tumor formation. The conclusion from this section of the study, “Together these data suggest that an extreme zone-3 Lgr5^{pos} hepatocyte fate is not permissive to oncogenic Wnt-driven growth” is not accurate since there is proliferation with Ctnnb1-MYC expression.

2) It is not clear why Ctnnb1 activation from different models leads to the same pattern of proliferation. Is it due to tamoxifen? Soluble factors? Background signal?

We thank the reviewer for their considered suggestion, we have restructured the original version of the paper. We now believe this new version of the manuscript is clearer and easier to read and better highlights the key findings of the paper.

In regards to the distribution of proliferation with the various zonal-Cres. If it was solely due to tamoxifen we would expect to see similar proliferation patterns in the Cyp1a2-WT and Igfbp2-WT groups. The region, zone-3 transitioning to Zone-2, is intriguing as mutant-Ctnnb1 seems to induce a transient proliferative response, this region is outside of that extreme Lgr5^{pos} GLUL^{pos} population. It has been documented that zone-3 has two distinct populations (Hiang Ang et al., Cell Reports, 2025, PMID: 39721024) it is therefore possible that the less extreme region of zone 3 adjacent to the Zone-2 is permissive to Wnt- driven proliferation.

Minor concerns:

2.7)

1) It would be important to highlight that Lgr5-CreER-Ctnnb1-MYC mice die to intestinal tumors, both in the text and the graph. Maybe plot “liver tumor free survival”?

We thank the reviewer for their feedback this has been changed in the figure and text. The text now reads (lines in manuscript 342-343):

“As expected, the central venous, Lgr5^{pos} Ctnnb1^{ex3/WT};R26^{LSL-MYC} mutants did not form tumours in the liver but eventually developed intestinal polyps”

2.8)

2) “Further to this, it appears that the heightened level of oncogenic Wnt-pathway activation, generated by Apc loss, is also not compatible with tumourigenesis in the liver, consistent with the relative absence of APC mutations in HCC compared, for example, with colorectal cancer.” Maybe say “less compatible” since the Apc model does end up with tumors.

We thank the reviewer for their feedback this has been changed in the text. The text now reads (lines in manuscript 296-298):

“Additionally, it appears that the heightened level of oncogenic Wnt-pathway activation, generated by Apc loss, is also less compatible with tumourigenesis in the liver, consistent with the relative absence of APC mutations in HCC.”

2.9)

3) “Over time (days 4–10 post induction), two notable features became sequentially apparent: firstly, the increase in hepatocyte proliferation was transient, occurring predominantly in GLUL^{neg} zone-1 -2 hepatocytes around day 4 and diminishing by day 10 post induction

(Fig. 1d,e and Extended Data Fig. 2d,e). Secondly, this transient proliferative response was succeeded by marked expansion of the zone-3 GLUL-expressing domain along the liver-lobule axis, underpinned by a significant increase in the levels of zone-3 gene transcripts (Fig. 1f and Extended Data Fig. 2b,c).” Is this expansion of pre-existent Glul-positive cells or late zone-3 induction of zone 1-2 hepatocytes? Please, clarify.

We thank the reviewer for their comment and have clarified that it is a progressive induction of zone 3 markers that starts from the CV and overtime extends towards the PN. We have adjusted the text in the manuscript to make this clearer. It now reads (lines in manuscript 171-173):

“Secondly, this transient proliferative response was succeeded by marked up-regulation of the zone-3 GLUL-expressing domain along the liver-lobule axis”

Referee #3 (Remarks to the Author):

The authors have added a convincing and impressive set of data with which they have greatly improved the manuscript. They included analysis of different inducible and zonally restricted Cre mouse models that strongly substantiate the observation that zone 3 hepatocytes are non-permissive for oncogenic transformation. Furthermore, they now provide an in-depth analysis of how Igfbp2 and mTOR signaling promote growth in Ctnnb1/Myc-mutant hepatocytes. Also, new results were added that potentially provide an explanation of how Ctnnb1 target expression may be reduced in lesions versus clones. Finally, how the experimental setup and results support the main model and how these findings relate to existing literature is now better explained and discussed.

Taken together, the authors have satisfactorily addressed my concerns. I think the revised manuscript has been greatly strengthened and presents convincing support for an original and important novel concept that will be of interest to researchers working in diverse areas, including liver homeostasis, cancer biology, cancer genetics and cell biology.

We thank the reviewer for their comments and constructive feedback, which has significantly strengthened the manuscript.